# CausalProfiler: Generating Synthetic Benchmarks for Rigorous and Transparent Evaluation of Causal ML

## Abstract

Causal machine learning (Causal ML) aims to answer "what if" questions using machine learning algorithms, making it a promising tool for high-stakes decision-making. Yet, empirical evaluation practices in Causal ML remain limited. Existing benchmarks often rely on a handful of hand-crafted or semi-synthetic datasets, leading to brittle, non-generalizable conclusions. To bridge this gap, we introduce CausalProfiler, a synthetic benchmark generator for Causal ML methods. Based on a set of explicit design choices about the class of causal models, queries, and data considered, the CausalProfiler randomly samples sets of data, assumptions, and ground truths constituting the synthetic causal benchmarks. In this way, Causal ML methods can be rigorously and transparently evaluated under a variety of conditions. This work offers the first random generator of synthetic causal benchmarks with coverage guarantees and transparent assumptions operating on the three levels of causal reasoning—observation, intervention, and counterfactual. We demonstrate its utility by evaluating several state-of-the-art methods under diverse conditions and assumptions, both in and out of the identification regime, illustrating the types of analyses and insights the CausalProfiler enables.

## 1 Introduction

Causal machine learning (Causal ML) seeks to estimate the effects of interventions and counterfactuals using machine learning techniques [28], enabling principled decision making in medicine, policy, and other high-stakes domains. Despite its theoretical maturity and growing relevance, Causal ML remains underutilized. A key barrier to adoption lies in the current empirical evaluation landscape, which is unable to support meaningful and generalizable evidence of method performance [40].

Recent critiques of evaluation practices in both predictive Machine Learning (ML) [21, 13, 30] and causal inference [10, 14, 6] have highlighted systemic shortcomings. Lessons from predictive ML show that narrow, static benchmarks can give a false sense of reliability [16, 21], underscoring the need for structured diversity: systematic variation of tasks under explicit, controllable assumptions. In the case of causal ML, evaluation is fundamentally more challenging due to the unobservability of counterfactual outcomes [23]. Hence, researchers can only rely on scarce real-world data sources. Typically, randomized controlled trials, considered as the gold standard, are expensive, ethically constrained, and often encompass a low amount of data [18, 44]. As a result, existing benchmarks often rely on a small number of semi-synthetic datasets (e.g., IHDP [22], Twins [32]) or model-driven synthetic datasets generated from fitted causal mechanisms [33, 34, 3, 12]. These datasets typically encode specific assumptions—such as structural constraints, identifiability conditions, or narrow function classes—which are rarely made explicit and are difficult to generalize beyond the original study context. Moreover, handcrafted synthetic datasets, where researchers explicitly define causal models and choose evaluation queries, are frequently designed with specific hypotheses or methods

Submitted to 39th Conference on Neural Information Processing Systems (NeurIPS 2025). Do not distribute.

in mind [17], risking bias through overfitting or implicit alignment. It has been argued [40] that the problem is not synthetic evaluation itself but the lack of rigor in its design and interpretation.

In this work, we take a concrete step toward addressing these concerns by introducing a synthetic benchmark generator, called CausalProfiler, that enables empirical evaluations grounded in transparently defined synthetic causal datasets. Central to our approach is the notion of a *Space of Interest* (*SoI*) (Definition 5.1), defining the domain from which causal datasets are sampled. Given a specified *SoI*, our benchmark generator samples SCMs, data, and queries, and estimates the ground truth value of the queries to enable the evaluation of Causal ML methods.

This approach departs from existing benchmarks in several key ways. Rather than evaluating methods on a fixed or narrow set of datasets, our framework enables controlled, repeatable, and diverse sampling over structured families of tasks. It replaces opaque or implicit design choices with fully transparent specifications of model structure, noise, and query types. Crucially, it shifts the focus of empirical evaluation from performance on individual datasets to trends and patterns across a well-characterized *SoI*, reframing the evaluation question from "what dataset to use" to specifying a *SoI* that defines the scope of evaluation. This enables researchers to evaluate not only performance but also under which conditions—on graph density, or causal mechanisms complexity for instance—a method succeeds or fails and helps practitioners identify methods that remain reliable when causal assumptions are likely violated. By aggregating results over many generated datasets, the evaluation yields more robust and reliable performance estimates, helping to uncover failure modes, generalization limits, and assumption sensitivities that remain hidden in conventional evaluations.

**Contributions.** This paper offers the first open-source implementation of such an evaluation framework[1], and illustrates its utility by comparing state-of-the-art causal ML methods across multiple synthetic causal datasets. Our contributions are twofold. First, we present a benchmark generator (Section 5) that enables principled sampling of synthetic causal datasets over user-defined *SoIs*, with built-in coverage guarantees that promote transparency and reproducibility. Secondly, we demonstrate through experiments (Section 6) how evaluation across different *SoIs* yields richer and more robust insights than single-dataset evaluations.

## 2 Related Work

**Evaluating causal ML methods.** Causal ML currently lacks a rigorous, systematic paradigm for empirical evaluation. Indeed, the community has largely turned to synthetic and semi-synthetic benchmarks. Semi-synthetic datasets, such as IHDP [22] and Twins [32], combine real covariates with simulated outcomes under assumed structural models. Fully synthetic datasets, in contrast, are generated entirely from researcher-defined SCMs, allowing for greater control and access to ground truth. Yet both synthetic and semi-synthetic approaches suffer from critical limitations.
First, synthetic evaluations often lack realism, relying on overly simplistic mechanisms such as additive noise or linear functions, and frequently omitting robustness analyses [17, 10, 39, 40]. These experiments rarely reflect the complexity of real-world causal processes and are insufficient to test the limits of modern causal inference methods.
Secondly, synthetic and semi-synthetic datasets are shaped by researcher-defined design decisions, including the structure of the causal graph, the form of the outcome function, and the noise distribution. These decisions, often made implicitly, can unintentionally introduce hidden biases that favor certain methods [9, 8, 14]. Such assumptions are rarely documented or systematically varied, hindering reproducibility and fair method comparison [39, 40].
Additionally, these benchmarks are typically small in scale and narrow in scope, often covering only a limited range of causal settings. As a result, empirical evaluations raise concerns about overfitting and generalization [17, 6]. For instance, it has been shown that even small changes to the data-generating process can lead to dramatic shifts in performance rankings [9]. Moreover, methods are often evaluated only under the very conditions that guarantee their identifiability, offering little insight into robustness under assumption violations, as is common in real-world settings [40, 38, 26].
In short, without broader and more transparent evaluation across diverse causal settings, the field risks drawing conclusions that do not generalize. Addressing this gap requires moving beyond fixed benchmarks toward frameworks that support transparent, structured, and diverse experimentation across well-defined spaces of causal assumptions.

[1]The code is provided in the supplementary material and will be publicly available after the review process.

**Recent benchmarking efforts.** Recent works have sought to address some of these gaps introducing tools to generate synthetic SCMs for causal discover [29, 19, 41] or support query estimation from hand-specified models [42, 45, 1]. However, none of these frameworks support all components required for robust evaluation of causal machine learning methods. First, causal discovery benchmarks [29, 19, 41] do not compute ground truth for intervention or counterfactual queries. Further, query estimation frameworks [42, 45, 1] often require manual specification of the SCM and do not support random sampling, diversity control, or analysis of the distribution of tasks. Even in cases where SCMs are sampled [41, 48], key properties (e.g., positivity) are neither reported nor constrained. In addition, the absence of randomness in the graph structures limits generalization. Our approach complements and extends these efforts by integrating SCM sampling, query ground truth computation, and coverage guarantees into a unified framework. To the best of our knowledge, this is the first benchmark generator that enables systematic exploration of how Causal ML methods behave across spaces of SCMs and queries defined by user-specified constraints.

## 3 Background & Notation

We use capital letters for random variables (e.g., $X$), lowercase for realizations (e.g., $x$), and boldface for vectors (e.g., $\mathbf{x}$). For a more complete background, please refer to Appendix A and Pearl [35].

**Causal Hierarchy.** The Pearl Causal Hierarchy (PCH) [36] classifies causal questions into three levels: $\mathcal{L}_1$ (associational), $\mathcal{L}_2$ (interventional), and $\mathcal{L}_3$ (counterfactual). While associative questions rely only on observed data, interventional and counterfactual questions require assumptions about the data-generating process. Importantly, lower layers are insufficient to answer higher-layer questions in almost all causal models [4].

**Structural Causal Models.** A Structural Causal Model (SCM) [35] is a tuple $\mathcal{M} := \{\mathbf{V}, \mathbf{U}, \boldsymbol{\mathcal{F}}, P(\mathbf{U})\}$, where $\mathbf{V}$ are endogenous variables, $\mathbf{U}$ are exogenous variables, $\boldsymbol{\mathcal{F}}$ is a set of structural equations $V_i = f_i(\boldsymbol{PA}(V_i), \mathbf{U}_{V_i})$, and $P(\mathbf{U})$ defines a distribution over the exogenous variables. SCMs induce a distribution $P_{\mathcal{M}}(\mathbf{V})$ over the endogenous variables. In this work, we additionally consider two types of endogenous variables: the observed variables, denoted $\mathbf{V}_O$, and the unobserved variables, denoted $\mathbf{V}_H$ with $\mathbf{V} = \mathbf{V}_O \cup \mathbf{V}_H$ and $\mathbf{V}_O \cap \mathbf{V}_H = \emptyset$.

**Causal Graphs.** We represent causal relationships using the *causal graph* of a Structural Causal Model (SCM). This is a directed acyclic mixed graph over the endogenous variables. Directed edges $X \to Y$ encode causal dependencies via structural equations, while bidirected edges $X \leftrightarrow Y$ indicate latent confounding due to shared exogenous causes.

**Interventions.** An *intervention* replaces one or more structural equations to model external manipulations. A common example is a *hard intervention*, written $\boldsymbol{do}(T = t)$, which sets a variable to a fixed value, disconnecting it from its natural causes. This defines a new SCM and alters the induced distribution.

**Counterfactuals.** *Counterfactual queries* reason about what would have happened under a different intervention, given an observed outcome called a factual realization. They are evaluated by conditioning on observed variables (abduction), modifying the SCM (action), and predicting outcomes under the new distribution (prediction)—a process known as the *three-step procedure* [35].

**Causal Queries.** A *causal query* refers to a probabilistic statement about the effect of hypothetical manipulations of the data-generating process. This includes *intervention queries*, such as Average Treatment Effect (ATE), and *counterfactual queries*, such as Counterfactual Total Effect (Ctf-TE).

**Identifiability.** A query is *identifiable* if its value can be uniquely determined from data, given a set of assumptions (e.g., a causal sufficiency) [35]. In other words, identifiability determines whether causal queries can be empirically estimated, and under what assumptions.

## 4 Problem Formulation

We consider the problem of causal inference, where the goal is to answer interventional and counterfactual queries using data drawn from an unknown SCM. Let $\mathcal{M}^\star = (\mathbf{V}, \mathbf{U}, \boldsymbol{\mathcal{F}}, P(\mathbf{U}))$ denote the unknown ground truth SCM, a causal query $Q$ (e.g., Average Treatment Effect (ATE)) is defined over $\mathcal{M}^\star$ and has a ground truth value $Q^\star = Q(\mathcal{M}^\star)$. As $\mathcal{M}^\star$ is unknown, causal estimators rely

on causal assumptions $\mathbf{H}$ (e.g., causal sufficiency) and available data $D$ drawn from $\mathcal{M}^\star$ to produce an estimate $\hat{Q}$ of the target quantity $Q^\star$. We introduce Definition 4.1 to formalize the elements of a causal dataset.

> **Definition 4.1** (Causal Dataset)**.** A **causal dataset** is a tuple $\mathcal{D} = \{Q, Q^\star, D, \mathcal{G}^\star, \mathbf{H}^\star\}$ constructed from a known SCM $\mathcal{M}^\star = (\mathbf{V}, \mathbf{U}, \boldsymbol{\mathcal{F}}, P(\mathbf{U}))$ where:
> - $Q$ is a causal query defined over $\mathbf{V}$;
> - $Q^\star = Q(\mathcal{M}^\star)$ is the exact value of the query $Q$;
> - $D = \{D_k \sim P_{\mathcal{M}^\star}(\mathbf{V} \mid \boldsymbol{do}(\mathbf{V}_k) = \mathbf{v}_k)\}_{k=1}^{I}$ is a collection of samples under $I$ interventional (or observational) settings;
> - $\mathcal{G}^\star$ is the causal graph associated with $\mathcal{M}^\star$;
> - $\mathbf{H}^\star$ is the set of assumptions satisfied by $\mathcal{M}^\star$.

Given a causal dataset $\mathcal{D} = (\{Q, Q^\star, D, \mathcal{G}^\star, \mathbf{H}^\star\})$, one can compute the estimation error $E(\hat{Q}, Q^\star)$ using a chosen error metric $E$ (e.g., squared error). As a result, one can evaluate causal ML methods in the identification-consistent setting—where the considered causal graph and assumptions match the ground truth ones, i.e., $\mathcal{G}^\star$ and $\mathbf{H}^\star$—but also test robustness by introducing assumption violations.

# 5 Sampling Causal Datasets with the CausalProfiler

To generate causal datasets, CausalProfiler relies on a parametric specification of the sampling domain, called the *Space of Interest*. Given an *SoI*, it samples an SCM (Section 5.2) and generates a corresponding causal dataset (Section 5.3). Appendices B, C, D, and E contain pseudocode for the sampling algorithms, and Appendix H presents a visual overview of the sampling strategy.

## 5.1 Defining a Space of Interest

The central abstraction of our framework is the *Space of Interest* (Definition 5.1), which provides a standardized way to specify synthetic causal datasets (Definition 4.1).

> **Definition 5.1** (Space of Interest)**.** A **Space of Interest** (*SoI*) is a tuple $\mathcal{S} = \{\mathbb{M}, \mathbb{Q}, \mathbb{D}\}$, where $\mathbb{M}$ is a class of SCMs, $\mathbb{Q}$ a class of causal queries, and $\mathbb{D}$ a class of data.

Table 3 in Appendix B lists all configurable *SoI* parameters.[2]

## 5.2 Sampling Structural Causal Models

CausalProfiler samples SCMs from a user-defined *SoI* in two steps: (i) sampling a causal graph, and (ii) sampling the corresponding mechanisms.

**Causal Graphs.** CausalProfiler first samples a Directed Acyclic Graph over a set of endogenous variables, which defines the causal structure of the SCM. Second, if specified in the SoI, CausalProfiler samples a subset of endogenous variables, $\mathbf{V}_H$, to be treated as unobserved and excluded from the observed dataset. To expose only the visible causal structure to the user, we apply Verma's latent projection algorithm [47] to the full causal graph, which produces an Acyclic Directed Mixed Graph.

**Mechanisms.** Given the causal graph, CausalProfiler assigns a mechanism to each endogenous variable given its parents and an exogenous noise whose distribution is set by the *SoI*. We support two types of mechanisms. First, **discrete mechanisms**, also called Regional Discrete mechanisms (see Appendix D.1 for a formal definition), are defined tabularly by associating each element of a partition of the exogenous noise with distinct parents-to-child mappings. This allows for controllable stochasticity and complexity, supporting highly non-linear and non-invertible behavior. Second, **continuous mechanisms** are defined using parametric function families—such as neural networks or linear functions—with randomly initialized parameters (e.g., He initialization [20]).

[2]While the current implementation of CausalProfiler supports only $\mathcal{L}_1$ training data and ATE, CATE, and CTF-TE queries, the *SoI* abstraction can, in principle, be defined over any class of queries, datasets, and SCMs.

### 5.3 Sampling Causal Datasets

**Data $D$.** Given an SCM $\mathcal{M}^\star$ sampled from the *SoI*, we generate an observational dataset $D$ by sampling i.i.d. data points from the entailed distribution of $\mathcal{M}^\star$ over observed variables. This involves forward-sampling from the structural equations in topological order, using the noise distributions specified for each variable and marginalizing out any latent variables.

**Query $Q$.** We first sample endogenous observable variables to play the role of treatment, outcome, covariates, and factuals, depending on the class of queries of the *SoI*. To ensure that queries are well-defined and empirically grounded, we draw realizations from a large, separately sampled observational dataset, rather than from the theoretical variable domains. This avoids defining queries on realizations that may be unrepresentative or impossible under the SCM. While the currently implemented queries only involve interventions and counterfactuals, CausalProfiler also supports benchmarking causal discovery methods as the ground-truth causal graph $\mathcal{G}^\star$ is directly provided in the causal dataset.

**Query ground truth $Q^\star$.** Each query is estimated by drawing samples from the (manipulated) ground truth SCM: interventional queries via do-operations (action and prediction), and counterfactual queries via the three-step procedure [35]. Queries that are duplicates or yield `NaN` estimates are rejected and resampled to ensure valid and computable values.

**Ground truth causal graph $\mathcal{G}^\star$.** As presented in Section 5.2, $\mathcal{G}^\star$ is built as the latent projection of the ground truth SCM's causal graph over the observed variables.

**Ground truth Causal Assumptions $\mathbf{H}^\star$.** To characterize the properties of the ground-truth SCM from the user's perspective, we provide an analysis module that computes summary metrics related to common causal assumptions (e.g., measuring linearity via Pearson correlation). A full list of available metrics is provided in Appendix F.

**Coverage guarantee.** Theorem 5.1 (proof in Appendix I) shows that, with sufficiently expressive discrete mechanisms, CausalProfiler's sampling strategy can theoretically generate any causal dataset within a given SoI, guaranteeing $\mathcal{L}_3$-expressivity. In addition, Appendix G provides an analysis exploring the empirical distribution of the sampled datasets.

**Theorem 5.1** (Coverage)**.** For a Space of Interest $\mathcal{S} = \{\mathbb{M}, \mathbb{Q}, \mathbb{D}\}$, whose class of Structural Causal Models is a class of Regional Discrete SCMs[1] with the maximum number of noise regions, denoted $\mathbb{M}_{\texttt{RD-SCM},r=R_{\max}}$, any causal dataset $\mathcal{D} = \{Q, Q^\star, D, \mathcal{G}^\star, \mathbf{H}^\star\}$ has a strictly positive probability to be generated.

$$\forall \mathcal{S} = \{\mathbb{M}, \mathbb{Q}, \mathbb{D}\}\ s.t.\ \mathbb{M} \subseteq \mathbb{M}_{\texttt{RD-SCM},r=R_{\max}},\ P(\mathcal{D}|\mathcal{S}) > 0$$

[1] Formal definition can be found in Appendix D.1.

**Benchmark Design.** Taken together, these design choices reflect four key properties that are considered essential for rigorous synthetic evaluation in causal ML [40]: **transparency**, by making all assumptions explicit via the parametrization of the SoI, which serves as a declarative specification of the evaluation domain; **repeatability**, through randomized but seed-controlled sampling procedures, ensuring that SCMs and queries can be exactly reproduced across runs; **bias awareness**, supported by the coverage guarantee and the empirical distribution analysis module and **control over experiments**, by exposing a wide range of configurable parameters in the *SoI* that allow users to tailor the causal dataset generation to their assumptions and research goals.

## 6 Experiments

### 6.1 Verification of Benchmark Correctness

To validate the soundness of our benchmark generator, we perform consistency checks based on the three levels of the Pearl Causal Hierarchy [36, 4]. Using the SCM sampler and query estimator of the CausalProfiler, we evaluate whether sampled SCMs satisfy the Markov condition, do-calculus rules, and structural counterfactual axioms [35]. We use discrete SCMs to enable exhaustive enumeration of conditioning sets for statistical tests. To ensure robustness, we iterate over a *SoI* parameter grid

spanning the number of variables, edge density, cardinalities, noise regions, and dataset sizes. For each configuration, we sample five SCMs. See Appendix J for full details and results.

**L1: Markov Property Verification.** We assess whether d-separations in the causal graph imply conditional independencies in the entailed observational distribution of the sampled SCMs. For each SCM, we enumerate d-separated triplets $(A, B, C)$ and test whether $A \perp B \mid C$ holds using Pearson's $\chi^2$ test [37]. We filter low-sample strata (Koehler [31]) and correct for multiple tests (BH [5]). The Markov property holds in roughly 95% of tested cases, with most violations attributable to finite-sample variability (see Table 4, Appendix J).

**L2: Do-Calculus Verification.** We test whether the three rules of do-calculus hold empirically. For each rule, we identify variable tuples that satisfy the rule's graphical preconditions. We then use the query estimator to generate two interventional datasets corresponding to the left- and right-hand sides of the rule. We use these datasets to compare the two distributions using Pearson's $\chi^2$ test, filtering low-sample strata (Koehler [31]) and correcting for multiple tests (BH [5]). Around 5.5% of tests fail, with discrepancies largely due to finite-sample noise (see Table 5, Appendix J).

**L3: Structural Counterfactual Axiom Verification.** We verify whether the axioms of *composition*, *effectiveness*, and *reversibility* hold exactly for sampled SCMs. Since the axioms involve deterministic functional relationships, we only count exact matches of the query estimator. All axioms hold exactly across our samples, confirming the estimator's consistency with structural counterfactual semantics.

### 6.2 Setup for Experiments using the CausalProfiler

We demonstrate the utility of our benchmark framework by evaluating several recent causal inference methods across a diverse set of *SoIs*. Our goal is not to exhaustively benchmark each method but to showcase the types of structured empirical investigations our framework enables — especially those exploring robustness and violations of causal assumptions.

**Evaluation Protocol.** All evaluations follow the process detailed in Algorithm 1. For each *SoI*, we evaluate each method using five random seeds. For each seed, we sample 100 SCMs. For each SCM, we generate one training set and five causal queries with ground-truth values. Results are aggregated across SCMs and seeds, enabling a rigorous and reproducible assessment of performance.

**Algorithm 1** Evaluation process for causal machine learning methods

```
Input: List of Spaces of Interest SoIs, list of seeds seeds number of examples per SCM
    num_examples
Initialize: method ← CausalMLMethod()
for each SoI in SoIs do
    for each seed in seeds do
        setGlobalSeed(seed)
        for each examples in num_examples do
            Generate samples, queries, and targets from the profiler
            Get estimates using the method on the generated samples and queries
            Calculate (and store) error by comparing estimates with targets
        end for
        Compute performance statistics for seed
    end for
    Compute performance statistics for SoI
end for
Output: Final summary with evaluation results
```

**Hardware.** All experiments were run on a single machine equipped with an Intel Core i9-14900K processor (24 cores, 32 threads) and 96GB of RAM. All CPU threads were utilized for parallel processing where applicable. Some methods (e.g., DCM) would benefit from GPU acceleration, which was not used in our evaluation.

**Experiment Types.** We perform two main sets of experiments: (1) ATE estimation over a set of continuous SCMs, and (2) counterfactual query estimation on discrete-variable SCMs. An additional experiment on ATE estimation under varying levels of hidden confounding is included in Appendix K.3.

**Metrics and Visualization.** We evaluate methods using the mean squared error between predicted and true query values. For each method and *SoI*, we report mean error, standard deviation, total runtime, and failure rate (i.e., the proportion of queries for which no valid output was returned due to numerical issues or exceptions). Tables include all numeric summaries, while box plots visualize error distributions via median, interquartile range (IQR), and whiskers extending to $1.5\times$IQR.

**Methods.** We evaluate several causal inference methods: Causal Normalizing Flows (CausalNF) [27], Neural Causal Models (NCM) [11], Variational Causal Graph Autoencoder (VACA) [43], Diffusion-based Causal Models (DCM) [7], and Deconfounding Causal Normalizing Flows (DeCaFlow) [2].

Additional experiment details, results, and *SoI* configurations are provided in Appendix K.

### 6.3 Experiment 1: General Evaluation across Diverse SCMs

To showcase the flexibility of our benchmarking framework, we evaluate VACA, CausalNF, DCM, and NCM on a set of continuous-variable SCMs. These experiments are designed to highlight how performance can vary across diverse SoIs. See Table 1 for a summary of results and Figure 1 for a box plot of ATE estimation errors.

**Spaces of Interest.** We evaluate methods on four distinct *SoIs*: **Linear-Medium**, with linear SCMs (15-20 nodes, 1000 samples); **NN-Medium**, with neural SCMs using a 2-layer ReLU network (8 hidden units per layer, 15-20 nodes, 1000 samples); **NN-Large**, with larger neural SCMs (20-25 nodes, 1000 samples); and **NN-Large-LowData**, identical to NN-Large but with only 50 samples.

Table 1: Performance summary of CausalNF, DCM, NCM, and VACA on the general experiments.

| Space | Method | Mean Error | Std Error | Max Error | Runtime (s) | Fail Rate (%) |
|---|---|---|---|---|---|---|
| `Linear-Medium` | CausalNF | 0.4625 | 0.8985 | 9.6079 | 13790.4 | 0.00 |
| `Linear-Medium` | DCM | 0.1530 | 1.5289 | 33.9766 | 16541.2 | 0.00 |
| `Linear-Medium` | NCM | 0.4618 | 0.9001 | 9.6134 | 7384.7 | 0.00 |
| `Linear-Medium` | VACA | 0.4209 | 0.6195 | 2.3807 | 2734.5 | 53.40 |
| `NN-medium` | CausalNF | 0.0160 | 0.0107 | 0.1209 | 10732.7 | 0.00 |
| `NN-medium` | DCM | 0.0276 | 0.0114 | 0.0746 | 15894.4 | 0.00 |
| `NN-medium` | NCM | 0.0111 | 0.0121 | 0.1484 | 7322.8 | 0.00 |
| `NN-medium` | VACA | 0.0090 | 0.0077 | 0.0479 | 5759.6 | 5.00 |
| `NN-Large` | CausalNF | 0.0159 | 0.0105 | 0.1535 | 15114.8 | 0.00 |
| `NN-Large` | DCM | 0.0267 | 0.0100 | 0.0739 | 19166.2 | 0.00 |
| `NN-Large` | NCM | 0.0101 | 0.0103 | 0.1161 | 9450.6 | 0.00 |
| `NN-Large` | VACA | 0.0090 | 0.0094 | 0.0535 | 5690.8 | 11.60 |
| `NN-Large-LowData` | CausalNF | 0.0359 | 0.0146 | 0.1712 | 22138.2 | 0.00 |
| `NN-Large-LowData` | DCM | 0.0777 | 0.0445 | 0.3701 | 2412.1 | 0.00 |
| `NN-Large-LowData` | NCM | 0.0097 | 0.0107 | 0.1263 | 404.7 | 0.00 |
| `NN-Large-LowData` | VACA | 0.0103 | 0.0134 | 0.1043 | 5217.4 | 0.00 |

**Findings (Linear-Medium vs. NN-Medium).** In the `Linear-Medium` setting, DCM achieves the lowest average error (0.1530), indicating excellent performance. However, its error standard deviation is notably high (1.5289), driven by a few extreme outliers (max error 33.98). This implies that DCM is highly effective for most queries but may produce large errors in rare cases—potentially problematic in safety-critical applications which match this *SOI*. VACA performs competitively with lower max error and faster runtime, but suffers a high failure rate (53.4%) due to NaNs.
When moving to the `NN-Medium` setting, where the causal mechanisms are implemented as small neural networks, DCM's advantage disappears. VACA emerges as the best performer, achieving both the lowest error mean (0.0090) and standard deviation (0.0077), while also reducing its failure rate to 5%. Interestingly, DCM becomes the weakest performer in this setting, highlighting that method rankings are highly sensitive to the underlying functional form of the mechanisms. This underscores the need for practitioners to evaluate methods within the *SoI* most relevant to their application.

**Findings (NN-Large vs. NN-Large-LowData).** In the second comparison, we increase SCM size to 20-25 nodes and investigate the effect of reducing data availability. Comparing `NN-Large` (1000 samples) to `NN-Large-LowData` (50 samples), we find that DCM is strongly affected by the data limitation: its error nearly triples (from 0.0267 to 0.0777) and its IQR expands noticeably. CausalNF also shows increased sensitivity to low-data regimes.

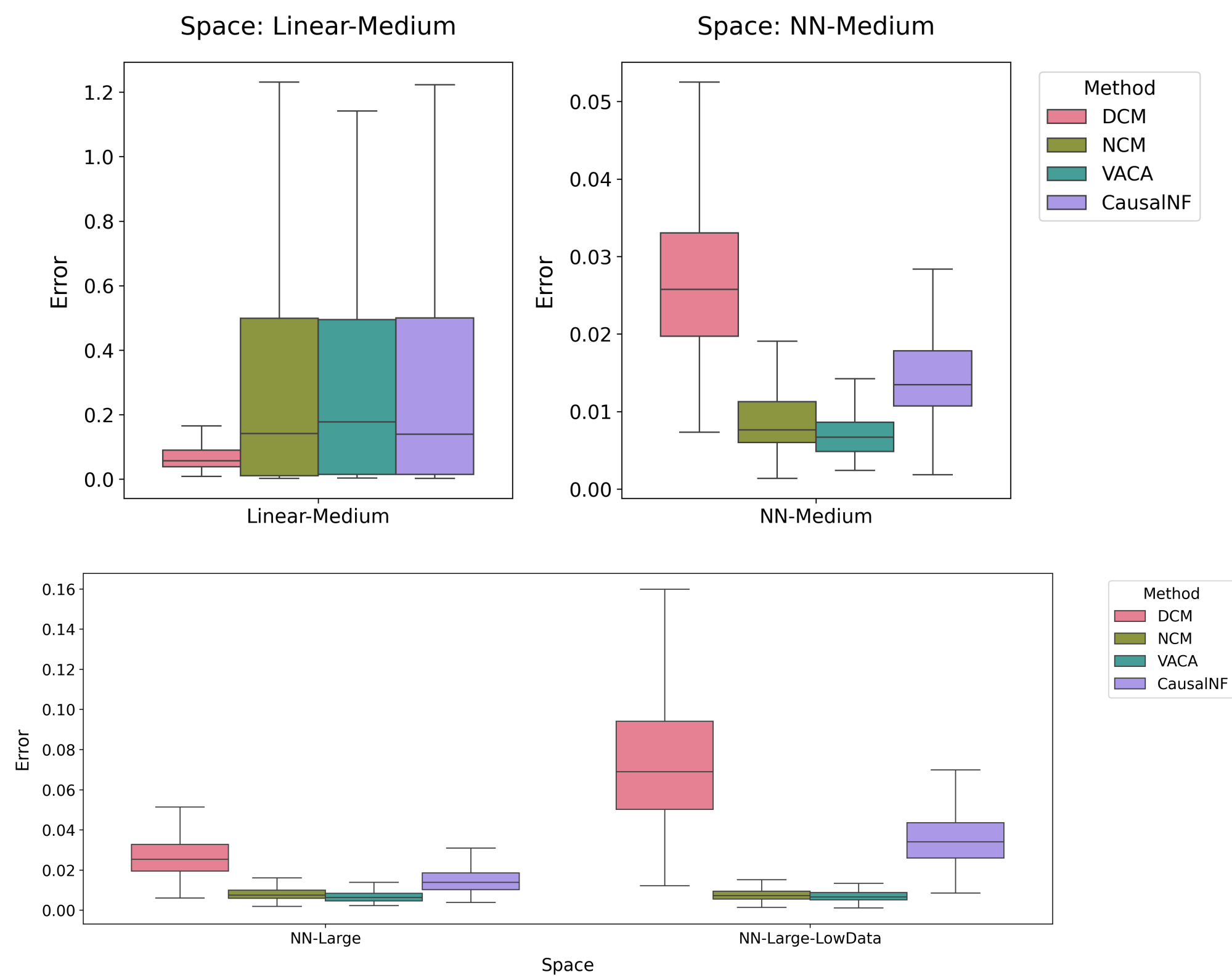

Figure 1: Box plots showing ATE estimation errors across different *SoIs*.

In contrast, both VACA and NCM maintain stable performance, with nearly unchanged mean and standard deviation values between the two *SoIs*. Notably, VACA achieves a 0% failure rate in this setting, with unexpectedly strong robustness under limited data.

**Insights.** While not intended as a comprehensive benchmark, these experiments illustrate the types of insights that can be drawn using our framework. For the selected Spaces of Interest, we observe that DCM tends to perform well on average but can produce large outlier errors or become less stable in low-data settings. Conversely, VACA shows promising generalization even with limited data, though it occasionally fails on certain SCMs. These findings are specific to the *SoIs* we explored, and should not be taken as general conclusions about the methods. Instead, they highlight how our framework enables structured, SoI-specific evaluations, helping practitioners assess which methods may be more suitable for their own modeling context.

### 6.4 Experiment 2: Counterfactual Estimation on Discrete SCMs

This experiment shows how our framework can evaluate counterfactual estimation methods on discrete-variable SCMs. We test CausalNF and DCM, originally designed for continuous settings, as a robustness check—motivated by prior work showing that CausalNF can sometimes effectively approximate discrete distributions [27, 11]. See Table 2 for a summary of results.

**Discrete SoIs.** We evaluate three discrete *SoIs*: **Disc-C2-Reject**, with 10-15 node graphs, binary variables, and rejection-based mechanism sampling; **Disc-C4-Unbias**, with the same graph size but 4-category variables and unbiased random mechanism sampling; and **Disc-Large-C2-Unbias**, which uses larger graphs (20-30 nodes), binary variables, and unbiased random mechanism sampling.

**Findings.** On `Disc-C2-Reject`, both CausalNF and DCM perform well and comparably, with low error means ($\sim$0.04) and low failure rates (8% for CausalNF, 4% for DCM). This suggests that both methods can produce reliable estimates even outside their original assumptions when the functional mechanisms are simple and binary.

Table 2: Performance summary of CausalNF and DCM on the discrete experiments.

| Space | Method | Mean Error | Std Error | Max Error | Runtime | Fail Rate |
|---|---|---|---|---|---|---|
| `Disc-C2-Reject` | CausalNF | 0.0415 | 0.1116 | 0.6240 | 212.8 s | 08.08 % |
| `Disc-C2-Reject` | DCM | 0.0424 | 0.1123 | 0.6240 | 4406.2 s | 04.28 % |
| `Disc-C4-Unbias` | CausalNF | 0.0431 | 0.1270 | 0.7071 | 190.7 s | 40.68 % |
| `Disc-C4-Unbias` | DCM | 0.0411 | 0.1199 | 0.7071 | 3839.4 s | 22.60 % |
| `Disc-Large-C2-Unbias` | CausalNF | NaN | NaN | NaN | 0.0 s | 100.00 % |
| `Disc-Large-C2-Unbias` | DCM | 0.0183 | 0.0814 | 0.5000 | 8192.7 s | 11.32 % |

However, when moving to `Disc-C4-Unbias`, where variables have 4 categories and mechanisms are sampled with unbiased random sampling, the failure rates increase significantly, especially for CausalNF, which fails on over 40% of SCMs (typically with NaN errors). This highlights how sensitive certain methods can be to changes in mechanism sampling or variable cardinality, even when mean errors remain similar.
To further probe robustness, we scale the graph size in `Disc-Large-C2-Unbias` while reverting to binary variables. CausalNF fails on all runs, returning NaNs and yielding a 100% failure rate. DCM remains functional, with an 11% failure rate, indicating greater resilience in this setting.

**Insights.** These results underscore the utility of our framework in systematically stress-testing methods beyond their nominal design assumptions. While CausalNF is not built for discrete data, prior examples suggested it could work in practice. Our benchmark can help clarify *when* and *how* it breaks: certain function classes and discrete configurations are more likely to cause divergence or failure. DCM appears more robust across these tests, though not immune. Importantly, this evaluation is not meant as a definitive comparison, but as a demonstration of how failure cases can be surfaced and studied in a principled way using the CausalProfiler.

## 7 Limitations and Future Work

**Causal Datasets Distribution.** While the coverage theorem guarantees that any causal dataset has a strictly positive probability of being sampled within a given *SoI* with sufficiently expressive discrete mechanisms, it does not give any information on the form of the distribution of the sampled causal datasets. In particular, certain classes of SCMs remain very unlikely to be sampled unless explicitly chosen in the *SoI* (e.g., linear SCMs). In addition, users should bear in mind that causal datasets are not uniformly generated when aggregating results, to avoid misleading interpretations. Future improvements may enable finer control over the datasets distribution and the underrepresented attributes when defining an *SoI*.

**Diversify Spaces of Interest.** Several directions remain open for extending the supported *SoI* by the CausalProfiler, such as support for mixed-variable SCMs, query identifiability diagnostics, sampling interventional training data, and more realistic data-generating scenarios, including selection bias, measurement noise, and partial knowledge of the causal graph.

**Towards realistic causal datasets.** More broadly, to increase real-world relevance, future work could enable users to define Spaces of Interest based on patterns observed in real data (e.g., a Bayesian approach), narrowing the gap between synthetic evaluation and practical deployment.

## 8 Conclusion

This work introduces CausalProfiler, a synthetic causal dataset generator for evaluating Causal Machine Learning methods across the three levels of the Pearl Causal Hierarchy. At its core is the notion of a Space of Interest, which replaces the ad hoc choice of a single evaluation dataset with a principled specification of the entire evaluation scope. This shift enables transparent, repeatable, and assumption-aware assessments under diverse causal conditions. We show that the performance of state-of-the-art Causal ML methods varies substantially across different SoIs, underscoring the importance of rigorous, distribution-level evaluation. CausalProfiler marks a first step toward more rigorous and systematic empirical practices in Causal ML—grounded not in fixed benchmarks, but in explicitly defined spaces that reflect the assumptions and structural properties relevant to each setting.

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

# A Additional definitions & Notations

**Definition A.1** (Semi-Markovian and Markovian SCMs)**.** An SCM is said to be **semi-Markovian** [35] if its set of structural equations is acyclic, meaning there exists an ordering of the equations such that for any two functions $f_i, f_j \in \mathcal{F}$, if $f_i < f_j$, then $V_j \notin \boldsymbol{PA}(V_i)$. This condition ensures that the causal dependencies among endogenous variables form a Directed Acyclic Graph (DAG).
An SCM is **Markovian** [35] if the exogenous variables influencing different endogenous variables are mutually independent. Formally, for all distinct $V_i, V_j \in \mathbf{V}$, we have $\mathbf{U}_{V_i} \perp\!\!\!\perp \mathbf{U}_{V_j}$. This implies the absence of latent confounding, allowing the model to be fully described by a DAG with independent noise terms.

**Definition A.2** (Causal Graph of a Semi-Markovian SCM)**.** The causal graph of a Semi-Markovian [4] SCM is n acyclic directed mixed graph (ADMG) with:

- Directed edge $V_i \to V_j$ if $V_i \in \boldsymbol{PA}(V_j)$
- Bi-directed edge $V_i \leftrightarrow V_j$ if $\mathbf{U}_{V_i} \not\perp\!\!\!\perp \mathbf{U}_{V_j}$

## A.1 Interventional Quantities ($\mathcal{L}_2$)

**Average Treatment Effect (ATE):**

$$\text{ATE}_{T\to Y} = \mathbb{E}[Y|\boldsymbol{do}(T=1)] - \mathbb{E}[Y|\boldsymbol{do}(T=0)]$$

**Conditional Average Treatment Effect (CATE):**

$$\text{CATE}_{T\to Y}(\mathbf{x}) = \mathbb{E}[Y|\boldsymbol{do}(T=1), \mathbf{X}=\mathbf{x}] - \mathbb{E}[Y|\boldsymbol{do}(T=0), \mathbf{X}=\mathbf{x}]$$

**Controlled Direct Effect (CDE):**

$$\text{CDE}_{T\to Y}(t, c, \mathbf{m}) = \mathbb{E}[Y|\boldsymbol{do}(T=t, \mathbf{M}=\mathbf{m})] - \mathbb{E}[Y|\boldsymbol{do}(T=c, \mathbf{M}=\mathbf{m})]$$

**Natural Direct Effect (NDE):**

$$\text{NDE}_{T\to Y}(t, c) = \mathbb{E}[Y|\boldsymbol{do}(T=t), \boldsymbol{do}(\mathbf{M}=\mathbf{M}_c)] - \mathbb{E}[Y|\boldsymbol{do}(T=c), \boldsymbol{do}(\mathbf{M}=\mathbf{M}_c)]$$

## A.2 Counterfactual Quantities ($\mathcal{L}_3$)

A counterfactual query such as $P(Y_{\boldsymbol{do}(T=t)}|\mathbf{V}_F = \mathbf{v}_F)$ is computed by abduction (conditioning on factual data), action (intervening), and prediction (computing the outcome) [35].

**Ctf-TE / Ctf-DE / Ctf-IE:**

$$\text{Ctf-TE}_{T\to Y}(y, t, c, \mathbf{v}_F) = P(y_{\boldsymbol{do}(T=t)}|\mathbf{V}_F = \mathbf{v}_F) - P(y_{\boldsymbol{do}(T=c)}|\mathbf{V}_F = \mathbf{v}_F)$$

$$\text{Ctf-DE}_{T\to Y}(y, t, c, \mathbf{v}_F) = P(y_{\boldsymbol{do}(T=t),\boldsymbol{do}(\mathbf{M}=\mathbf{M}_c)}|\mathbf{V}_F = \mathbf{v}_F) - P(y_{\boldsymbol{do}(T=c)}|\mathbf{V}_F = \mathbf{v}_F)$$

$$\text{Ctf-IE}_{T\to Y}(y, t, c, \mathbf{v}_F) = P(y_{\boldsymbol{do}(T=c),\boldsymbol{do}(\mathbf{M}=\mathbf{M}_t)}|\mathbf{V}_F = \mathbf{v}_F) - P(y_{\boldsymbol{do}(T=c)}|\mathbf{V}_F = \mathbf{v}_F)$$

# B Space of Interest

Each Space of Interest is defined by a set of parameters that control the *SCM space*, the causal queries of interest (*Query space*), and the dataset used for estimation (*Data space*). Table 3 provides an overview of all configurable parameters in a Space of Interest instance, along with their default values. Some parameters are only relevant under specific conditions—for instance, kernel parameters are used only with continuous variables (e.g., when evaluating conditional expectations), function

sampling strategies apply exclusively to discrete mechanisms, noise regions apply only for discrete SCMs, and noise mode is ignored for tabular mechanisms (noise is already embedded in the table). Note that one can use symbolic expressions involving `N` (the number of nodes) and `V` (the cardinality of a variable) to define parameters that depend on sampled values. For example, the expected number of edges can be set as `0.5 * N`, or the number of noise regions in a discrete SCM can be set to `V`.

| Category | Parameter | Default Value |
|---|---|---|
| | Number of endogenous variables | [5, 15] |
| | Variable dimensionality | [1, 1] |
| | Expected number of edges (required) | — |
| SCM structure | Proportion of hidden variables | 0.0 |
| | Markovian boolean flag | True |
| | Semi-Markovian boolean flag | False |
| | Predefined causal graph | — |
| | Mechanism family (e.g., Linear, NN, Tabular) | Linear |
| | Mechanism arguments (used to define custom NN/tabular mechanisms) | — |
| Mechanisms | Endogenous variable cardinality (for discrete variables only) | 2 |
| | Variable type | Continuous |
| | Discrete function sampling | Sample Rejection |
| | Noise mode | Additive |
| | Noise distribution | Uniform |
| Noise | Noise distribution arguments | [-1, 1] |
| | Number of noise regions (controls stochasticity) | N |
| | Number of queries per sample | 1 |
| Query | Query type | ATE |
| | Specific query (overrides random query sampling) | — |
| | Kernel type | Gaussian |
| Kernel | Kernel bandwidth | 0.1 |
| | Custom kernel function | — |
| Data | Number of samples in the set of observed data | 1000 |

Table 3: Parameters defining a Space of Interest instance and their default values. The double lines in the table conceptually separate the SCM space, Query space, and Data space.

# C Causal Graph Sampling

We first generate a random Directed Acyclic Graph (DAG) that specifies causal relations between variables. This structure is then extended by designating a subset of variables as hidden/unobserved, enabling the creation of both Markovian and semi-Markovian SCMs depending on the *SoI* spec. We separate these two steps in separate algorithms for clarity (algorithm 3 uses algorithm 2).

**Algorithm 2** Generate a Random DAG with Expected Degree

**Inputs:** number of nodes $N$, expected degree $d$

$V \leftarrow \{1, \ldots, N\}$
$E \leftarrow \{\}$
$p_{edge} \leftarrow \frac{2d}{N-1}$
**for** $i \in [1, N]$ **do**
    $N_{\boldsymbol{PA}(i)} \sim B(i-1, p_{edge})$
    $\boldsymbol{PA}(i) \leftarrow N_{\boldsymbol{PA}(i)}$ nodes sampled without replacement from $V$
    $E \leftarrow E \cup \{j \to i \mid j \in \boldsymbol{PA}(i)\}$
**end for**

**Output:** $\mathcal{G} = \{V, E\}$

**Algorithm 3** Generate a DAG with Observed and Hidden Variables

**Inputs:** number of nodes $N$, expected degree $d$, proportion of hidden variables $p_h$

$\mathcal{G} = (V, E) \leftarrow DAG_sampling(N, d)$ *(see algorithm 2)*
$N_h \sim B(N, p_h)$
$V_h \leftarrow N_h$ nodes sampled without replacement from $V$
$V_o \leftarrow V \backslash V_h$

**Output:** $\mathcal{G} = \{V = V_o V_h, E\}$

Because some variables in the DAG are unobserved, we expose only the observed structure to the user in the form of an Acyclic Directed Mixed Graph. To obtain this, we apply Verma's latent projection algorithm to the causal graph of each sampled regional discrete SCM (see Algorithm 4). If a method requires the true SCM, including the hidden confounders, that can be accessed as well.

**Algorithm 4** Projection Algorithm [47]

**Input:** an Acyclic-Directed Mixed Graph (ADMG) $\mathcal{G} = \{\mathbf{V_O}, \mathbf{V_H}, \mathbf{E}\}$, with $\mathbf{V_O}$ the set of observed variables, $\mathbf{V_H}$ the set of hidden variables and $\mathbf{E}$ the mixed edges

$\mathbf{E}' \leftarrow \{\}$
**for** $A, B \in \mathbf{V_O}$ **do**
  **if** there is a directed path $A \rightarrow \ldots \rightarrow B$ in $\mathcal{G}$ with all intermediate nodes belonging to $\mathbf{V_H}$ **then**
    $\mathbf{E}' \leftarrow \mathbf{E}' \cup \{A \rightarrow B\}$
  **end if**
  **if** there is a collider-free path $A \leftarrow \ldots \rightarrow B$ in $\mathcal{G}$ with all intermediate nodes belonging to $\mathbf{V_H}$ **then**
    $\mathbf{E}' \leftarrow \mathbf{E}' \cup \{A \leftrightarrow B\}$
  **end if**
**end for**
$\mathbf{G}' \leftarrow \{\mathbf{V_O}, E'\}$

**Output:** $\mathbf{G}'$ the latent projection of $\mathbf{G}$ over $\mathbf{V_O}$

# D Sampling Discrete SCMs

## D.1 Regional Discrete SCMs

In this work, we sample discrete Markovian SCMs inspired by [49] and [48] which we refer to as **Regional discrete SCMs** as presented in definition D.1. For a description of how we generate the causal graph, check Appendix C.

**Definition D.1. Regional discrete SCM**

A **regional discrete SCM** is a markovian SCM $\mathcal{M} := \{\mathbf{V}, \mathbf{U}, \boldsymbol{\mathcal{F}}, P(\mathbf{U})\}$ where:

- $\mathbf{V} = \{V_1, ..., V_d\}$ the set of finite discrete endogenous variables is divided into two sets $\mathbf{V}_o$ and $\mathbf{V}_h$ respectively representing the set of observed and hidden variables such that $\mathbf{V} = \mathbf{V}_o \cup \mathbf{V}_h$ and $\mathbf{V}_o \cap \mathbf{V}_h = \emptyset$
- $\mathbf{U} = \{U_1, ..., U_d\}$ the set of mutually independent continuous exogenous variables is such that $\forall i \in [1, d],\ U_{V_i} = U_i$
- $\boldsymbol{\mathcal{F}}$ the structural equations are regional discrete mechanisms as defined in definition D.2

The class of regional discrete SCMs is denoted $\mathbb{M}_{\texttt{RD-SCM}}$.

**Definition D.2.** **Regional discrete mechanism**

Given $\mathbf{I}_V = \{I_V^r\}_{r\in[1,R]}$ a partition of $R$ parts of $\Omega_{U_V}$ and $m_V = \{m_V^r : \Omega_{\boldsymbol{PA}(V)} \mapsto \Omega_V\}_{r\in[1,R]}$ a set of $R$ distinct mappings from $\Omega_{\boldsymbol{PA}(V)}$ to $\Omega_V$, the **regional discrete mechanism** of an endogenous variables $V$ is a function $f_V : \Omega_{\boldsymbol{PA}(V)}, \Omega_{U_V} \mapsto \Omega_V$ such that:

$$f_V(\mathbf{pa}(V),\ u_V) = m_r(\boldsymbol{PA}(V) \mapsto V) \text{ when } u_V \in I_V^r$$

$I_V^r$ and $m_r$ are called the $r^{th}$ noise region and mapping of the regional discrete mechanism $f_V$.

**Remark on $\Omega_{U_V}$ and $R$:** In the definition of a regional discrete mechanism (definition D.2), no constraints are imposed on $\Omega_{U_V}$. However, if $\Omega_{U_V}$ is discrete, then $|\Omega_{U_V}| \geq R$ is required to form a partition of $R$ elements of $\Omega_{U_V}$. Consequently, in order to be able to constitute such a partition for any finite $R$, we decided to consider continuous exogenous variables in the definition of a regional discrete SCM (definition D.1). In addition, since the $m_V^r$ mappings are considered distinct and there are exactly $|\Omega_V|^{|\Omega_{\boldsymbol{PA}(V)}|}$ different mappings from $V$ to $\boldsymbol{PA}(V)$, $R \leq |\Omega_V|^{|\Omega_{\boldsymbol{PA}(V)}|}$ is required.

The fact that regional discrete SCMs contains two types of endogenous variables (i.e., observed and unobserved by the user) enables the representation of complex situations where not all variables are observable. This induces the presence of potential hidden confounders from the user's perspective. As a result, the causal sufficiency assumption is no longer always respected. In our parametric definition of a Space of Interest (SoI), this phenomenon is controlled by the parameter specifying the proportion of unobserved variables among the endogenous variables. Thus, if this parameter is set to 0, the SoI's class of SCMs is included in the class of causally sufficient discrete SCMs.

The complexity of discrete mechanisms can be controlled by the number of noise regions. Indeed, as the number of noise regions increases, so does the complexity of the causal mechanism, in the sense that it becomes a mixture of a larger number of mappings. The distribution of a variable given its parents is, hence, more stochastic. As a result, the user-defined class of regional discrete SCMs can be very broad. and therefore more oversimplified. This provides an additional degree of complexity to make our synthetic causal datasets less trivial.

## D.2 Discrete Mechanism Sampling strategies

We use *regional discrete mechanisms* (definition D.2), which define tabular mappings from parent variables to a target variable, conditioned on regions of the exogenous noise space. Each region induces a distinct mapping, enabling both stochasticity and high functional expressivity.

To generate these mechanisms, we support three sampling strategies described below. All methods define a partition of the exogenous noise domain $\Omega_U$ into $R$ regions, and assign a parent-to-child mapping to each region. Let $C$ be the cardinality of the variables, and $\Omega_{\mathrm{Pa}(V)}$ the space of parent configurations for variable $V$.

**Controlling complexity.** The number of possible mappings from parent configurations to output values grows as $|\Omega_V|^{|\Omega_{\mathrm{Pa}(V)}|}$. To keep simulations tractable, users can control the number of noise regions $R$. When $R$ is small, sampling provides diverse but lightweight mechanisms. When $R$ approaches the total number of mappings, full enumeration becomes feasible but computationally expensive.

We now describe the three supported sampling strategies.

### Exhaustive partition

This strategy enumerates all possible mappings from parent configurations to output values and assigns each one to a distinct noise region ($R = |\Omega_V|^{|\Omega_{\mathrm{Pa}(V)}|}$), ensuring complete coverage of the function space. This method guarantees maximal functional diversity across regions and can serve as a stress test for generalization under highly non-linear mechanisms. This is the only strategy where the number of noise regions is not decided by the user but rather set to the maximum.

### Sample rejection

This strategy samples parent-to-output mappings uniformly at random, rejecting duplicates to ensure that each region corresponds to a distinct function. As mappings are sampled with replacement, rejection may require several attempts when $R$ approaches the number of possible mappings.

We provide below a pseudocode version of this strategy. Note that lines 10–12 correspond to the rejection logic.

**Algorithm 5** Generating regional discrete mechanisms with sample rejection

**Inputs:** set of endogenous variables $\mathbf{V}$ of cardinality $C$, causal graph $\mathcal{G}$, $\Omega_U$ domain of exogenous variables, number of noise regions $R$

1: $\boldsymbol{\mathcal{F}} \leftarrow \{\}$
2: **for** $V \in \mathbf{V}$ **do**
3: $\quad \Omega_V \leftarrow \{1, \ldots, C\}$
4: $\quad \Omega_{\boldsymbol{PA}_{\mathcal{G}}(V)} \leftarrow \{1, \ldots, C\}^{|\boldsymbol{PA}_{\mathcal{G}}(V)|}$
5: $\quad R \leftarrow \min(R, |\Omega_V|^{|\Omega_{\boldsymbol{PA}(V)}|})$
6: $\quad l_{\min} \leftarrow \inf(\Omega_U)$
7: $\quad l_{\max} \leftarrow \sup(\Omega_U)$
8: $\quad \mathbf{L} = \{l_i \sim \mathcal{U}[l_{\min}, l_{\max}] \mid i \in [1, R-1]\} \cup \{l_{\min}, l_{\max}\}$
9: $\quad$ Sort $\mathbf{L}$ in ascending order
10: $\quad f_V \leftarrow \{\}$
11: $\quad m_V \leftarrow \{\}$
12: $\quad$ **for** $r \in [1, R]$ **do**
13: $\quad\quad I_V^r \leftarrow [\mathbf{L}_r, \mathbf{L}_{r+1}[$ with $\mathbf{L}_r$ the $r^{th}$ element of $\mathbf{L}$
14: $\quad\quad m_V^r \leftarrow \{\}$
15: $\quad\quad$ **while** $m_V^r = \{\}$ or $m_V^r \in m_V$ **do**
16: $\quad\quad\quad m_V^r \leftarrow |\Omega_{\boldsymbol{PA}(V)}|$ elements sampled with replacement from $\Omega_V$
17: $\quad\quad$ **end while**
18: $\quad\quad m_V \leftarrow m_V \cup m_V^r$
19: $\quad\quad f_V \leftarrow f_V \cup \{m_V^r; I_V^r\}$
20: $\quad$ **end for**
21: $\quad \boldsymbol{\mathcal{F}} \leftarrow \boldsymbol{\mathcal{F}} \cup f_V$
22: **end for**

**Output:** $\boldsymbol{\mathcal{F}}$

### Unbiased random assignment

In this strategy, each noise region is assigned a mapping sampled independently and without enforcing uniqueness. As a result, multiple regions may correspond to the same function from parent configurations to outputs.

For example, suppose a variable has one binary parent taking values in $\{0, 1\}$, and the output variable takes values in $\{0, 1, 2\}$. One randomly sampled mapping might assign output $0$ to parent value $0$, and output $2$ to parent value $1$. Since mappings are sampled independently for each region, this same function ($0 \to 0, 1 \to 2$) may appear in multiple regions by chance.

This approach reflects scenarios where mechanisms are drawn independently from a distribution over functions, without enforcing any requirements on uniqueness or coverage. As a result, the effective variability in the entire system may be lower compared to other strategies, but the sampling is a lot more computationally efficient.

# E Query Sampling and Estimation

In this work, we consider the following types of queries: ATE, Conditional Average Treatment Effect (CATE) and Counterfactual Total Effect (Ctf-TE). Their definitions can be found in Appendix A. All the queries can be defined for sets of covariates and factuals belonging to the set of endogenous variables. In other words, we do not implement multi-interventions, but we consider conditioning and observing factuals on several variables. Finally, the values taken by these variables (e.g., treatment and control values for ATE) must belong to their definition domain. The only parameter that controls the queries class is the type of queries chosen by the user (i.e., ATE, CATE and Ctf-TE). Thus, the class of considered queries can be defined as follows:

$$\mathcal{Q}_{\text{ATE}} = \{\text{ATE}_{T \rightarrow Y}(t, c) \mid T, Y \subseteq \mathbf{V} \text{ and } t, c \in \Omega_T\}$$

$$\mathcal{Q}_{\text{CATE}} = \{\text{CATE}_{T \rightarrow Y|\mathbf{X}}(t, c, \mathbf{x}) \mid T, Y \subseteq \mathbf{V},\ \mathbf{X} \subseteq \mathbf{V} \backslash \{T, Y\} \text{ and } t, c \in \Omega_T,\ \mathbf{x} \in \Omega_{\mathbf{X}}\}$$

$$\mathcal{Q}_{\text{Ctf-TE}} = \{\text{Ctf-TE}_{T \rightarrow Y}(y, t, c, \boldsymbol{v}_F) \mid T, Y, \boldsymbol{V}_F \subseteq \mathbf{V} \text{ and } t, c \in \Omega_T,\ y \in \Omega_Y,\ \boldsymbol{v}_F \in \Omega_{\boldsymbol{V}_F}\}$$

Formally speaking, we have not integrated the causal graph as a causal query but rather as a hypothesis or prior knowledge. Indeed, except for causal discovery tasks, the causal graph is most often assumed to be known (or at least some information derived from the graph, such as the constitution of a valid adjustment set, or a valid causal ordering). Nevertheless, one can use our random causal dataset generator to evaluate causal discovery or causal representation learning methods. To do so, one just needs to retrieve the causal graph from the causal dataset directly instead of using a query.

Finally, a user can also implement a specific query and use it to generate synthetic causal datasets. To do this, the user has to use the Query class in our code base.

In the following algorithms, given a dataset $D$, a variable $X$ and a realization $x$ of $X$, we use the notation $D_{|X}$ (resp. $D_{|X=x}$) to represent the dataset $D$ restricted to the variable $X$ (resp. restricted to the samples whose $X$ realization equals $x$). In addition, $B(n, p)$ denotes the Binomial law of parameters $n$ and $p$.

## E.1 Query Sampling

The following algorithms detail the procedures for sampling ATE, CATE, and CTF-TE queries.

**Algorithm 6** Generating sets of observed data

**Inputs:** causal graph $\mathcal{G}$, causal mechanisms $\boldsymbol{\mathcal{F}}$, distribution of the exogenous variables $P(\mathbf{U})$, dataset size $N$

1: $D \leftarrow \{\}$
2: $D_o \leftarrow \{\}$
3: $\{\mathbf{u}_1, \dots, \mathbf{u}_N\} \sim P(\mathbf{U})$
4: **for** $V \in \mathbf{V}$ following a causal order given by $\mathcal{G}$ **do**
5: $\quad \{\mathbf{pa}(V)_1, \dots, \mathbf{pa}(V)_N\} \leftarrow D_{|\boldsymbol{PA}(V)}$
6: $\quad \{u_{V_1}, \dots, u_{V_N}\} \leftarrow D_{|\mathbf{U}_V}$
7: $\quad \{v_1, \dots, v_N\} \leftarrow f_V(\{\mathbf{pa}(V)_1, \dots, \mathbf{pa}(V)_N\}, \{u_{V_1}, \dots, u_{V_N}\})$
8: $\quad D \leftarrow D \cup \{v_1, \dots, v_N\}$
9: $\quad$ **if** $V \in \mathbf{V}_o$ **then**
10: $\quad\quad D_o \leftarrow D_o \cup \{v_1, \dots, v_N\}$
11: $\quad$ **end if**
12: **end for**

**Output:** $D_o$

**Algorithm 7** Generating ATE queries

**Inputs:** set of observable endogenous variables $\mathbf{V}_o$, training set $D$

$T \leftarrow$ one variable randomly sampled from $\mathbf{V}_o$
$Y \leftarrow$ one variable randomly sampled from $\mathbf{V}_o$
$t \leftarrow$ one realization of $T$ randomly sampled from $D_{|T}$
$c \leftarrow$ one realization of $T$ randomly sampled from $D_{|T}$

**Output:** $Q_{ATE} = \{T, Y, t, c\}$

**Algorithm 8** Generating CATE queries

**Inputs:** set of observable endogenous variables $\mathbf{V}_o$, training set $D$

$T \leftarrow$ one variable randomly sampled from $\mathbf{V}_o$
$Y \leftarrow$ one variable randomly sampled from $\mathbf{V}_o$
$d_{\mathbf{X}} \leftarrow$ an integer randomly sampled from $[1, \ldots, |\mathbf{V}_o| - 2]$
$\mathbf{X} \leftarrow d_{\mathbf{X}}$ variables randomly sampled from $\mathbf{V}_o \backslash \{T, Y\}$
$t \leftarrow$ one realization of $T$ randomly sampled from $D_{|T}$
$c \leftarrow$ one realization of $T$ randomly sampled from $D_{|T}$
$\mathbf{x} \leftarrow$ one realization of $\mathbf{X}$ randomly sampled from $D_{|\mathbf{X}}$

**Output:** $Q_{CATE} = \{T, Y, \mathbf{X}, t, c, \mathbf{x}\}$

**Algorithm 9** Generating Ctf-TE queries

**Inputs:** set of observable endogenous variables $\mathbf{V}_o$, training set $D$

$T \leftarrow$ one variable randomly sampled from $\mathbf{V}_o$
$Y \leftarrow$ one variable randomly sampled from $\mathbf{V}_o$
$d_{\mathbf{V}_F} \leftarrow$ an integer randomly samples from $[1, \ldots, |\mathbf{V}_o|]$
$\mathbf{V}_F \leftarrow d_{\mathbf{V}_F}$ variables randomly sampled from $\mathbf{V}_o$
$t \leftarrow$ one realization of $T$ randomly sampled from $D_{|T}$
$c \leftarrow$ one realization of $T$ randomly sampled from $D_{|T}$
$\mathbf{v}_F \leftarrow$ one realization of $\mathbf{V}_F$ randomly sampled from $D_{|\mathbf{V}_F}$

**Output:** $Q_{CTF-TE} = \{T, Y, \mathbf{V}_F, t, c, \mathbf{v}_F\}$

### E.2 SCM-Based Query Estimation

Each query is evaluated by modifying the SCM, sampling the exogenous variables, and computing expectations over the outcomes. In practice, we simulate interventions and counterfactuals by directly manipulating structural equations and conditioning on sampled variables. Our implementation supports efficient batch estimation using the same random seeds for reproducibility.

Counterfactual queries are estimated using the standard three-step procedure [35]:

1. **Abduction:** Condition on the factual realization to compute $P(\mathbf{U}|\mathbf{V}_F = \mathbf{v}_F)$
2. **Action:** Modify the SCM with the desired intervention
3. **Prediction:** Compute the outcome using the intervened model and posterior samples

The following algorithms detail the procedures for estimating ATE, CATE, and CTF-TE queries.

**Algorithm 10** Estimating ATE queries

**Inputs:** ATE query to estimate $Q = \{T, Y, t, c\}$, causal graph $\mathcal{G}$, causal mechanisms $\boldsymbol{\mathcal{F}}$, distribution of the exogenous variables $P(\mathbf{U})$, number of samples to draw for estimation $N$

$\{\mathbf{u}_1, \ldots, \mathbf{u}_N\} \sim P(\mathbf{U})$
$D_t \leftarrow \{\mathbf{u}_1, \ldots, \mathbf{u}_N\}$
**for** $V \in \mathbf{V}$ following a causal order given by $\mathcal{G}$ **do**
  **if** $V = T$ **then**
    $\{v_1, \ldots, v_N\} \leftarrow \{t, \ldots, t\}$
  **else**
    $\{\mathbf{pa}(V)_1, \ldots, \mathbf{pa}(V)_N\} \leftarrow D_{t|\boldsymbol{PA}(V)}$
    $\{u_{V_1}, \ldots, u_{V_N}\} \leftarrow D_{t|\mathbf{U}_V}$
    $\{v_1, \ldots, v_N\} \leftarrow f_V(\{\mathbf{pa}(V)_1, \ldots, \mathbf{pa}(V)_N\}, \{u_{V_1}, \ldots, u_{V_N}\})$
  **end if**
  $D_t \leftarrow D_t \cup \{v_1, \ldots, v_N\}$
**end for**
$D_c \leftarrow \{\mathbf{u}_1, \ldots, \mathbf{u}_N\}$
**for** $V \in \mathbf{V}$ following a causal order given by $\mathcal{G}$ **do**
  **if** $V = T$ **then**
    $\{v_1, \ldots, v_N\} \leftarrow \{c, \ldots, c\}$
  **else**
    $\{\mathbf{pa}(V)_1, \ldots, \mathbf{pa}(V)_N\} \leftarrow D_{c|\boldsymbol{PA}(V)}$
    $\{u_{V_1}, \ldots, u_{V_N}\} \leftarrow D_{c|\mathbf{U}_V}$
    $\{v_1, \ldots, v_N\} \leftarrow f_V(\{\mathbf{pa}(V)_1, \ldots, \mathbf{pa}(V)_N\}, \{u_{V_1}, \ldots, u_{V_N}\})$
  **end if**
  $D_c \leftarrow D_c \cup \{v_1, \ldots, v_N\}$
**end for**
$Q^\star \leftarrow \text{avg}(D_{t|Y}) - \text{avg}(D_{c|Y})$

**Output:** $Q^\star$

**Algorithm 11** Estimating CATE queries

**Inputs:** CATE query to estimate $Q = \{T, Y, \mathbf{X}, t, c, \mathbf{x}\}$, causal graph $\mathcal{G}$, causal mechanisms $\boldsymbol{\mathcal{F}}$, distribution of the exogenous variables $P(\mathbf{U})$, number of samples to draw for estimation $N$

$\{\mathbf{u}_1, \ldots, \mathbf{u}_N\} \sim P(\mathbf{U})$
$D_t \leftarrow \{\mathbf{u}_1, \ldots, \mathbf{u}_N\}$
**for** $V \in \mathbf{V}$ following a causal order given by $\mathcal{G}$ **do**
  **if** $V = T$ **then**
    $\{v_1, \ldots, v_N\} \leftarrow \{t, \ldots, t\}$
  **else**
    $\{\mathbf{pa}(V)_1, \ldots, \mathbf{pa}(V)_N\} \leftarrow D_{t|\boldsymbol{PA}(V)}$
    $\{u_{V_1}, \ldots, u_{V_N}\} \leftarrow D_{t|\mathbf{U}_V}$
    $\{v_1, \ldots, v_N\} \leftarrow f_V(\{\mathbf{pa}(V)_1, \ldots, \mathbf{pa}(V)_N\}, \{u_{V_1}, \ldots, u_{V_N}\})$
  **end if**
  $D_t \leftarrow D_t \cup \{v_1, \ldots, v_N\}$
**end for**
$D_c \leftarrow \{\mathbf{u}_1, \ldots, \mathbf{u}_N\}$
**for** $V \in \mathbf{V}$ following a causal order given by $\mathcal{G}$ **do**
  **if** $V = T$ **then**
    $\{v_1, \ldots, v_N\} \leftarrow \{c, \ldots, c\}$
  **else**
    $\{\mathbf{pa}(V)_1, \ldots, \mathbf{pa}(V)_N\} \leftarrow D_{c|\boldsymbol{PA}(V)}$
    $\{u_{V_1}, \ldots, u_{V_N}\} \leftarrow D_{c|\mathbf{U}_V}$
    $\{v_1, \ldots, v_N\} \leftarrow f_V(\{\mathbf{pa}(V)_1, \ldots, \mathbf{pa}(V)_N\}, \{u_{V_1}, \ldots, u_{V_N}\})$
  **end if**
  $D_c \leftarrow D_c \cup \{v_1, \ldots, v_N\}$
**end for**
$D_t \leftarrow D_{t|\mathbf{X}=\mathbf{x}}$
$D_c \leftarrow D_{c|\mathbf{X}=\mathbf{x}}$
$Q^\star \leftarrow \text{avg}(D_{t|Y}) - \text{avg}(D_{c|Y})$

**Output:** $Q^\star$

**Algorithm 12** Estimating Ctf-TE queries

**Inputs:** Ctf-TE query to estimate $Q = \{T, Y, \mathbf{V}_F, t, c, \mathbf{v}_F\}$, causal graph $\mathcal{G}$, causal mechanisms $\boldsymbol{\mathcal{F}}$, distribution of the exogenous variables $P(\mathbf{U})$, number of samples to draw for estimation $N$

$\{\mathbf{u}_1, \ldots, \mathbf{u}_N\} \sim P(\mathbf{U})$
$D_{\mathbf{U}_{\mathbf{v}_F}} \leftarrow \{\mathbf{u}_1, \ldots, \mathbf{u}_N\}$
**for** $V \in \mathbf{V}$ following a causal order given by $\mathcal{G}$ **do**
  $\{\mathbf{pa}(V)_1, \ldots, \mathbf{pa}(V)_N\} \leftarrow D_{\mathbf{U}_{\mathbf{v}_F}|\boldsymbol{PA}(V)}$
  $\{u_{V_1}, \ldots, u_{V_N}\} \leftarrow D_{\mathbf{U}_{\mathbf{v}_F}|\mathbf{U}_V}$
  $\{v_1, \ldots, v_N\} \leftarrow f_V(\{\mathbf{pa}(V)_1, \ldots, \mathbf{pa}(V)_N\}, \{u_{V_1}, \ldots, u_{V_N}\})$
  $D_{\mathbf{U}_{\mathbf{v}_F}} \leftarrow D_{\mathbf{U}_{\mathbf{v}_F}} \cup \{v_1, \ldots, v_N\}$
**end for**
$D_{\mathbf{U}_{\mathbf{v}_F}} \leftarrow D_{\mathbf{U}_{\mathbf{v}_F}|\mathbf{V}_F=\mathbf{v}_F}$
$M \leftarrow |D_{\mathbf{U}_{\mathbf{v}_F}}|$
$\{\mathbf{u}_1, \ldots, \mathbf{u}_M\} \leftarrow D_{\mathbf{U}_{\mathbf{v}_F}|\mathbf{U}}$
$D_t \leftarrow \{\mathbf{u}_1, \ldots, \mathbf{u}_M\}$
**for** $V \in \mathbf{V}$ following a causal order given by $\mathcal{G}$ **do**
  **if** $V = T$ **then**
    $\{v_1, \ldots, v_N\} \leftarrow \{t, \ldots, t\}$
  **else**
    $\{\mathbf{pa}(V)_1, \ldots, \mathbf{pa}(V)_N\} \leftarrow D_{t|\boldsymbol{PA}(V)}$
    $\{u_{V_1}, \ldots, u_{V_N}\} \leftarrow D_{t|\mathbf{U}_V}$
    $\{v_1, \ldots, v_N\} \leftarrow f_V(\{\mathbf{pa}(V)_1, \ldots, \mathbf{pa}(V)_N\}, \{u_{V_1}, \ldots, u_{V_N}\})$
  **end if**
  $D_t \leftarrow D_t \cup \{v_1, \ldots, v_N\}$
**end for**
$D_c \leftarrow \{\mathbf{u}_1, \ldots, \mathbf{u}_M\}$
**for** $V \in \mathbf{V}$ following a causal order given by $\mathcal{G}$ **do**
  **if** $V = T$ **then**
    $\{v_1, \ldots, v_N\} \leftarrow \{c, \ldots, c\}$
  **else**
    $\{\mathbf{pa}(V)_1, \ldots, \mathbf{pa}(V)_N\} \leftarrow D_{c|\boldsymbol{PA}(V)}$
    $\{u_{V_1}, \ldots, u_{V_N}\} \leftarrow D_{c|\mathbf{U}_V}$
    $\{v_1, \ldots, v_N\} \leftarrow f_V(\{\mathbf{pa}(V)_1, \ldots, \mathbf{pa}(V)_N\}, \{u_{V_1}, \ldots, u_{V_N}\})$
  **end if**
  $D_c \leftarrow D_c \cup \{v_1, \ldots, v_N\}$
**end for**
$Q^\star \leftarrow \text{avg}(D_{t|Y}) - \text{avg}(D_{c|Y})$

**Output:** $Q^\star$

# F Assumptions analysis module's metrics

In order to analyze the characteristics of the sampled SCMs we implemented the following metrics. Let us imagine we sampled a regional discrete SCM $\mathcal{M} := \{\mathbf{V}, \mathbf{U}, \boldsymbol{\mathcal{F}}, P(\mathbf{U})\}$ with $\mathbf{V} = (\mathbf{V}_o, \mathbf{V}_h)$ and whose causal graph is denoted $\mathcal{G}$. The projection of $\mathcal{G}$ over the observable variables $\mathbf{V}_o$ is denoted $\mathcal{G}_{\mathbf{V}_o}$.

**Analysis of the causal graph $\mathcal{G}$:**

- Average in-degree: $\bar{d}_{in} = \frac{1}{|\mathbf{V}|} \sum_{V \in \mathbf{V}} |\boldsymbol{PA}(V)|$
- Variance of in-degree: $\text{var}(d_{in}) = \frac{1}{|\mathbf{V}|} \sum_{V \in \mathbf{V}} (|\boldsymbol{PA}(V)| - \bar{d}_{in})^2$
- Average number of ancestors: $\overline{|An(V)|} = \frac{1}{|\mathbf{V}|} \sum_{V \in \mathbf{V}} |An(V)|$ where $An(V)$ denotes the set of ancestors of $V$
- Variance of number of ancestors: $\text{var}(|An(V)|) = \frac{1}{|\mathbf{V}|} \sum_{V \in \mathbf{V}} (|An(V)| - \overline{|An(V)|})^2$
- Average number of descendants: $\overline{|De(V)|} = \frac{1}{|\mathbf{V}|} \sum_{V \in \mathbf{V}} |De(V)|$ where $De(V)$ denotes the set of descendants of $V$
- Variance of number of descendants: $\text{var}(|De(V)|) = \frac{1}{|\mathbf{V}|} \sum_{V \in \mathbf{V}} (|De(V)| - \overline{|De(V)|})^2$
- Average length of causal paths: $\overline{L} = \frac{1}{|\mathbf{p}_\mathcal{G}|} \sum_{p \in \mathbf{p}_\mathcal{G}} |p|$ where $\mathbf{p}_\mathcal{G}$ denotes the set of directed paths in $\mathcal{G}$
- Variance length of causal paths: $\text{var}(L) = \frac{1}{|\mathbf{p}_\mathcal{G}|} \sum_{p \in \mathbf{p}_\mathcal{G}} (|p| - \overline{L})^2$
- Maximum length of causal paths: $L_{\max} = \max_{p \in \mathbf{p}_\mathcal{G}} |p|$

**Analysis of the projected causal graph $\mathcal{G}_{\mathbf{V}_o}$:**

- Average number of siblings[3]: $\overline{|Si(V)|} = \frac{1}{|\mathbf{V}_o|} \sum_{V \in \mathbf{V}_o} |Si(V)|$ where $Si(V)$ denotes the set of siblings of $V$
- Variance of number of siblings: $\text{var}(|Si(V)|) = \frac{1}{|\mathbf{V}_o|} \sum_{V \in \mathbf{V}_o} (|Si(V)| - \overline{|Si(V)|})^2$
- Number of maximal confounded components (c-comps)[4]: $|\mathbf{C}|$ where $\mathbf{C}$ denotes the set of maximal c-comps in $\mathcal{G}_{\mathbf{V}_o}$
- Average size of maximal c-comps: $\overline{|\mathbf{C}|} = \frac{1}{|\mathbf{C}|} \sum_{C \in \mathbf{C}} |C|$
- Variance of the size of maximal c-comps: $\text{var}(|\mathbf{C}|) = \frac{1}{|\mathbf{C}|} \sum_{C \in \mathbf{C}} (|C| - \overline{|\mathbf{C}|})^2$

**Analysis of the observational distribution $P_\mathcal{M}(\mathbf{V_o})$:**

- Minimum probability of the joint distribution: $p_{\mathbf{V}_o, \min} = \min_{\mathbf{v}_o \in \Omega_{\mathbf{V}_o}} P_\mathcal{M}(\mathbf{V}_o = \mathbf{v}_o)$
- Proportion of events with a null probability: $p_0 = \frac{1}{|\Omega_{\mathbf{V}_o}|} \sum_{\mathbf{v}_o \in \Omega_{\mathbf{V}_o}} \mathbf{1}_{P_\mathcal{M}(\mathbf{V_o} = \mathbf{v}_o) = 0}$ where $\mathbf{1}_{_}$ denotes the indicator function
- Minimum probability of the marginal distributions:
$$p_{\min} = \min_{V \in \mathbf{V}_o} \min_{v \in \Omega_V} P_\mathcal{M}(V = v)$$
- Average minimum probability of the marginal distributions:
$$\bar{p}_{\min} = \frac{1}{|\mathbf{V}_o|} \sum_{V \in \mathbf{V}_o} \frac{1}{|\Omega_V|} \min_{v \in \Omega_V} P_\mathcal{M}(V = v)$$

[3] Two variables are considered siblings if they are linked by a bi-directed edge.

[4] We use [46] definition of (maximal) confounded components.

- Variance of the minimum probability of the marginal distributions:

$$\text{var}(p_{\min}) = \frac{1}{|\mathbf{V}_o|} \sum_{V \in \mathbf{V}_o} (\min_{v \in \Omega_V} P_{\mathcal{M}}(V = v) - \bar{p}_{\min})^2$$

- Distance ($L_1$) of the joint distributions to the uniform one:

$$d(P_{\mathcal{M}}; \mathcal{U}) = \sum_{\mathbf{v}_o \in \Omega_{\mathbf{V}_o}} |P_{\mathcal{M}}(\mathbf{V}_o = \mathbf{v}_o) - \frac{1}{|\Omega_{\mathbf{V}_o}|}|$$

- Average distance ($L_1$) of the marginal distributions to the uniform one:

$$\overline{d(P_{\mathcal{M}}; \mathcal{U})} = \frac{1}{|\mathbf{V}_o|} \sum_{V \in \mathbf{V}_o} \sum_{v \in \Omega_V} |P_{\mathcal{M}}(V = v) - \frac{1}{|\Omega_V|}|$$

- Variance of the distance ($L_1$) of the marginal distributions to the uniform one:

$$\text{var}(d(P_{\mathcal{M}}; \mathcal{U})) = \frac{1}{|\mathbf{V}_o|} \sum_{V \in \mathbf{V}_o} \left( \sum_{v \in \Omega_V} |P_{\mathcal{M}}(V = v) - \frac{1}{|\Omega_V|}| - \overline{d(P_{\mathcal{M}}; \mathcal{U})} \right)^2$$

- Entropy of the joint distribution: $\mathrm{H}(P_{\mathcal{M}}(\mathbf{V}))$

All the above-mentioned probabilities are computed from a set of 1M samples drawn from the SCM $\mathcal{M}$.

Let us note that $p_{\min}$ enables the user to check if the strong positivity assumption holds. If $p_{\mathbf{V}_o,\min} > 0$, then strong positivity is respected. In addition, if strong positivity does not hold, $p_{\mathbf{V}_o,\min}$ and $p_0$ indicate the extent to which the assumption is not met – the higher the metrics, the less the hypothesis is respected. On the other hand, $p_{\min}$ indicates whether the weak positivity assumption holds. If $p_{\min} > 0$, then weak positivity is respected. Finally, $d(P_{\mathcal{M}}; \mathcal{U})$, $\overline{d(P_{\mathcal{M}}; \mathcal{U})}$ and $\text{var}(d(P_{\mathcal{M}}; \mathcal{U}))$ enables the user to assess to which extent the observational distribution is imbalanced.

**Analysis of the causal mechanisms $\mathcal{F}$:**

- Average Pearson's correlation between the parent-child pairs[5]:

$$\bar{\rho}_P = \frac{1}{|\mathbf{V}|} \sum_{V \in \mathbf{V}} \frac{1}{|\boldsymbol{PA}(V) \cup U_V|} \sum_{V_j \in \boldsymbol{PA}(V) \cup U_V} \rho_P(V, V_j)$$

- Variance of Pearson's correlation between the parent-child pairs:

$$\text{var}(\rho_P) = \frac{1}{|\mathbf{V}|} \sum_{V \in \mathbf{V}} \frac{1}{|\boldsymbol{PA}(V) \cup U_V|} \sum_{V_j \in \boldsymbol{PA}(V) \cup U_V} (\rho_P(V, V_j) - \bar{\rho}_P)$$

- Average Spearman's correlation between the parent-child pairs[3]

$$\bar{\rho}_S = \frac{1}{|\mathbf{V}|} \sum_{V \in \mathbf{V}} \frac{1}{|\boldsymbol{PA}(V) \cup U_V|} \sum_{V_j \in \boldsymbol{PA}(V) \cup U_V} \rho_S(V, V_j)$$

- Variance of Spearman's correlation between the parent-child pairs:

$$\text{var}(\rho_S) = \frac{1}{|\mathbf{V}|} \sum_{V \in \mathbf{V}} \frac{1}{|\boldsymbol{PA}(V) \cup U_V|} \sum_{V_j \in \boldsymbol{PA}(V) \cup U_V} (\rho_S(V, V_j) - \bar{\rho}_S)$$

- Average conditional entropy of a variable given its parents:

$$\overline{\mathrm{H}} = \frac{1}{|\mathbf{V}|} \sum_{V \in \mathbf{V}} \mathrm{H}(V | \boldsymbol{PA}(V))$$

[5] $\rho_P$ and $\rho_S$ respectively denote the Pearson's and Spearman's correlation

- Variance of conditional entropy of a variable given its parents:

$$\text{var}(\text{H}) = \frac{1}{|\mathbf{V}|} \sum_{V \in \mathbf{V}} (\text{H}(V|\boldsymbol{PA}(V)) - \overline{\text{H}})^2$$

In order to be able to use person correlations, spearman correlations, and conditional entropy as indicators of degrees of linearity, monotonicity, and stochasticity of causal mechanisms, we do not derive these quantities from samples drawn from the entailed distribution. Instead, for each variable, we create a dataset resulting from the application of its causal mechanism to the cartesian product of the values taken by its endogenous and exogenous parents. In other words, we analyze the mechanisms' images of their input space. This allows us to analyze each mechanism independently of the others.

Thus, $\bar{\rho}_P$ and $\text{var}(\rho_P)$ can be interpreted as the average degree of linearity of causal mechanisms and their variance. Furthermore, $\bar{\rho}_S$ and $\text{var}(\rho_S)$ can be interpreted as the average degree of monotonicity of causal mechanisms and their variance. Finally, $\overline{\text{H}}$ and $\text{var}(\text{H})$ can be interpreted as the average level of stochasticity of causal mechanisms and its variance.

# G Analysis of the empirical distribution of the generated SCMs

As we do not provide the user with an expression of the distribution of the sampled regional discrete SCMs, we need to investigate if some SCMs classes are over/underrepresented. This analysis is important to identify the potential biases our random causal dataset generator might create to take them into account when using it to evaluate any Causal machine learning (Causal ML) method. Indeed, as our goal is to provide a tool for rigorous empirical evaluation of causal methods, we need to be transparent on the limitations of our generator such that researchers and practitioners can interpret the results of their methods with full knowledge of the potential biases coming from the generator.

## G.1 Experiment

To visualize the distribution of the SCMs generated, we analyze the distribution of the metrics of the assumption analysis module characterizing the SCMs. For each SCM sampled, all the implemented metrics (see Appendix F) are computed.

The studied SCMs are sampled from the SoIs defined by the cartesian product of the following parameters:

- **Number of endogenous variables**: $[3, 4, 5]$
- **Expected edge probability**: $[0.2, 0.4, 0.6, 0.8]$
- **Proportion of unobserved endogenous variables**: $[0, 0.1, 0.2, 0.3]$
- **Number of noise regions**: $[2, 5, 10, 20, 50]$
- **Cardinality of endogenous variables**: $[2, 3, 4, 7]$
- **Distribution of exogenous variables**: set to $\mathcal{U}[0, 1]$

For each SoI 10 SCMs are sampled, making a total of 9600 SCMs studied. Let us mention that we sample more SCMs than for verification (Section 6.1 for two reasons. First, it enables us to have a better approximation of the SCMs distribution. Second, the computation of all the assumptions and characteristics metrics is, in fact, less computationally expensive than computing all the independence tests.

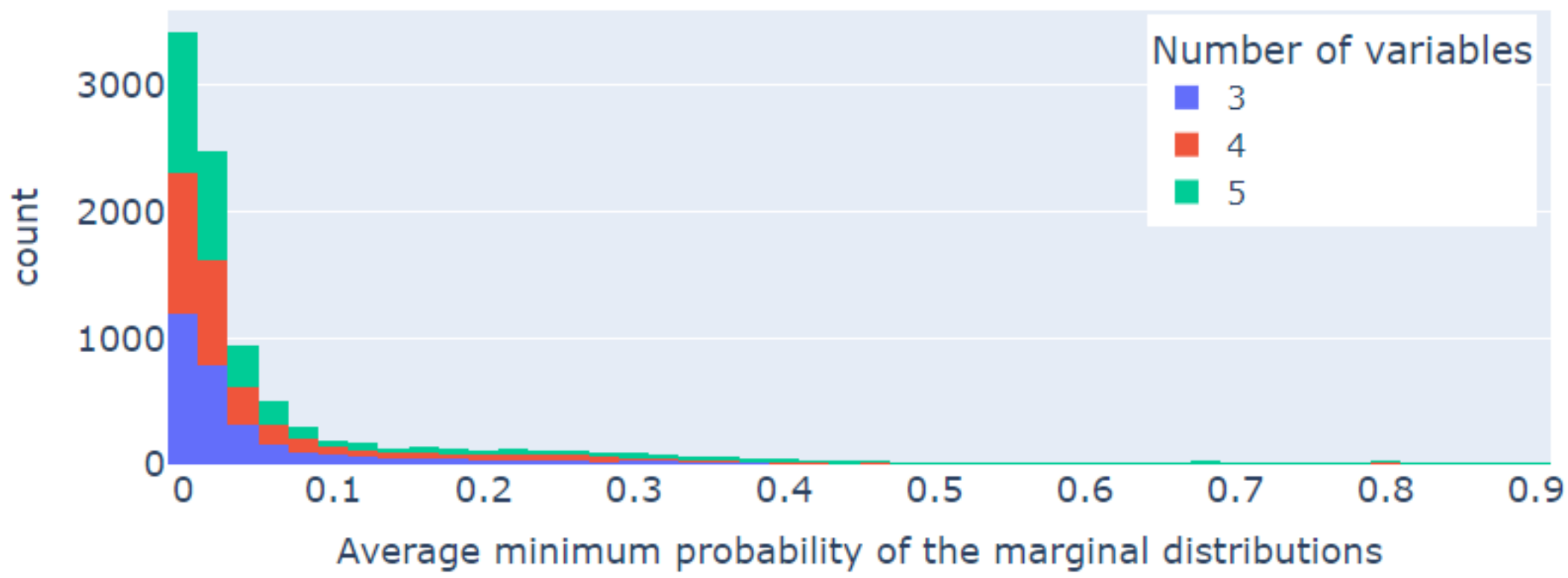

(a) Pearson

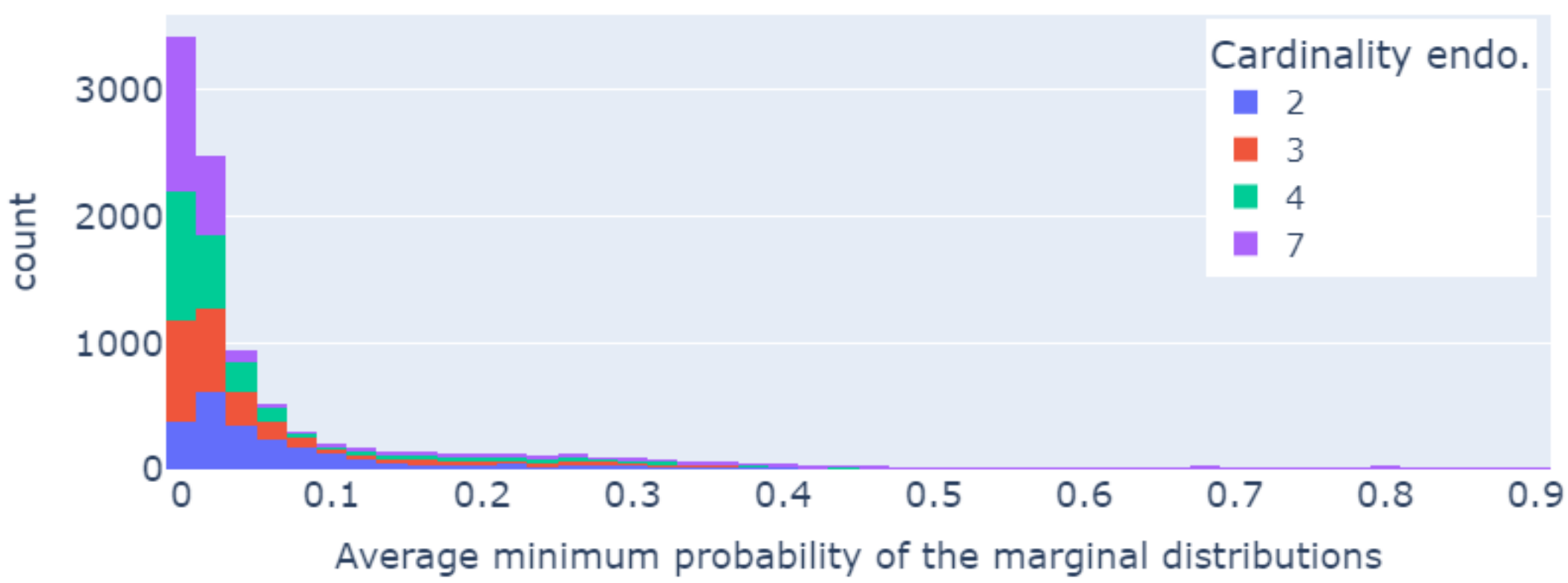

(b) Spearman

Figure 2: Distribution of the average minimum probability of the marginal entailed distributions of the generated SCMs depending on the number of variables and their cardinality

### G.2 Results

A number of findings about the distribution of the sampled SCMs can be drawn. For instance, the level of stochasticity of the SCMs roughly follows a long-tailed distribution whose mean increases with the number of variables and their cardinality. This can be seen in fig. 4.

Then, the levels of linearity and monotonicity (measured using Pearson and Spearman correlations respectively) follow roughly Gaussian distributions, see fig. 3. Distribution of mean of $0.3$ and a standard deviation of $0.1$ for linearity, while for monotonicity, the standard deviation increases to $0.2$. This means that, on average, the causal mechanisms are neither linear nor monotonic.

Moreover, the number and size of confounded components follow a roughly exponential distribution (i.e., high mass close to 0, followed by exponential decay) as depicted in fig. 6. Hence, "highly confounded" SCMs are rare.

Finally, the assumption of strong positivity is rarely respected for all kind of SCMs, whereas weak positivity is more often respected. In addition, there does not seem to be a correlation between the size of the SCMs (i.e., number of endogenous variables and their cardinality) and the validation of the positivity assumption. This is illustrated in figs. 2 and 5. Failure to respect these assumptions is a direct consequence of working with finite data where infinitesimal probabilities are rounded to 0.

As a result, the generated SCMs belong mainly to the non-identifiable domain of Causal ML methods, as positivity is poorly respected. Users must, therefore, be careful in their interpretations when evaluating methods, as identifiable SCMs are much less represented than non-identifiable ones. We recommend starting the evaluation on small SoIs close to the identifiable domain, before

progressively increasing the complexity of the causal datasets generated.

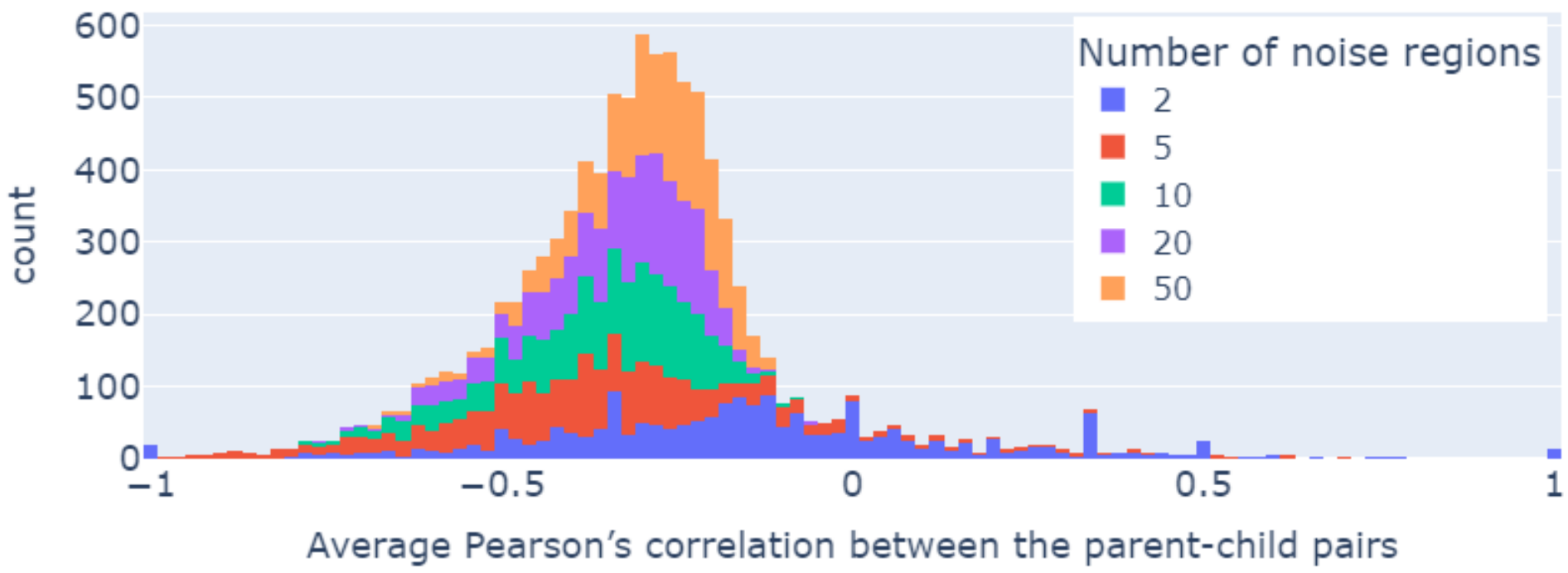

(a) Pearson

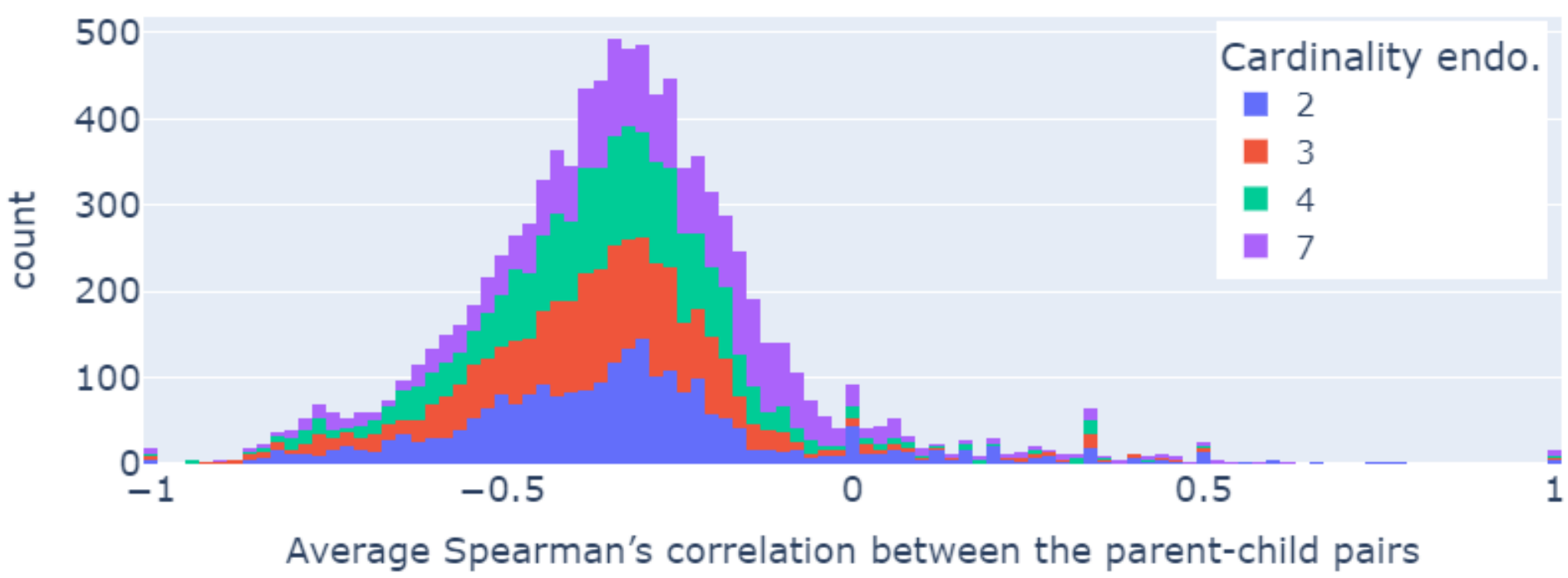

(b) Spearman

Figure 3: Distribution of the average Pearson’s and Spearman’s correlation between the parent-child pairs of the generated SCMs

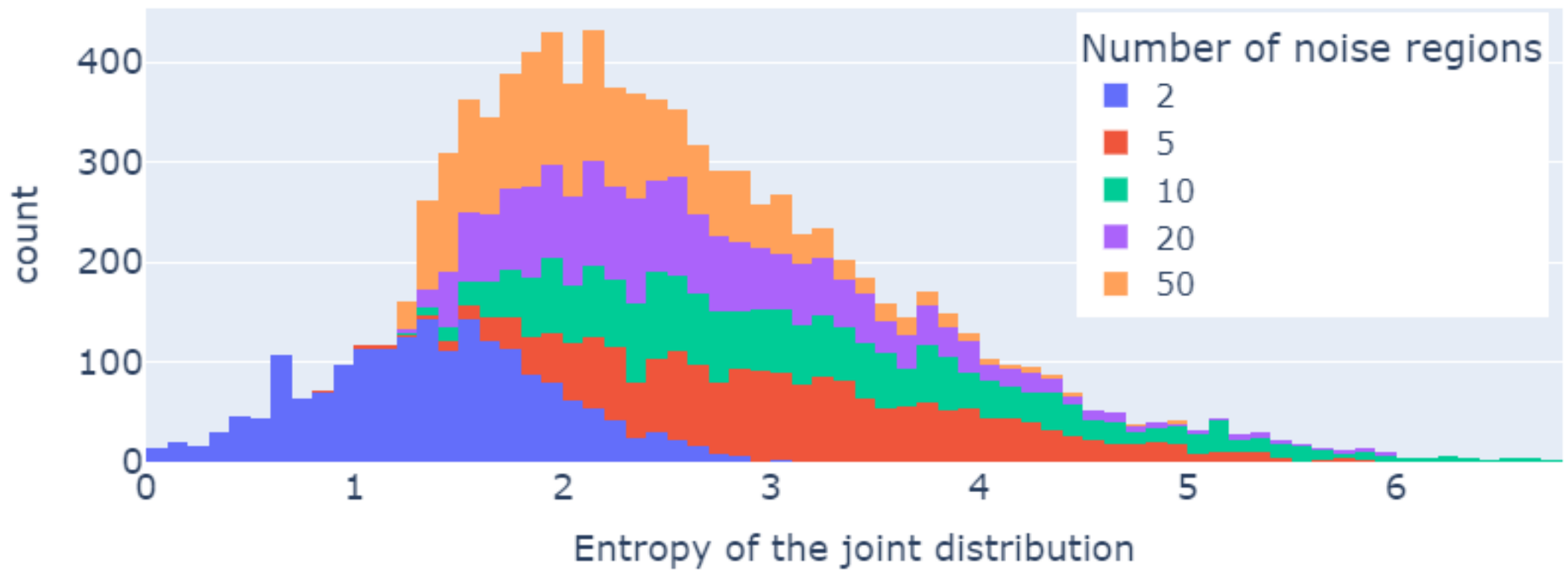

(a) Entropy distribution depending on the number of noise regions

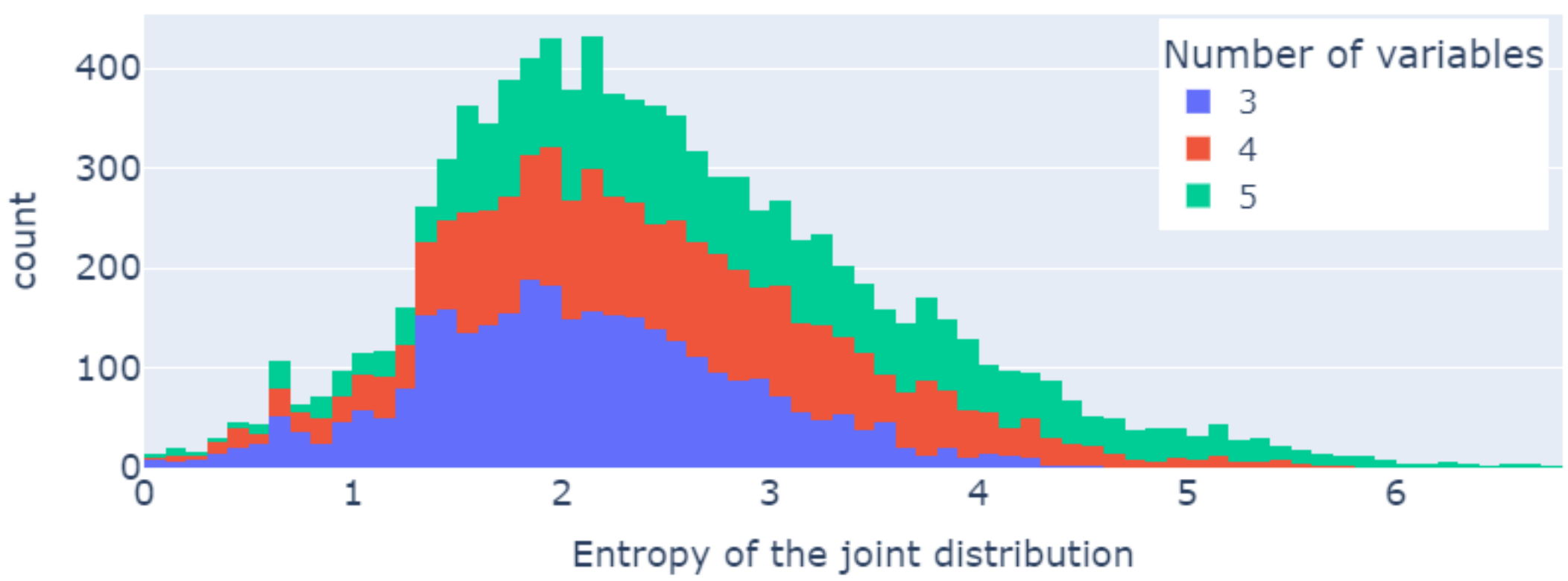

(b) Entropy distribution depending on the number of variables

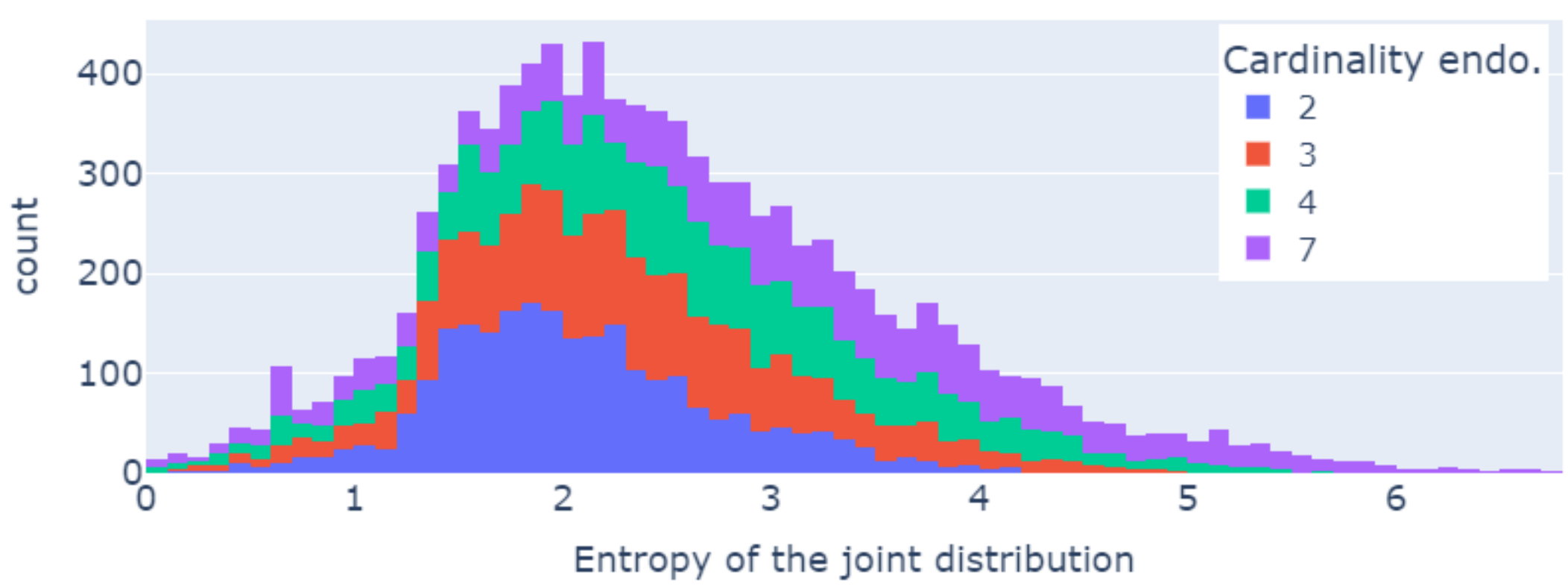

(c) Entropy distribution depending on the cardinality of variables

Figure 4: Distribution of the entropy of the entailed distribution of the generated SCMs depending on the number of noise regions, the number of variables and their cardinality

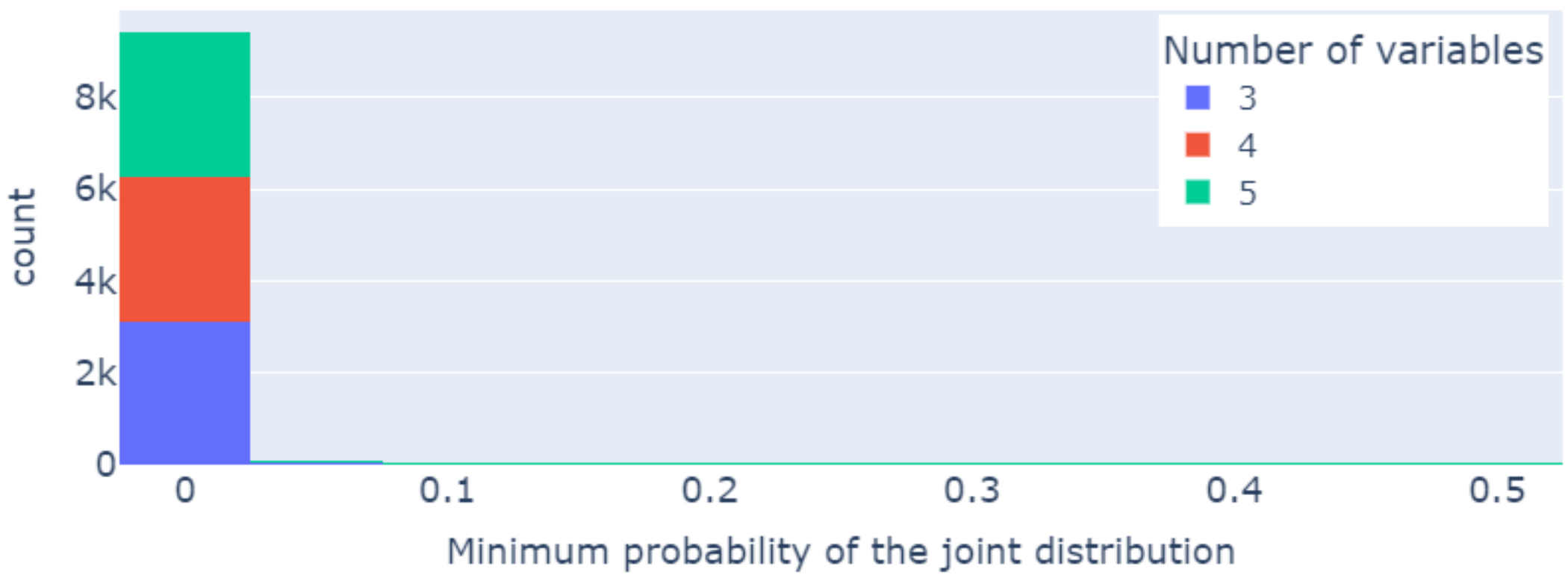

(a) Pearson

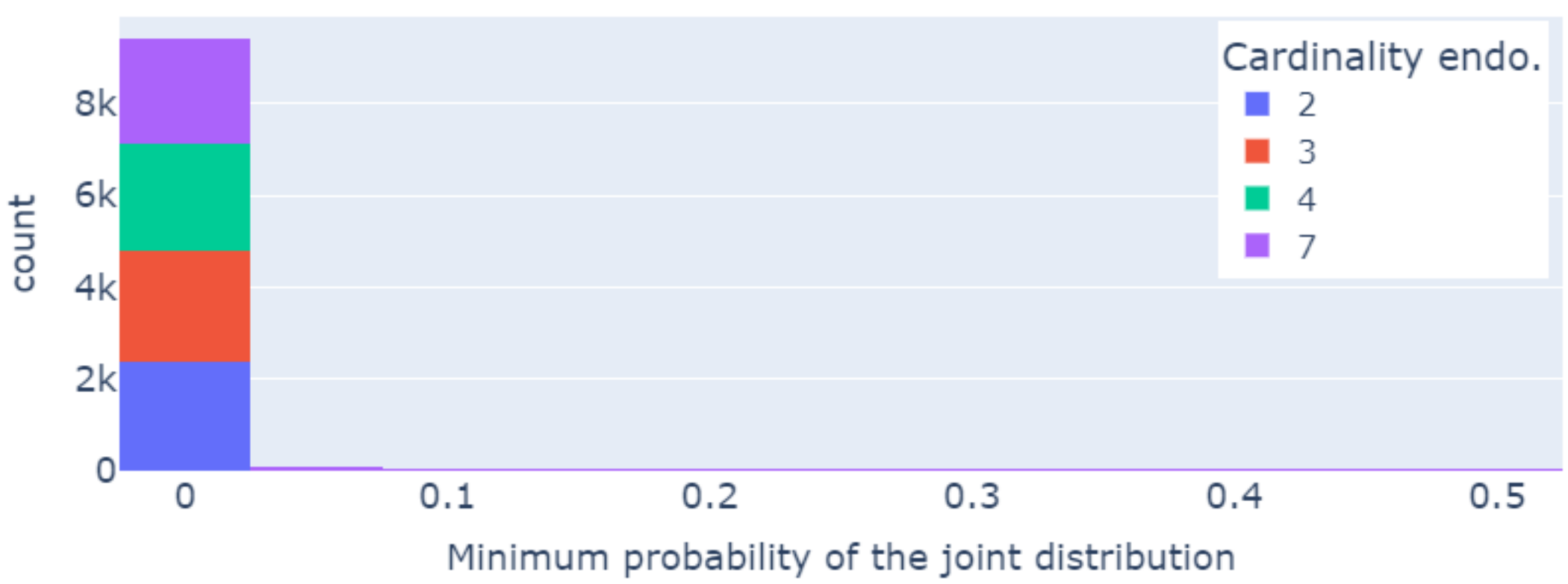

(b) Spearman

Figure 5: Distribution of the minimum probability of the joint entailed distribution of the generated SCMs depending on the number of variables and their cardinality

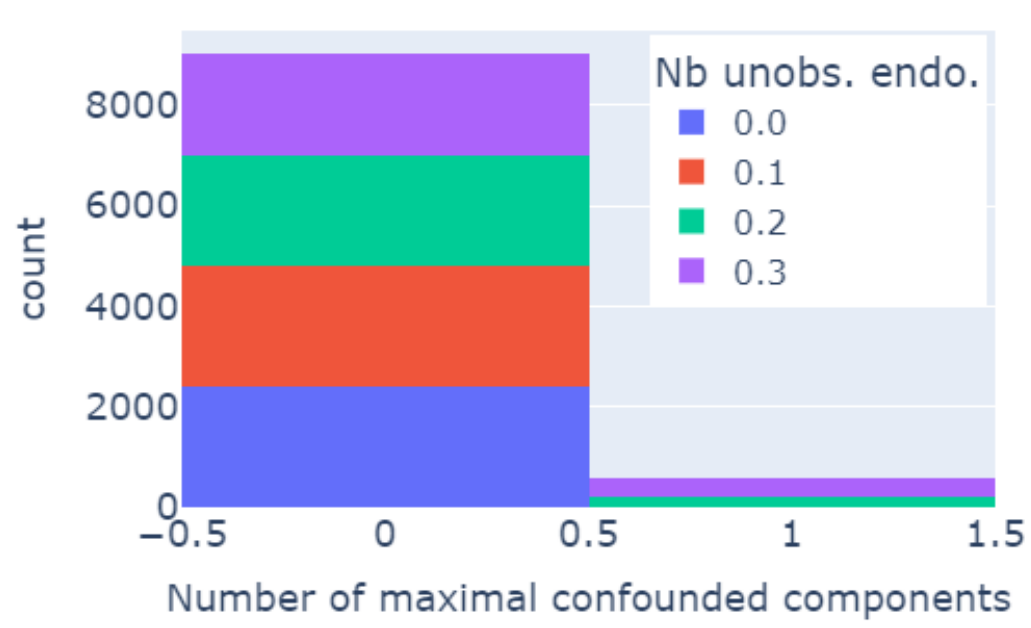

(a) Number of components

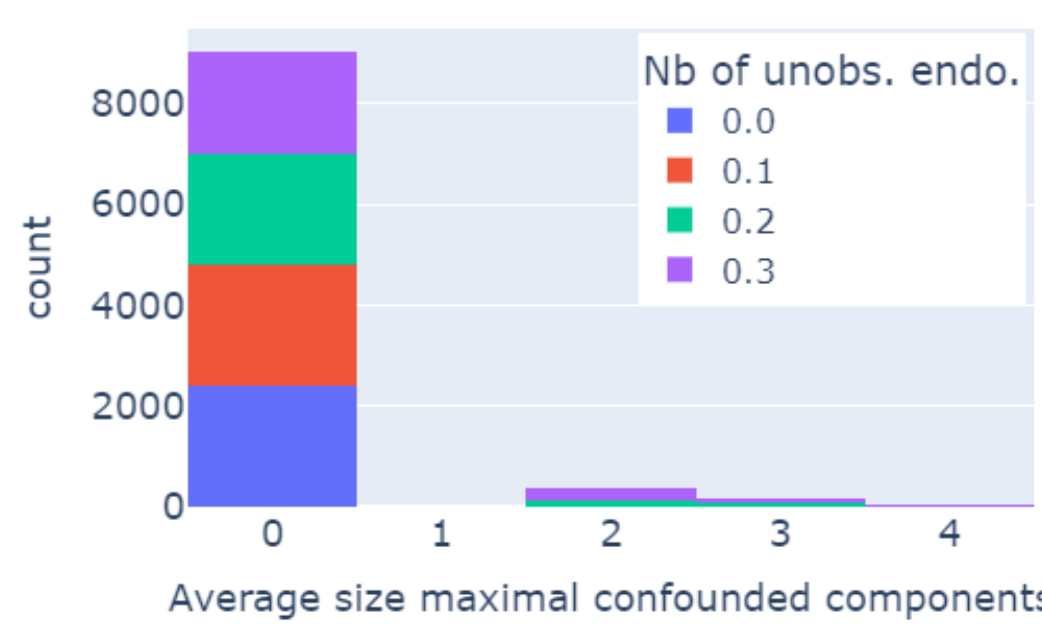

(b) Average size of the components

Figure 6: Distribution of the number and average size of maximal confounded components in the causal graphs of the generated SCMs depending on the number of unobserved variables

# H Visual overview of CausalProfiler's sampling strategy

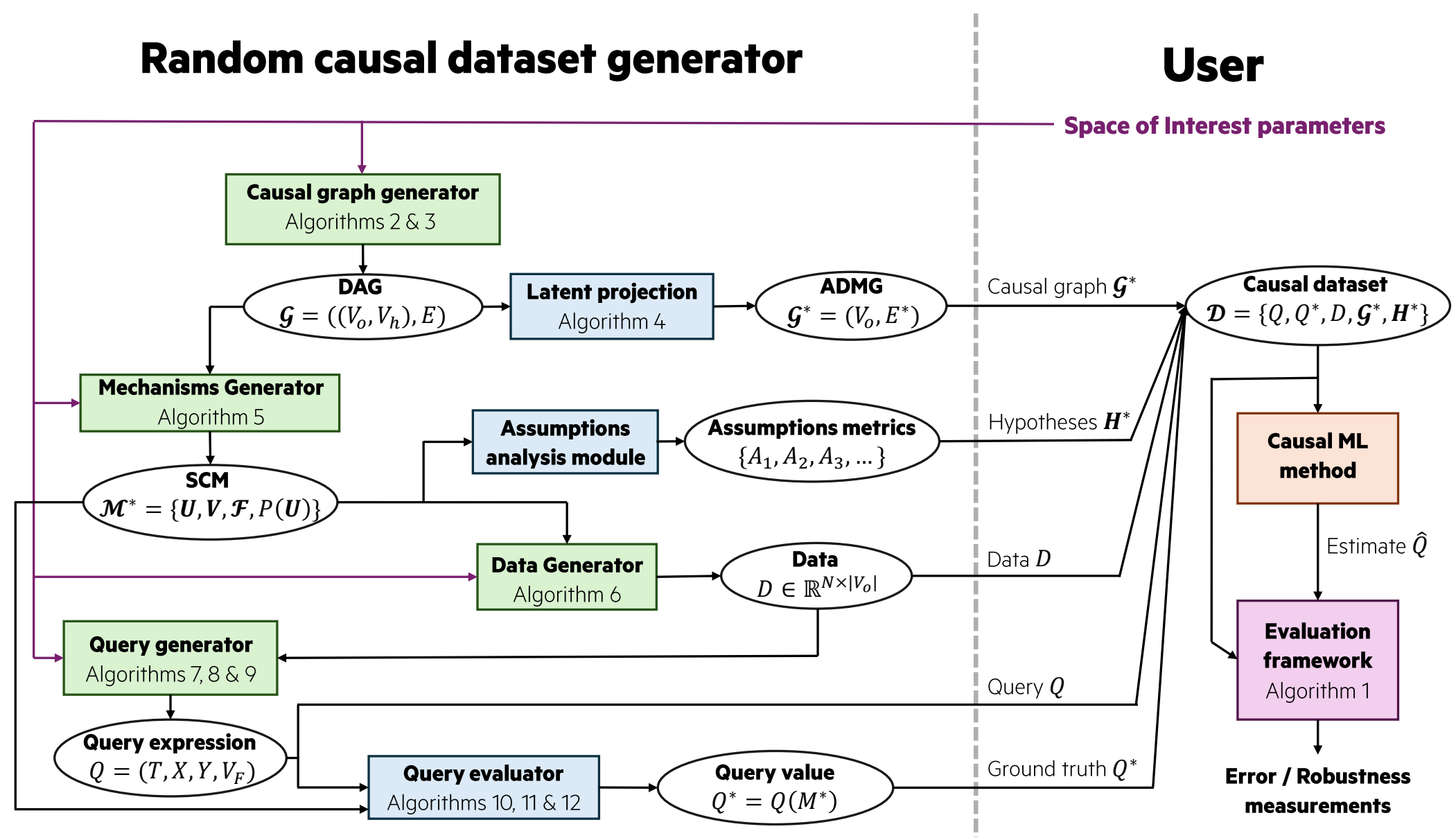

Figure 7: CausalProfiler structure. The left-hand side of the figure represents the code structure of the causal dataset generator. The right-hand side represents the user code. It illustrates how CausalProfiler can be used to evaluate a Causal ML method.

# I Proof of Theorem 5.1 (Coverage)

This section presents the proof of Theorem 5.1 stating that: For a Space of Interest $\mathcal{S} = \{\mathbb{M}, \mathbb{Q}, \mathbb{D}\}$, whose class of Structural Causal Models is a class of Regional Discrete SCMs with the maximum number of noise regions, any causal dataset $\mathcal{D} = \{Q, Q^\star, D, \mathcal{G}^\star, \mathbf{H}^\star\}$ has a strictly positive probability to be generated.

Firstly, let us note that:

- Stating that any query $Q$ can have any ground truth value $Q^\star$ given $\mathcal{S}$ is equivalent to saying that the class of considered SCMs, i.e., the class of Regional Discrete SCMs with the maximum number of noise regions, is $\mathcal{L}_3$-expressive with regards to the class of Markovian discrete SCMs (i.e., any $\mathcal{L}_3$-distribution of the class of Markovian discrete SCMs can be expressed with a Regional Discrete SCM).
- As the set of hypotheses $\mathbf{H}^\star$ can contain at most $\mathcal{L}_3$ conditions, if the class of considered SCMs is $\mathcal{L}_3$-expressive, then any set of hypotheses $\mathbf{H}^\star$ can be represented.
- If the class of considered SCMs is $\mathcal{L}_3$-expressive, then it is also $\mathcal{L}_1$-expressive, hence, $D$ can be sampled from any distribution

As a result, our proof consists of showing that $P(Q, \mathcal{G}^\star|\mathcal{S}) > 0$ and that the class of Regional Discrete SCMs with the maximum number of noise regions, denoted $\mathbb{M}_{\texttt{RD-SCM}, r=R_{\max}}$, is $\mathcal{L}_3$-expressive with regards to the class of Markovian discrete SCMs given an *SoI* $\mathcal{S}$ and a causal graph $\mathcal{G}$.

Let us consider a *SoI* $\mathcal{S} = \{\mathbb{M}, \mathbb{Q}, \mathbb{D}\}$ with $\mathbb{M} \subseteq \mathbb{M}_{\texttt{RD-SCM}, r=R_{\max}}$.

**Proving $P(\mathcal{G}^\star|\mathcal{S}) > 0$:**

$\mathcal{G}^\star$ is built through Algorithm 3 as the latent projection of a DAG $\mathcal{G} = \{(\mathbf{V}_H, \mathbf{V}_O), E\}$ over $\mathbf{V}_O$ where $\mathcal{G}$ is sampled using Algorithm 2. As a result, following the steps of Algorithms 2 and 3:

$$\begin{aligned}
P(\mathcal{G}^\star|\mathcal{S}) &= P(\{(\mathbf{V}_H, \mathbf{V}_O), E\}|\mathcal{S}) \\
&= P(E|\mathbf{V})P(\mathbf{V}_H, \mathbf{V}_O|\mathcal{S}) && \text{Edges are sampled independently of the observability of the variables} \\
&= P(E|\mathbf{V})P(\mathbf{V}_H, \mathbf{V}_O|\,|\mathbf{V}|)P(|\mathbf{V}|) && |\mathbf{V}| \text{ and } p_h \text{ are the only parameters influencing the observability of the variables} \\
&= P(E|\mathbf{V})P(\mathbf{V}_H, \mathbf{V}_O|\,|\mathbf{V}|)\frac{1}{N_{\max} - N_{\min}} && |\mathbf{V}| \sim \mathcal{U}[N_{\min}, N_{\max}] \\
&= P(E|\mathbf{V})\frac{|\mathbf{V}_H|!}{|\mathbf{V}|!}\frac{1}{N_{\max} - N_{\min}} && \mathbf{V}_H \subseteq \mathbf{V} \text{ sampled without replacement} \\
&= \frac{|\mathbf{V}_H|!}{|\mathbf{V}|!(N_{\max} - N_{\min})}P(E|\mathbf{V})
\end{aligned}$$

As $E = \{V_k \to V_i \,|\, V_k \in \boldsymbol{PA}(V_i), \forall V_i \in \mathbf{V}\}$ and the edges are sampled along the causal order $[1, N]$ with probability $p_{edge}$:

$$\begin{aligned}
P(\mathcal{G}^\star|\mathcal{S}) &= \frac{|\mathbf{V}_H|!}{|\mathbf{V}|!(N_{\max} - N_{\min})}\prod_{i=1}^{N} P(\{V_k \to V_i \,|\, V_k \in \boldsymbol{PA}(V_i)\}) \\
&= \frac{|\mathbf{V}_H|!}{|\mathbf{V}|!(N_{\max} - N_{\min})}\prod_{i=1}^{N} {p_{edge}}^{|\boldsymbol{PA}(V_i)|}(1 - p_{edge})^{i-1-|\boldsymbol{PA}(V_i)|}
\end{aligned}$$

Let us note that $p_{edge} = 0 \implies |\boldsymbol{PA}(V_i)| = 0$ and $p_{edge} = 1 \implies |\boldsymbol{PA}(V_i)| = i - 1$. As a result, $P(\mathcal{G}^\star|\mathcal{S}) > 0$.

**Proving that $\mathbb{M}_{\texttt{RD-SCM}, r=R_{\max}}$ is $\mathcal{L}_3$-expressive with regards to the class of Markovian discrete SCMs:** Regional discrete SCMs are, by construction, Markovian Canonical SCMs [49]. Furthermore, if the number of noise regions is chosen to be large enough (typically set to its maximum value), any Markovian Canonical SCM can be represented using a Regional Discrete SCM[6]. Thus, applying Zhang et al. [49] Theorem 2.4, we can assert that: for an arbitrary Markovian discrete SCM, there exists a Regional Discrete SCM such that they both have the same causal graph and the same $\mathcal{L}_3$-distribution. Consequently, the class of Regional Discrete SCMs is $\mathcal{L}_3$-expressive with respect to the class of Markovian discrete SCMs given the causal graph $\mathcal{G}$. Moreover, $P(\mathcal{G}) > 0$ for all $\mathcal{G}$ because $\prod_{i=1}^{N} {p_{edge}}^{|\boldsymbol{PA}(V_i)|}(1 - p_{edge})^{i-1-|\boldsymbol{PA}(V_i)|} > 0$ (cf. previous paragraph). Thus, more generally, the class of Regional Discrete SCMs sampled by our CausalProfiler is $\mathcal{L}_3$-expressive with respect to the class of Markovian SCMs.

**Proving $P(Q|\mathcal{G}^\star, \mathcal{S}) > 0$:** $Q$ is sampled given $\mathbb{Q}, D$ and $\mathcal{G}^\star$. Even though we currently only implement queries sampling for the classes $\mathcal{Q}_{\text{ATE}}$, $\mathcal{Q}_{\text{CATE}}$ and $\mathcal{Q}_{\text{Ctf-TE}}$ (cf. Appendix E and Algorithms 7, 8 and 9), we can generalize our proof to any other query class (e.g., CDE, NDE). We simply assume that these classes translate the set of constraints on the variables under consideration (e.g., conditioning variables have to be distinct from treatment variables or any other graphical constraints that can be checked with $\mathcal{G}^\star$) and express the probabilistic causal formula to be estimated. Once such a query class $\mathbb{Q}$ is defined, our method randomly samples variables from $\mathbf{V}_O$ in accordance with $\mathbb{Q}$ constraints and by sampling realizations from $D$. We showed in the previous paragraph that $\mathbb{M}_{\texttt{RD-SCM}, r=R_{\max}}$ is $\mathcal{L}_3$-expressive implying that it is $\mathcal{L}_1$-expressive too. So, any realization can be present in $D$. As a result, for a given query class $\mathbb{Q}$, any $Q$ can be generated. Hence, $P(Q|\mathcal{G}^\star, \mathcal{S}) > 0$.

[6] The distinction between $\mathbf{V}_O$ and $\mathbf{V}_H$ is of no importance for $\mathcal{L}_3$-expressiveness.$\mathbf{V}_O$ and $\mathbf{V}_H$ are only used to determine what will be visible to the user as benchmark.

**Proving Theorem 5.1 by combining previous results:** We proved that $\mathbb{M}_{\texttt{RD-SCM},r=R_{\max}}$ is $\mathcal{L}_3$-expressive, hence any training set $D$, ground truth query $Q^\star$ and set of hypotheses $\mathbf{H}^\star$ can be generated given an *SoI* $\mathcal{S}$, a causal graph $\mathcal{G}$ and a causal query $Q$. In addition, $P(Q, \mathcal{G}^\star|\mathcal{S}) = P(Q|\mathcal{G}^\star, \mathcal{S})P(\mathcal{G}^\star|\mathcal{S})$ and we also prove that $P(Q|\mathcal{G}^\star, \mathcal{S}) > 0$ and $P(\mathcal{G}^\star|\mathcal{S}) > 0$. Hence, $P(Q, \mathcal{G}^\star|\mathcal{S}) > 0$. As a result, any causal dataset $\mathcal{D}$ has a strictly positive probability to be generated.

**Remark on continuous SCMs.** The universal approximation theorem [24] states that NNs (with non-polynomial activation functions) are dense in the space of continuous functions, meaning that any continuous function can be approximated by a sequence of NNs converging to this function. However, this does not guarantee that they strictly cover the space of continuous functions. In particular, whenever the number of layers and neurons is finite, one can always build a continuous function too complex to be represented with this finite number of parameters. Hence, Theorem 5.1 cannot be extended to any class of continuous SCMs. However, it could potentially be adapted not to ask for strict coverage but rather density. We leave this question for future work.

# J Verfication Results

We design and run verification experiments targeting each level of the Pearl Causal Hierarchy (PCH).

## J.1 $\mathcal{L}_1$ verification

Consistency with $\mathcal{L}_1$ level of the Pearl Causal Hierarchy (PCH) is tested through the verification that the Markov property holds on randomly sampled regional discrete SCMs. Below is a description of the experimental design choices made and the associated results.

### J.1.1 Experiment

For a given SCM $\mathcal{M} := \{\mathbf{V}, \mathbf{U}, \boldsymbol{\mathcal{F}}, P(\mathbf{U})\}$, we check that the Markov property is satisfied by assessing whether there is a statistically significant amount of d-separations not leading to conditional independence in the entailed distribution.
To do so, we first enumerate the list of sets of variables $(\mathbf{A}, \mathbf{B}, \mathbf{C})$ in $\mathbf{V}$ corresponding to d-separations in $\mathcal{M}$'s causal graph $\mathcal{G}_\mathcal{M}$, ie $\mathbf{A} \perp\!\!\!\perp_{\mathcal{G}_\mathcal{M}} \mathbf{B}|\mathbf{C}$. Second, for each d-separated set $(\mathbf{A}, \mathbf{B}, \mathbf{C})$, we test whether $\mathbf{A} \perp\!\!\!\perp_{P_\mathcal{M}} \mathbf{B}|\mathbf{C}$ by sampling 50k data points from the entailed distribution $P_\mathcal{M}$.

In practice, enumerating all the d-separations can be very costly. Moreover, as the set of variables $\mathbf{C}$ increases, it becomes increasingly complicated to robustly test the conditional independence $\mathbf{A} \perp\!\!\!\perp_{P_\mathcal{M}} \mathbf{B}|\mathbf{C}$. Indeed, as the cardinality of $\mathbf{C}$ increases, so does the number of combinations of values for which to test independence between variables $\mathbf{A}$ and $\mathbf{B}$. Running the statistical test becomes costly, and the data volume required for robust independence test results increases exponentially. This is why we limit ourselves to listing the d-separated sets $(A, B, \mathbf{C})$ such that $A \in \mathbf{V}$, $B \in \mathbf{V} \backslash A$, and $C \in \mathbf{V} \cup \mathbf{V}^2 \cup \mathbf{V}^3$ by enumerating all the possible $(A, B, \mathbf{C})$ tuples, and testing whether they are d-separated in $\mathcal{G}_\mathcal{M}$.

As the sampled SCMs are regional discrete, the conditional independence $A \perp\!\!\!\perp_{P_\mathcal{M}} B|\mathbf{C}$ can be tested with Pearson's $\chi^2$ independence tests [37]. More precisely, $A$ and $B$ are considered independent conditionally to $\mathbf{C}$ if for all values $\mathbf{c}$ of $\mathbf{C}$, the $H_0$ hypothesis "$A$ and $B$ are independent" is not rejected. Since Pearson's $\chi^2$ test is based on the assumption that the number of samples is large, we decide to skip tests where the Koehler criterion [31] is not met. Based on empirical analyses, this criterion indicates whether the $\chi^2$ test is reliable depending on the number of samples considered. In addition, as we conduct tests for each observed value $c$, we need to control for the expected proportion of false positives (represented by the Type I error of the test). To do so, we apply the Benjamini-Hochberg correction [5].

For each SoI, defined by the Cartesian product of the following parameters, we sample 5 SCMs:

- **Number of endogenous variables**: $[4, 5, 6]$
- **Expected edge probability**: $[0.1, 0.4]$

- **Proportion of unobserved endogenous variables**: set to 0 because the Markov property only hold for Markovian SCMs
- **Number of noise regions**: $[5, 10]$
- **Cardinality of endogenous variables**: $[2, 3, 10]$
- **Distribution of exogenous variables**: set to $\mathcal{U}[0, 1]$
- **Number of data points**: 50000

#### J.1.2 Results

| Conditioning set size | $A \perp\!\!\!\perp_{P_{\mathcal{M}}} B \mid \mathbf{C}$ tests | | | | $\chi^2$ independence tests | | | |
|---|---|---|---|---|---|---|---|---|
| | Total | Pass | Fail | Skip | Total | Pass | Fail | Skip |
| $\lvert\mathbf{C}\rvert = 1$ | 100 (2 391) | 91.76 (2 194) | 4.94 (118) | 3.3 (79) | 100 (9 130) | 85.4 (7 797) | 1.43 (131) | 13.17 (1 202) |
| $\lvert\mathbf{C}\rvert = 2$ | 100 (2 986) | 91.16 (2 722) | 5.63 (168) | 3.22 (96) | 100 (53 040) | 45.2 (23 976) | 0.33 (177) | 54.46 (28 887) |
| $\lvert\mathbf{C}\rvert = 3$ | 100 (1 693) | 91.08 (1 542) | 5.67 (96) | 3.25 (55) | 100 (145 320) | 18.49 (26 874) | 0.07 (106) | 81.43 (118 340) |
| TOTAL | 100 (7 070) | 91.34 (6 458) | 5.40 (382) | 3.25 (230) | 100 (207 490) | 28.26 (58 647) | 0.2 (414) | 71.54 (148 429) |

Table 4: Conditional independence tests based on $\chi^2$ independence tests to assess compliance of sampled SCMs with the Markov property. Results are expressed as a percentage of the total of each test type for each conditioning set size. The number of tests is also shown in brackets.

The experimental results are summarized in table 4, where it can be seen that $5.4\%$ of the conditional independence tests failed. Despite the use of the Koehler criterion and Benjamini-Hochberg correction, some tests can still be rejected due to the random nature of finite data sampling, which can produce slight artificial correlations in the data. Moreover, on closer inspection, the majority of the failed tests (at least 350 out of 382) are unsuccessful because of a single failed $\chi^2$ independence test. This reinforces our previous argument about the random nature of finite data sampling.

One can also notice that the number of skipped $\chi^2$ independence tests increases with the size of the conditioning set. Such behavior is to be expected, since the number of realizations of the conditioning set increases exponentially with its cardinality, while the number of observations sampled to perform the independence tests remains constant. As a result, there are fewer and fewer observations available to perform each $\chi^2$ test. In contrast, the number of fully skipped conditional independence tests remains constant. This means that the $\chi^2$ skipped tests are relatively homogeneously distributed across all the conditional independence tests.

Someone might argue that the number of sampled observations should simply be automatically computed to verify the Koehler criterion. However, in general, such a calculation is complicated, if not impossible, to automate, as causal mechanisms are randomly sampled. As a result, all kinds of observational distributions can be induced with potentially very low probability realizations, for which the Koehler criterion could never be validated because the number of data to be sampled would be too large.

To conclude, these results are sufficient to conclude that the Markov property is empirically verified by the sampled SCMs.

### J.2 $\mathcal{L}_2$ verification

Consistency with $\mathcal{L}_2$ level of the PCH is tested through the verification that the Do-calculus rules hold on randomly sampled regional discrete SCMs. Below is a description of the experimental design choices made (Section J.2.1) and the associated results (Section J.2.2).

### J.2.1 Experiment

**Definition J.1. Do-Calculus rules** [35]
Given an SCM $\mathcal{M} := \{\mathbf{V}, \mathbf{U}, \boldsymbol{\mathcal{F}}, P(\mathbf{U})\}$ whose causal graph $\mathcal{G}$ is a DAG, and disjoint subsets $\mathbf{X}, \mathbf{Y}, \mathbf{Z}$, and $\mathbf{W}$ of $\mathbf{V}$, the rules of the **Do-Calculus** are defined as follows:

1. **Insertion/deletion of observation**: if $\mathbf{Y}$ and $\mathbf{Z}$ are d-separated by $\mathbf{X} \cup \mathbf{W}$ in $\mathcal{G}_{\overline{\mathbf{X}}}$, then $P(\mathbf{Y}|\boldsymbol{do}(\mathbf{X}=\mathbf{x}), \mathbf{W}, \mathbf{Z}) = P(\mathbf{Y}|\boldsymbol{do}(\mathbf{X}=\mathbf{x}), \mathbf{W})$
2. **Action/observation exchange**: if $\mathbf{Y}$ and $\mathbf{Z}$ are d-separated by $\mathbf{X} \cup \mathbf{W}$ in $\mathcal{G}_{\overline{\mathbf{X}},\underline{\mathbf{Z}}}$, then $P(\mathbf{Y}|\boldsymbol{do}(\mathbf{X}=\mathbf{x}), \boldsymbol{do}(\mathbf{Z}=\mathbf{z}), \mathbf{W}) = P(\mathbf{Y}|\boldsymbol{do}(\mathbf{X}=\mathbf{x}), \mathbf{Z}, \mathbf{W})$
3. **Insertion/deletion of action**: if $\mathbf{Y}$ and $\mathbf{Z}$ are d-separated by $\mathbf{X} \cup \mathbf{W}$ in $\mathcal{G}_{\overline{\mathbf{X}},\overline{\mathbf{Z}(\mathbf{W})}}$, then $P(\mathbf{Y}|\boldsymbol{do}(\mathbf{X}=\mathbf{x}), \boldsymbol{do}(\mathbf{Z}=\mathbf{z}), \mathbf{W}) = P(\mathbf{Y}|\boldsymbol{do}(\mathbf{X}=\mathbf{x}), \mathbf{W})$

where $\mathcal{G}_{\overline{\mathbf{X}}}$ (resp. $\mathcal{G}_{\underline{\mathbf{X}}}$) represents the graph $\mathcal{G}$ where the incoming edges in (resp. outgoing edges from) $\mathbf{X}$ have been removed and $\mathbf{Z}(\mathbf{W})$ is the subset of nodes in $\mathbf{Z}$ that are not ancestors of any node in $\mathbf{W}$ in $\mathcal{G}_{\overline{\mathbf{X}}}$

**Theorem J.1. Soundness and Completeness of the Do-Calculus rules** [25]
The rules of the do-calculus are **sound** and **complete**; that is, they hold in all causal models, and all identifiable intervention distributions can be computed by an iterative application of these three rules.

For a given SCM, we check each rule by first enumerating the sets of d-separated variables of interest. Second, for each d-separated set, we test whether the distributions are statistically significantly similar by sampling 50k data points from the intervened SCMs and testing whether they are drawn from the same distribution.

For the same computational cost reasons as for $\mathcal{L}_1$ verification, we consider only univariate sets of variables $X, Y, Z$, and $W$. In addition, the studied SCMs are sampled from the same *SoIs* as defined in the $\mathcal{L}_1$-verification experiment (Section J.1.1). Finally, to assess whether two conditional distributions are identical, we used Pearson's $\chi^2$ goodness of fit tests [37]. As done in Section J.1, we also use the Koehler criterion [31] and the Benjamini-Hochberg correction [5].

For each SoI, defined by the Cartesian product of the following parameters, we sample 5 SCMs:

- **Number of endogenous variables**: $[4, 5, 6]$
- **Expected edge probability**: $[0.1, 0.4]$
- **Proportion of unobserved endogenous variables**: set to 0 because the Markov property only hold for Markovian SCMs
- **Number of noise regions**: $[5, 100]$
- **Cardinality of endogenous variables**: $[2, 5]$
- **Distribution of exogenous variables**: set to $\mathcal{U}[0, 1]$
- **Number of data points**: $50000$

### J.2.2 Results

The experimental results are summarized in table 5 where it can be seen that they are very similar to the $\mathcal{L}_1$ verification ones: roughly 6% of the conditional goodness of fit tests were not validated, some tests are rejected due to the random nature of finite data sampling but the majority them (at least 570 out of 755) are unsuccessful because of a single failed $\chi^2$ goodness of fit test.

One can also notice that the percentage of skipped $\chi^2$ goodness of fit tests is similar for rules 1 and 3 but increases by roughly 50% for rule 2. Such behavior is to be expected as rule 2 is the only rule to have conditioning sets of size 3 on both sides of the equality. However, the number of skipped tests remains low, with a maximum of 16%.

| | Cond. goodness of fit | | | | $\chi^2$ goodness of fit | | | |
|---|---|---|---|---|---|---|---|---|
| **Do-Calculus Rule** | Total | Pass | Fail | Skip | Total | Pass | Fail | Skip |
| **Rule 1** Insertion/deletion of observation | 100 (3 378) | 96.15 (3 248) | 3.85 (130) | 0 (0) | 100 (171 092) | 88.84 (152 004) | 0.1 (172) | 11.06 (18 916) |
| **Rule 2** Action/observation exchange | 100 (5 065) | 94.04 (4 763) | 5.96 (302) | 0 (0) | 100 (259 509) | 83.84 (217 578) | 0.09 (241) | 16.06 (41 690) |
| **Rule 3** Insertion/deletion of action | 100 (5 169) | 93.75 (4 846) | 6.25 (323) | 0 (0) | 100 (282 184) | 89.21 (251 731) | 0.06 (157) | 10.74 (30 296) |
| **TOTAL** | 100 (13 612) | 94.45 (12 857) | 5.55 (755) | 0 (0) | 100 (712 785) | 87.17 (621 313) | 0.08 (570) | 12.75 (90 902) |

Table 5: Conditional independence tests based on $\chi^2$ goodness of fit tests to assess compliance of sampled SCMs with the Do-Calculus rules. Results are expressed as a percentage of the total of each test type for each conditioning set size. The number of tests is also shown in brackets.

As a result, we estimate that these results are sufficient to conclude that the Do-calculus rules are respected by the sampled SCMs.

### J.3 $\mathcal{L}_3$ verification

Consistency with $\mathcal{L}_3$ level of the PCH is tested through the verification that the axiomatic characterization of structural counterfactuals holds on randomly sampled regional discrete SCMs. Below is a description of the experimental design choices made (Section J.3.1) and the associated results (Section J.3.2).

**Definition J.2. Axiomatic characterization of structural counterfactuals** [35]
Given an SCM $\mathcal{M} := \{\mathbf{V}, \mathbf{U}, \boldsymbol{\mathcal{F}}, P(\mathbf{U})\}$ whose causal graph $\mathcal{G}$ is a DAG, the **axioms of structural counterfactuals** are defined as follows:

1. **Composition**: For any sets of endogenous variables $\mathbf{X}$, $\mathbf{Y}$, and $\mathbf{W}$ in $\mathbf{V}$ and any realization $\mathbf{u}$ of $\mathbf{U}$, if $\mathbf{W}_{\boldsymbol{do}(\mathbf{X}=\mathbf{x})}(\mathbf{u}) = \mathbf{w}$ then $\mathbf{Y}_{\boldsymbol{do}(\mathbf{X}=\mathbf{x}),\boldsymbol{do}(\mathbf{W}=\mathbf{w})}(\mathbf{u}) = \mathbf{Y}_{\boldsymbol{do}(\mathbf{X}=\mathbf{x})}(\mathbf{u})$
2. **Effectiveness**: For any disjoint sets of endogenous variables $\mathbf{X}$, and $\mathbf{W}$ in $\mathbf{V}$ and any realization $\mathbf{u}$ of $\mathbf{U}$, $\mathbf{X}_{\boldsymbol{do}(\mathbf{X}=\mathbf{x}),\boldsymbol{do}(\mathbf{W}=\mathbf{w})}(\mathbf{u}) = \mathbf{x}$
3. **Reversibility**: For any two distinct variables $Y$ and $W$ and any sets of other variables $\mathbf{X}$ in $\mathbf{V}$ and any realization $\mathbf{u}$ of $\mathbf{U}$, if $Y_{\boldsymbol{do}(\mathbf{X}=\mathbf{x}),\boldsymbol{do}(W=w)}(\mathbf{u}) = y$ and $W_{\boldsymbol{do}(\mathbf{X}=\mathbf{x}),\boldsymbol{do}(Y=y)}(\mathbf{u}) = w$ then $Y_{\boldsymbol{do}(\mathbf{X}=\mathbf{x})}(\mathbf{u}) = y$

Note that we do not write $P(\mathbf{W}_{\boldsymbol{do}(\mathbf{X}=\mathbf{x})}|\mathbf{U})$ but rather $\mathbf{W}_{\boldsymbol{do}(\mathbf{X}=\mathbf{x})}(\mathbf{u})$ as it is a deterministic expression. Indeed, if $\mathbf{U}$ is fixed, there is no stochastically anymore, so we no longer need to reason in distributions but rather in functional forms.

**Theorem J.2. Soundness and Completeness of structural counterfactual axioms** [15]
Completeness, effectiveness, and reversibility are **sound** and **complete** in structural causal model semantics; that is they hold in all causal models and all identifiable counterfactual distributions can be computed by an iterative application of these three axioms.

#### J.3.1 Experiment

For a given SCM, using definition J.1 notations, we check that:

1. The **Composition** axiom is satisfied by assessing whether $\mathbf{W}_{\boldsymbol{do}(\mathbf{X}=\mathbf{x})}(\mathbf{u}) = \mathbf{w}$ implies $\mathbf{Y}_{\boldsymbol{do}(\mathbf{X}=\mathbf{x}),\boldsymbol{do}(\mathbf{W}=\mathbf{w})}(\mathbf{u}) = \mathbf{Y}_{\boldsymbol{do}(\mathbf{X}=\mathbf{x})}(\mathbf{u})$ for any sets of endogenous variables $\mathbf{X}$, $\mathbf{Y}$, and $\mathbf{W}$ in $\mathbf{V}$ and any realization $\mathbf{u}$ of $\mathbf{U}$

2. The **Effectiveness** axiom is satisfied by assessing whether $\mathbf{X}_{\boldsymbol{do}(\mathbf{X}=\mathbf{x}),\boldsymbol{do}(\mathbf{W}=\mathbf{w})}(\mathbf{u}) = \mathbf{x}$ for any sets of endogenous variables $\mathbf{X}$, and $\mathbf{W}$ in $\mathbf{V}$ and any realization $\mathbf{u}$ of $\mathbf{U}$

3. The **Reversibility** axiom is satisfied by assessing whether $Y_{\boldsymbol{do}(\mathbf{X}=\mathbf{x}),\boldsymbol{do}(W=w)}(\mathbf{u}) = y$ and $W_{\boldsymbol{do}(\mathbf{X}=\mathbf{x}),\boldsymbol{do}(Y=y)}(\mathbf{u}) = w$ implies $Y_{\boldsymbol{do}(\mathbf{X}=\mathbf{x})}(\mathbf{u}) = y$ for any two (distinct) variables $Y$ and $W$ and any sets of variables $\mathbf{X}$ in $\mathbf{V}$ and any realization $\mathbf{u}$ of $\mathbf{U}$

For each SoI, defined by the Cartesian product of the following parameters, we sample 5 SCMs:

- **Number of endogenous variables**: $[3, 5, 10]$
- **Expected edge probability**: $[0.1, 0.5, 0.7]$
- **Proportion of unobserved endogenous variables**: set to 0 because the Markov property only hold for Markovian SCMs
- **Number of noise regions**: $[3, 5, 10]$
- **Cardinality of endogenous variables**: $[2, 5, 7]$
- **Distribution of exogenous variables**: set to $\mathcal{U}[0, 1]$
- **Number of data points**: $50000$

For each SCM, instead of enumerating all the possible four sets of variables $\mathbf{X}$, $\mathbf{Y}$ and $\mathbf{W}$, we sample a partition of three elements of a randomly sampled subset of $\mathbf{V}$ of a size randomly picked in $[3, |\mathbf{V}|]$. This sampling strategy enables us to make sure the three sets are disjoint and of randomly varying size. In addition, for each four sets, we sample $50$k realizations of $\mathbf{U}$.

Let us note that the axioms now correspond to exact realizations and not equal probabilities. As a result, we expect no failure as no approximation is made in this experiment.

### J.3.2 Results

As expected, all the tested equalities are verified in our experiments. We can, therefore, consider that the SCMs created by our generator allows the estimation of any structural counterfactual queries.

## K Extended Experimental Results

### K.1 Experiment 1: Additional Information

We provide more details about the *SoI* used in our experiments in Table 6 and present extended performance metrics in Table 7, complementing those already shown in Table 1. Parameters not explicitly listed for a given *SoI* are set to their default values as per the benchmark configuration. Neural Networks for our experiments have two 8-neuron layers and use ReLU activation.

Table 6: Specification of each *SoI* used in the general experiments. $N$ denotes the sampled number of nodes.

| **Name** | `Linear-Medium` |
|---|---|
| # Nodes | 15-20 |
| Mechanism | Linear |
| Expected Edges | $2 \times N$ |
| Variable Type | Continuous |
| Samples | 1000 |
| Query Type | ATE |
| Seeds | [10, 11, 12, 13, 14] |

| **Name** | `NN-Medium` |
|---|---|
| # Nodes | 15-20 |
| Mechanism | NN |
| Expected Edges | $2 \times N$ |
| Variable Type | Continuous |
| Samples | 1000 |
| Query Type | ATE |
| Seeds | [10, 11, 12, 13, 14] |

| **Name** | `NN-Large` |
|---|---|
| # Nodes | 20-25 |
| Mechanism | NN |
| Expected Edges | $2 \times N$ |
| Variable Type | Continuous |
| Samples | 1000 |
| Query Type | ATE |
| Seeds | [10, 11, 12, 13, 14] |

| **Name** | `NN-Large-LowData` |
|---|---|
| # Nodes | 20-25 |
| Mechanism | NN |
| Expected Edges | $2 \times N$ |
| Variable Type | Continuous |
| Samples | 50 |
| Query Type | ATE |
| Seeds | [10, 11, 12, 13, 14] |

Table 7: Additional performance metrics of CausalNF, DCM, NCM, and VACA on the general experiments.

| Space | Method | Min Error | Total Fail | Runtime Mean | Runtime Std |
|---|---|---|---|---|---|
| Linear-Medium | CausalNF | 0.0024 | 0 | 27.58 s | 18.33 s |
| Linear-Medium | DCM | 0.0086 | 0 | 33.08 s | 9.71 s |
| Linear-Medium | NCM | 0.0024 | 0 | 14.77 s | 1.42 s |
| Linear-Medium | VACA | 0.0038 | 1335 | 11.69 s | 4.54 s |
| NN-Medium | CausalNF | 0.0019 | 0 | 21.47 s | 19.52 s |
| NN-Medium | DCM | 0.0073 | 0 | 31.79 s | 10.62 s |
| NN-Medium | NCM | 0.0014 | 0 | 14.65 s | 1.43 s |
| NN-Medium | VACA | 0.0024 | 125 | 12.13 s | 4.41 s |
| NN-Large | CausalNF | 0.0038 | 0 | 30.23 s | 25.33 s |
| NN-Large | DCM | 0.0060 | 0 | 38.33 s | 14.02 s |
| NN-Large | NCM | 0.0018 | 0 | 18.90 s | 1.38 s |
| NN-Large | VACA | 0.0023 | 290 | 12.88 s | 4.31 s |
| NN-Large-LowData | CausalNF | 0.0086 | 0 | 44.28 s | 17.10 s |
| NN-Large-LowData | DCM | 0.0121 | 0 | 4.82 s | 1.34 s |
| NN-Large-LowData | NCM | 0.0013 | 0 | 0.81 s | 0.11 s |
| NN-Large-LowData | VACA | 0.0010 | 0 | 10.43 s | 4.59 s |

### K.2 Experiment 2: Additional Information

We provide more details about the *SoI* used in our experiments in Table 8 and present extended performance metrics in Table 9, complementing those already shown in Table 2. Parameters not explicitly listed for a given *SoI* are set to their default values as per the benchmark configuration.

Table 8: Specification of the Spaces of Interest used for evaluating discrete SCMs with CTF-TE queries. $N$ denotes the sampled number of nodes.

| **Name** | `Disc-C2-Reject` |
|---|---|
| # Nodes | 10–15 |
| # Categories | 2 |
| Mechanism | Tabular |
| Sampling Strategy | Rejection |
| Edges | $N$ |
| Samples | 500 |
| Query Type | Ctf-TE |
| Seeds | [1, 2, 3, 4, 5] |

| **Name** | `Disc-C4-Unbias` |
|---|---|
| # Nodes | 10–15 |
| # Categories | 4 |
| Mechanism | Tabular |
| Sampling Strategy | Random |
| Edges | $N$ |
| Samples | 500 |
| Query Type | Ctf-TE |
| Seeds | [1, 2, 3, 4, 5] |

| **Name** | `Disc-Large-C2-Unbias` |
|---|---|
| # Nodes | 20–30 |
| # Categories | 2 |
| Mechanism | Tabular |
| Sampling Strategy | Random |
| Edges | $N$ |
| Samples | 500 |
| Query Type | Ctf-TE |
| Seeds | [1, 2, 3, 4, 5] |

Table 9: Additional performance metrics of CausalNF and DCM on the discrete experiments.

| Space | Method | Min Error | Total Fail | Runtime Mean | Runtime Std |
|---|---|---|---|---|---|
| Disc-C2-Reject | CausalNF | 0.0000 | 202 | 0.46 s | 0.04 s |
| Disc-C2-Reject | DCM | 0.0000 | 107 | 8.81 s | 3.55 s |
| Disc-C4-Unbias | CausalNF | 0.0000 | 1017 | 0.42 s | 0.03 s |
| Disc-C4-Unbias | DCM | 0.0000 | 565 | 7.68 s | 3.43 s |
| Disc-Large-C2-Unbias | CausalNF | NaN | 2500 | 0 s | 0 s |
| Disc-Large-C2-Unbias | DCM | 0.0000 | 283 | 16.39 s | 6.42 s |

### K.3 Experiment 3: ATE Estimation under Hidden Confounding

In this experiment, we demonstrate how our framework can be used to evaluate methods in the presence of latent confounders — a common challenge in real-world causal inference. A key goal here is not only to confirm theoretical limitations but to investigate how quickly and severely performance degrades when assumptions are violated. While theory can tell us whether identification holds, it is often agnostic to the *degree* of failure. See Table 11 for a summary of results, Table 12 for a few additional performance metrics, and Figure 8 for a boxplot of ATE estimation errors over the different *SoI*.

We focus on two linear SCM settings:

- **Linear-No-Hidden:** Linear SCMs with 10-15 nodes and full observability (no hidden confounders), using 1000 data points per SCM.
- **Linear-60-Hidden:** Same setup as above, but with 60% of the variables unobserved (hidden).

We provide more details about the *SoI* used in our experiments in Table 10. Parameters not explicitly listed for a given *SoI* are set to their default values as per the benchmark configuration.

Table 10: Specification of the *SoIs* used to evaluate performance under hidden confounding. $N$ denotes the sampled number of nodes.

| **Name** | `Linear-No-Hidden` |
|---|---|
| # Nodes | 10-15 |
| Mechanism | Linear |
| Expected Edges | $2 \times N$ |
| Variable Type | Continuous |
| Prop. Hidden Nodes | 0% |
| Samples | 1000 |
| Query Type | ATE |
| Seeds | [42, 43, 44, 45, 46] |

| **Name** | `Linear-60-Hidden` |
|---|---|
| # Nodes | 10-15 |
| Mechanism | Linear |
| Expected Edges | $2 \times N$ |
| Variable Type | Continuous |
| Prop. Hidden Nodes | 60% |
| Samples | 1000 |
| Query Type | ATE |
| Seeds | [42, 43, 44, 45, 46] |

**Setup.** We evaluate three methods: CausalNF, DCM, and DeCaFlow. The first two methods assume causal sufficiency, and therefore cannot, in theory, handle hidden confounding. DeCaFlow, in contrast, is explicitly designed for this setting but requires access to the full causal graph (including hidden variables) and does not run when all variables are observed. Thus, we include it only in the hidden confounding *SoI*.

**Results (Linear-No-Hidden).** As expected, both CausalNF and DCM perform well when all variables are observed. DCM achieves lower mean error (0.0845) and standard deviation (0.1515), with a maximum error of 2.89. The upper whisker of DCM's box plot lies below the median of CausalNF, indicating consistent superior performance. These results serve as a reference point for comparison when introducing hidden variables.

**Results (Linear-60-Hidden).** With 60% of variables hidden, method performance degrades significantly. DeCaFlow performs reliably, with an error mean of 0.3405 and low variance. In contrast, CausalNF—despite a box plot that visually appears well-behaved—has a massive error mean of $2.67 \times 10^{12}$ and a maximum error exceeding $10^{15}$. This is due to a small subset of SCMs producing extremely large errors (14 with error $> 1000$), illustrating that, when assumptions are violated, error can become arbitrarily large. While DCM does not show such instability on this particular sample, its theoretical limitations under hidden confounding still hold — the expectation is that if we evaluate over enough SCMs we will eventually also get arbitrarily large errors due to the violation of the causal sufficiency assumption.

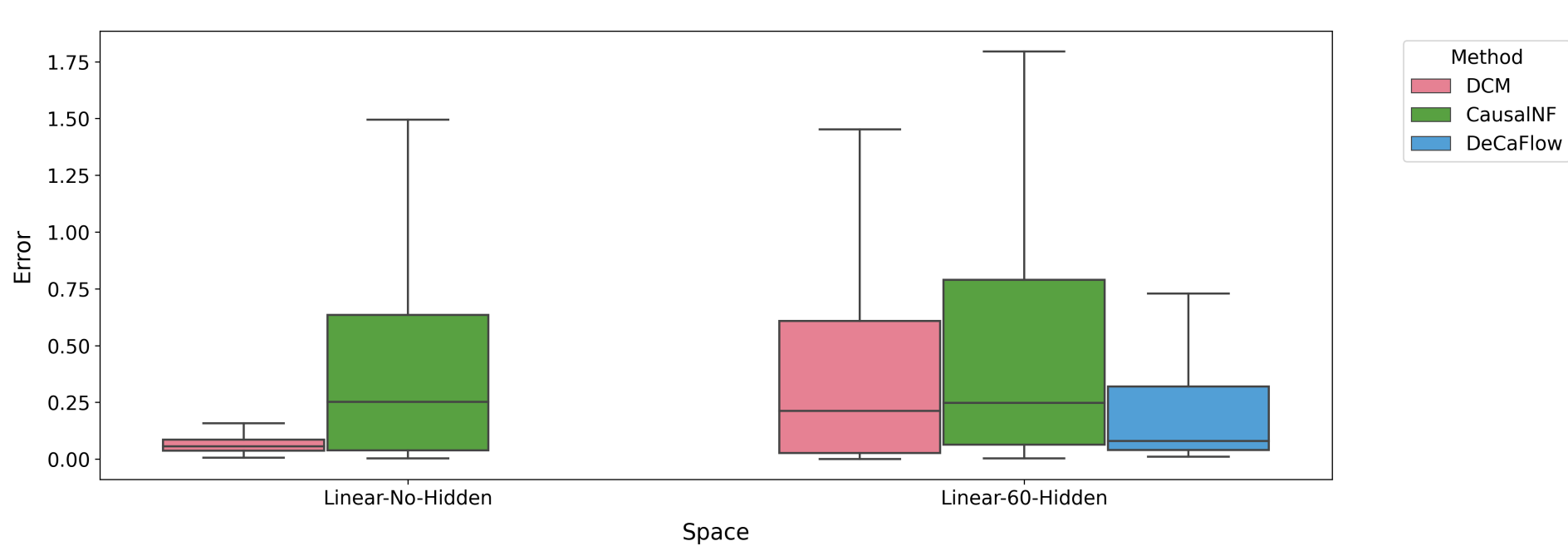

Figure 8: Box plots of ATE estimation errors in the presence and absence of hidden confounding. Each box shows the interquartile range and median, with whiskers extending to 1.5× IQR. CausalNF and DCM are shown for both *SoIs*; DeCaFlow is shown only for the hidden setting.

Table 11: Performance summary of CausalNF, DCM, and DeCaFlow on the hidden confounder experiments.

| Space | Method | Mean Error | Std Error | Max Error | Runtime (s) |
|---|---|---|---|---|---|
| `Linear-No-Hidden` | CausalNF | 0.5538 | 0.9866 | 14.2495 | 8570.0 |
| `Linear-No-Hidden` | DCM | 0.0845 | 0.1515 | 2.8954 | 12144.6 |
| `Linear-60-Hidden` | CausalNF | 2.667e+12 | 5.497e+13 | 1.225e+15 | 293.2 |
| `Linear-60-Hidden` | DCM | 0.5584 | 1.2122 | 17.2049 | 4187.6 |
| `Linear-60-Hidden` | DeCaFlow | 0.3405 | 0.6799 | 5.9435 | 2264.0 |

Table 12: Additional performance metrics of CausalNF, DCM, and DeCaFlow on the hidden confounder experiments.

| Space | Method | Min Error | Total Fail | Runtime Mean | Runtime Std |
|---|---|---|---|---|---|
| Linear-No-Hidden | CausalNF | 0.0036 | 0 | 17.14 s | 10.61 s |
| Linear-No-Hidden | DCM | 0.0068 | 0 | 24.29 s | 7.64 s |
| Linear-60-Hidden | CausalNF | 0.0029 | 0 | 0.59 s | 0.02 s |
| Linear-60-Hidden | DCM | 0.0000 | 0 | 8.38 s | 3.45 s |
| Linear-60-Hidden | DeCaFlow | 0.0108 | 0 | 4.53 s | 1.27 s |