# OpenReview forum: "CausalProfiler: Generating Synthetic Benchmarks for Rigorous and Transparent Evaluation of Causal ML"
_NeurIPS.cc/2025/Datasets_and_Benchmarks_Track — Submitted to NeurIPS 2025 Datasets and Benchmarks Track_

### Official Review · Reviewer_LZq3 · 2025-06-18

**Rating:** 5
**Confidence:** 4

**Summary:**

This paper introduces CausalProfiler, a benchmark for causal machine learning algorithms. CausalProfiler allows for the generation of synthetic causal datasets under a large variety of different settings and assumptions. This way, methods can be tested in different settings, and it can be systematically evaluated under which conditions they perform well. Some experimental evaluation is included, showcasing how CausalProfiler could be used to assess the strengths and weaknesses of causal machine learning methods.

**Additional Feedback:**

Questions

1. In lines 180-182, it is stated: "To ensure that queries are well defined and empirically grounded, we draw realizations from a large, separately sampled observational dataset, rather than from the theoretical variable domains." Could there not also be an argument made in favor of sampling from outside of the observational dataset? Not as a standard procedure, but to test certain generalization behavior, for example, for linear models.
2. It is mentioned that CausalProfiler can also be used to benchmark causal discovery methods. I am surprised that this is only mentioned in passing. I understand that benchmarking causal discovery is strictly simpler and does not require all components of CausalProfiler, but I think that causal discovery algorithms could also benefit from this benchmark a lot. My question regarding this is, therefore, if there are any problems, challenges, or limitations associated with using CausalProfiler for causal discovery that go beyond those already listed in Section 7?

Minor points

1. In line 91, "discover" should be "discovery"
2. In line 303, "rejection-based mechanism sampling" is mentioned. I believe this was not introduced up to this point. I know that there is a more detailed explanation in the appendix, but a short description in the main body before it is mentioned would be useful.
3. In Definition A.2, it says "SCM is n acyclis" and it should say "SCM is an acyclic"
4. In Algorithm 6, $\mathbf{V}_o$ is not introduced before being used in line 9.
5. Figures 2 to 6 are low resolution. A higher resolution or including them as vector graphics would be better.
6. In the appendix, "Figure" and "fig" are used inconsistently.

**Dataset Code Accessibility:**

Yes

**Dataset Code Comments:**

The code and instructions on how to use it are included in the supplementary material.

**Ethical Considerations:**

No, there are no or only very minor ethics concerns

**Final Justification:**

The authors addressed my concerns well, and I have no new concerns as a result of this discussion phase. Therefore, I maintain my original recommendation of acceptance.

**Limitations Weaknesses:**

The paper already states some limitations itself in Section 7. I do agree that these are limitations, but I will not repeat them here.

Two other weaknesses are:

1. This paper does not provide any meaningful comparisons between current causal machine learning methods for evaluating their general usefulness. The provided experiments in Sections 6.3 and 6.4 are better described as a demo on how to use the CausalProfiler benchmark than a thorough comparison of causal models.
2. It has been shown that the variance can be a great indicator for identifying causal graphs in simulated data [1]. This paper should be discussed. In addition, an option to standardize data would improve the benchmark.

[1] Reisach, Alexander, Christof Seiler, and Sebastian Weichwald. "Beware of the simulated dag! causal discovery benchmarks may be easy to game." *Advances in Neural Information Processing Systems* 34 (2021): 27772-27784.

**Strengths Contributions:**

1. Introducing a general and extensive benchmark for evaluating causal machine learning methods in a large variety of settings is an important and good contribution.
2. CausalProfiler supports a large number of meaningful parameters, including confounding, different noise distributions, and different queries (see Appendix B). It also supports a variety of metrics (Appendix F).
3. The description of the framework is explained very well, both in the text and using Definitions 4.1 and 5.1.
4. Full coverage for discrete variables (as shown by Theorem 5.1) is a valuable property of the data generation process.
5. An experimental verification of the correctness of the benchmark is included (Section 6.1).
6. Limitations are stated clearly and transparently.
7. The appendix is very extensive and describes the data and problem generation process of CausalProfiler in much detail.
8. The code in the supplementary material is documented very well.

---

> ### Author Rebuttal · Authors · 2025-07-31
>
> We thank the reviewer for the thoughtful and detailed feedback.
>
> **Weakness 1: Experiments are illustrative, not comprehensive**: We agree that a long-term goal of CausalProfiler is to enable large-scale, rigorous comparisons of Causal ML methods. This paper focuses on presenting the framework itself—its design, theoretical guarantees (e.g., coverage), core abstractions (e.g., Space of Interest), and sampling pipeline—and showing how it can surface method strengths and weaknesses under varying assumptions. A comprehensive empirical study would constitute a separate contribution and is left for future work.
>
> **Weakness 2: Lack of discussion of variance-based causal graph benchmarks (e.g., Reisach et al.)**: We agree this is a valuable and complementary line of work. On the one hand, for regional discrete SCMs, there is no reason to expect variability to increase along the causal order—although we acknowledge that we have not empirically tested this, and agree it is worth investigating. On the other hand, for continuous SCMs, users may choose to work with additive noise models, which fall within the scope of Reisach et al.'s findings.
> We therefore agree that offering data standardization is a useful addition, and we will include it as a user-configurable input parameter in CausalProfiler.
> We thank the reviewer for raising this point.
>
> **Feedback 1: Could sampling outside the observational dataset improve generalization evaluation?**
>
> We would like to clarify that the dataset from which we sample values of the query variables is distinct from the dataset provided to the user for training which is typically ten times smaller (default is 10k for query variables values and 1k for user training). The reason is that, given the randomness of the causal mechanisms, we cannot know in advance the domain over which the SCM is defined. Even when variable cardinalities are fixed, the sampled mechanisms may be non-surjective, making certain values impossible to observe. For this reason, we approximate the domain of definition through data sampling, ensuring that queries are computed only for realizable variable configurations. Moreover, since the dataset given to the user is smaller, it is possible that queries use values outside of the observational dataset or that they are non-identifiable (this can be configured if not desirable, but this is the default behavior). For more discussion on this, check Weakness 5 of Reviewer c5Yz.
>
> That said, we agree that explicitly enabling queries to be outside the observed dataset can be useful for studying generalization—especially in settings where the support is known, such as linear SCMs. We can expose this as a user-configurable option in SoIs, even allowing users to define a custom domain for the query variables. Since this is an open-source framework, we expect it to evolve based on the needs of the community.
>
> **Feedback 2: Why is causal discovery only briefly mentioned?**
>
> While there already exist SCM generators for causal discovery tasks (e.g., [8,9]), they typically target continuous-variable settings and do not have coverage guarantees. CausalProfiler supports both discrete and continuous SCMs, offers a broader range of structural and functional variability, and includes a formal coverage guarantee over the defined SoI.
>
> However, our experiments focus primarily on interventional and counterfactual estimation, as our main contribution lies in generating synthetic SCMs with ground-truth values for such queries.
>
> That said, CausalProfiler can already be used to evaluate causal discovery methods, since each sampled dataset includes the ground-truth causal graph, enabling comparison via standard metrics (e.g., SHD, F1). Users can also disable query sampling to generate datasets specifically for discovery tasks more efficiently.
>
> We thank the reviewer for highlighting this, and we will make this capability clearer in the paper.
>
> **Minor points:** We thank the reviewer for highlighting the typos and lack of clarity; we will correct them.
>
> We hope our explanations helped ease some of your concerns. Please let us know if you have any additional questions. Thank you!
>
> [8] Diviyan Kalainathan, Olivier Goudet, and Ritik Dutta. Causal discovery toolbox: Uncovering causal relationships in python. Journal of Machine Learning Research, 21(37):1–5, 2020
>
> [9] Shantanu Gupta, Cheng Zhang, and Agrin Hilmkil. Learned causal method prediction. arXiv:2311.03989, 2023

---

> > ### Comment · Reviewer_LZq3 · 2025-08-05
> >
> > I thank the authors for their detailed response. I agree that weakness 1 can be left for future work and appreciate the changes made in response to weakness 2. Thus, I have no major concerns and continue to recommend acceptance.

---

### Official Review · Reviewer_Mc7P · 2025-06-20

**Rating:** 4
**Confidence:** 4

**Summary:**

This paper introduces a framework for generating synthetic causal inference benchmarks. The idea is to specify the class of causal models, queries, and data regimes to evaluate, and then randomly sample many datasets and causal questions from that space. It includes experiments demonstrating that the framework can differentiate algorithm performance across scenarios.

**Additional Feedback:**

- Pg. 3 - typo: “causal discover”
- Pg. 4 - does causalProfiler support binary or categorical treatments?
- Pg. 10 - de Vassimon Manela et al. 2024 is entered twice in the reference list

**Dataset Code Accessibility:**

Yes

**Dataset Code Comments:**

The authors have provided their code in the supplementary material and have committed to releasing it publicly as a GitHub repository after the review period. Since the framework generates data on command, no external dataset needs to be downloaded for replication.

**Ethical Comments:**

The submission does not raise ethical concerns. The work deals with synthetic data generation and benchmarking for methodological evaluation.

**Ethical Considerations:**

No, there are no or only very minor ethics concerns

**Final Justification:**

The paper is technically solid but still lacks evidence that its insights will generalize to real-data settings. The original rating (4) stands.

Resolved:
- Question of whether CausalProfiler support binary / categorical treatments. Authors clarified that discrete SCMs can be generated with any user‑specified cardinality.
-  Typos / duplicate references. Authors agreed to fix.

Remaining concerns:
- External‑validity gap: No performance comparison on real or semi‑synthetic datasets; authors’ t‑SNE analysis is helpful structurally but does not establish whether the synthetic rankings generalize. Authors agree to add to discussion.
- Scope limitations: Current implementation covers a narrow set of query types and data modalities; extensions are future work. Authors acknowledged the issue and agree to include it in the "Limitations" discussion.

**Limitations Weaknesses:**

- All evaluation data in this work are fully synthetic, and there is no demonstration that these results translate to real-world datasets. The paper does not include any experiments on real or semi-synthetic benchmarks, nor attempts to correlate synthetic performance with real-world performance. Empirical realism is questionable: the synthetic SCMs, by design, obey the authors’ specified assumptions, but real-world causal systems may violate those in unpredictable ways; e.g., unmodeled biases. The paper itself notes that handcrafted or narrow synthetic datasets often fail to generalize beyond their original study context. The broader usefulness of CausalProfiler would be strengthened by evidence that it can model scenarios reflective of real problems. As a weakness, the current evaluation lacks external validation — the authors could improve this by, for example, applying their benchmark to methods on an existing real-world causal inference task or by showing that insights gained synthetically — e.g. a method’s sensitivity to confounding — also appear with real data.
- Related to the point above — the paper discusses common benchmarks like IHDP and Twins in background but does not actually show how the new synthetic evaluation compares to those established datasets. The absence of these comparisons is a missed opportunity: it would have been informative to see, for example, if a method that performs best on IHDP also excels across a broader space of interest, or if CausalProfiler reveals weaknesses that a static dataset might hide.
- The current implementation of CausalProfiler has some restrictions in scope that should be acknowledged. It appears focused on a certain type of causal estimation; for example, treatment effect estimation in tabular data. It does not yet cover other important causal query types (e.g., time-varying causal effects) or different data modalities (such as time-series, text, or image data).

**Strengths Contributions:**

- The work addresses a clear gap in causal ML evaluation. The framework provides a novel random benchmark generator with broad coverage. The notion of specifying an evaluation space replaces ad hoc choice of a single dataset with a systematic way to explore many scenarios. This is an important step toward more generalizable and reliable evaluation of causal inference methods.
- The framework makes evaluation more transparent and repeatable. All modeling assumptions and data-generating conditions are explicitly defined via the space parameters. There’s a “coverage guarantee”  that the sampled tasks cover the intended range of conditions, and the authors include an empirical distribution analysis to quantify the diversity of generated data.
- The work conducts comprehensive experiments with multiple state-of-the-art causal ML methods, evaluating them across diverse spaces. The evaluation results are interesting: for example, a diffusion-based method excelled in linear scenarios, whereas a variational autoencoder method performed best with complex nonlinear SCMs. These findings both demonstrate the utility of the framework, and also uncover how each algorithm’s performance shifts under different assumptions.

---

> ### Author Rebuttal · Authors · 2025-07-31
>
> We sincerely thank the reviewer for their thoughtful and constructive review.
>
> **Weakness 1: No experiments on real or semi-synthetic datasets; synthetic-to-real generalization is unclear.**
>
> We agree that understanding generalization from synthetic to real data is a critical direction for future work. In the current paper, we focus on building a foundation for rigorous synthetic evaluation. While CausalProfiler currently relies on manually specified SoIs, our vision is to enable future pipelines that learn SoIs from real data. For instance, SoIs could be constructed using Bayesian methods that infer distributions over SCMs, graphs, or mechanisms from observational data. Currently, the only option is to manually create a SoI using domain knowledge (e.g., known motifs or structural assumptions).
>
> This line of work is promising and necessary to connect synthetic benchmarks to real-world problems. However, mapping from observational data to SCM distributions is fundamentally underconstrained, and we believe such inference must be done carefully—especially given challenges around identifiability and inductive bias.
> We will add a discussion on this in the Discussion section.
>
> **Weakness 2: Lack of comparison with established datasets**
>
> We agree that the paper currently lacks a direct comparison of CausalProfiler's synthetic SCMs with established datasets. For this reason, we ran additional experiments to illustrate how CausalProfiler contributes to SCM diversity for practitioners wishing to evaluate Causal ML methods. Specifically, we compared the sampled SCMs from Appendix G to some existing datasets. Below, we present the results of these additional experiments without providing new figures as per the rebuttal format. We will add these results to the paper and make them reproducible through code in the repository.
>
> For counterfactual tasks, there are no real or semi-synthetic established datasets available. Hence, we decided to compare our SCMs to those from the CausalNF paper because they have been reused by other works [4,5] as de facto benchmarks. To do so, we reimplemented the synthetic SCMs of CausalNF using our package and evaluated them using the full set of metrics from the Assumptions analysis module (Appendix F), applying the same procedure used for our own SCMs.
> We then used these metrics to compare the two groups of SCMs.
> To ensure fairness and avoid penalizing CausalNF for having made design assumptions, we excluded certain metrics—namely, those related to hidden confounding and positivity. This is because all CausalNF SCMs satisfy causal sufficiency and strong positivity by construction, while our SCMs do not necessarily, by design. To visualize the comparison, we projected all SCMs into a two-dimensional space using t-SNE [6] with perplexity 30, based on the retained metrics.
> Results show, as expected, that our SCMs are more diverse overall. Regarding graph metrics, the CausalNF SCMs already have good diversity. The fact that we have greater support could also stem from the fact that we sampled a larger number of SCMs. However, for distributional and mechanism-related metrics, the increase in diversity is clear: the CausalNF SCMs are clustered into a small region of the space, while ours span a much wider area. For instance, the total scale of variability across metrics is approximately 9 times bigger in our SCMs compared to CausalNF ones.
> In addition, we can see that some of the CausalNF SCMs and our SCMs overlap or are very close on the t-SNE plots, showing that CausalProfiler is able to reproduce similar SCMs.
> As a result, we conclude that CausalProfiler can enable practitioners to evaluate CausalML methods on similar but also more diverse sets of SCMs and allow for richer conclusions.
>
> For interventional tasks, there exist several real and semi-synthetic datasets, such as IHDP and Twins. However, for this analysis, we chose the CANCER and EARTHQUAKE datasets from the bnlearn package [7] because they include ground-truth causal graphs and are directly comparable to the SCMs we already analyzed in Appendix G (i.e., they have a similar number of variables). In contrast, IHDP and Twins have over 20 variables, and analyzing SCMs of that scale would require us to generate and process new SCMs.
> As with the previous comparison, we applied two-dimensional t-SNE projections over the graph and distribution metrics (mechanism metrics could not be included, as they are unavailable for these datasets). For the sake of having a fair comparison, we also excluded the hidden confounders metrics (as both bnlearn graphs are DAGs) but kept the positivity metrics for this analysis.
> The results show that the two bnlearn datasets are not confined to a small region of the two-dimensional space. Instead, they fall within the bottom left region of the t-SNE plot, overlapping with some of our generated SCMs. Hence, the conclusion of this analysis is similar to the previous one: CausalProfiler can generate SCMs producing similar causal datasets to established ones while also generating more diverse sets of SCMs.
>
> **Weakness 3: No support for time-varying effects or alternative modalities** We agree that supporting time-varying effects and richer modalities is a valuable and exciting direction for future work. We will explicitly mention this in the revised version of our ``Limitations and Future Work'' section. We emphasize that the CausalProfiler will be released as an open-source framework and, like most community tools, is designed to evolve. We hope to prioritize these extensions based on community interest.
>
> **Feedback 1, 2, 3:** We thank the reviewer for highlighting the typos and lack of clarity—we will correct them.
>
> We hope our explanations are clear, and they could make you reassess the evaluation of our work. Please let us know if you have any additional questions or concerns.
>
> [4] Beate Sick, and Oliver Dürr. Interpretable Neural Causal Models with TRAM-DAGs. Conference on Causal Learning and Reasoning, 2025
>
> [5] Qingyang Zhou, Kangjie Lu, and Meng Xu. Causally Consistent Normalizing Flow. Proceedings of the AAAI Conference on Artificial Intelligence, 2025
>
> [6] Laurens van der Maaten, and Geoffrey Hinton. Visualizing data using t-SNE. Journal of machine learning research, 9:2579-2605, 2008
>
> [7] Marco Scutari. Package ‘bnlearn’. Bayesian network structure learning, parameter learning and inference, R package version. 2019, https://www.bnlearn.com/

---

> > ### Comment · Reviewer_Mc7P · 2025-08-01
> >
> > Thank you for the detailed response. To clarify a few of the points:
> >
> > - Does CausalProfiler currently support binary and categorical treatment variables? If so, how are they handled in the SoI and mechanism generation?
> > - Were any causal‐effect estimation methods run on the CANCER and EARTHQUAKE datasets, and if so, how their rankings compared with those observed on CausalProfiler‑generated data? If no such run was performed, what would be required to enable a direct performance comparison? Do you foresee including this type of cross‑dataset performance validation in a revision?

---

> > > ### Author Response · Authors · 2025-08-02
> > >
> > > We thank the reviewer for the interest and engagement. We try to answer the clarification questions below:
> > >
> > > **Q1. Binary and Categorical Treatments**
> > >
> > > Yes, CausalProfiler supports both binary and categorical treatment variables. When defining an SoI, users must first choose between continuous and discrete SCMs. For discrete SCMs, the variable cardinality is controlled via the *number_of_categories* field, which can be a fixed integer or a range to sample from. Since any variable can be selected as a treatment during query generation, treatment variables inherit their cardinality from this setting. We note, however, that if users want to focus on specific treatment and outcome variables, these can be explicitly specified in the SoI using the *specific_query* parameter.
> > >
> > > During mechanism generation (done before query sampling, so before any variable is chosen as treatment or outcome), categorical variables
> > > are handled using Regional Discrete Mechanisms (Appendix D). Each variable's exogenous noise space is partitioned into regions, with each region defining a tabular mapping from parent configurations to output values. These mappings are sampled using one of three strategies: exhaustive partition (the most general, providing coverage guarantees), sample rejection (Algorithm 5), or unbiased random assignment. Appendix D has more details on how this works.
> > >
> > > Only after the whole SCM (i.e., endogenous and exogenous variables, causal graph, and mechanisms) is sampled from the SoI, queries are generated with the possibility of having any variable as treatment and outcome (unless specified otherwise).
> > >
> > >
> > >
> > >
> > >
> > > **Q2. Cross-Dataset Validation**
> > >
> > > While we included the CANCER and EARTHQUAKE datasets in our structural and distributional analyses, we did not run causal effect estimation methods on them, and therefore didn't compare method rankings with those observed on CausalProfiler-generated SCMs.
> > >
> > > Such experiments are feasible, but doing so meaningfully would require constructing SoIs that accurately reflect the properties of the datasets. While our structural comparisons showed that CausalProfiler can generate similar SCMs, this does not imply that the broader SoI distribution over the dataset properties matches that of the original datasets.
> > >
> > > We believe that meaningful cross-dataset performance comparison requires bridging the gap between real data and the SoIs used in CausalProfiler. As discussed in our response to Reviewer c5Yz (Feedback 5), the key challenge is developing methods to map from real data to SoIs—so that synthetic SCMs reflect the structural and distributional characteristics of real domains.
> > >
> > > This would make cross-dataset validation of causal estimators far more grounded and systematic. While this is beyond the scope of the current paper, it is a direction we are actively exploring. In the meantime, a rough comparison is possible by manually constructing an SoI that approximates the target dataset—similar in spirit to an expert-specified SCM—but this remains an approximation.

---

### Official Review · Reviewer_c5Yz · 2025-06-30

**Rating:** 4
**Confidence:** 4

**Summary:**

This paper highlights the limitations of current evaluation methods in Causal Machine Learning, which often rely on a small number of handcrafted or semi-synthetic datasets. The authors introduce CausalProfiler, an innovative randomized benchmark generation framework that supports evaluation across all three levels of causal reasoning: observational, interventional, and counterfactual. Built on clearly defined causal models, the framework systematically generates synthetic datasets along with their true structures and modeling assumptions, significantly improving the rigor and reproducibility of evaluations.

**Additional Feedback:**

In addition to the issues mentioned in the Limitations Weaknesses* section, the other problems are as follows:

1. Can users weight or stratify the SoI to obtain approximately uniform coverage over selected graph or mechanism attributes, rather than relying on raw random draws?

2. Beyond the 30-variable range tested, how does runtime scale, and can  CausalProfiler generate datasets with 100+ variables in a acceptable time?

3. Since practical causal learning often relies on a blend of observational and limited interventional data, do you plan to allow SoIs that generate observed do-samples alongside L1 data?

4. Have you benchmarked generated distributions (marginal moments, causal graphs, effect sizes) against any real datasets to calibrate how plausible SoIs are?

5. Will you provide templates or heuristics that help practitioners map a real application domaininto an SoI, reducing the risk of unrealistic settings?

6. Have you compared method rankings obtained with CausalProfiler against those on long-standing semi-synthetic sets such as IHDP/Twins? If divergences occur, how should practitioners calibrate trust?

**Dataset Code Accessibility:**

Partly

**Ethical Considerations:**

No, there are no or only very minor ethics concerns

**Final Justification:**

After reviewing the authors' response, I vote for a 'Borderline Accept' rating.

**Limitations Weaknesses:**

1. Although the coverage theorem is mathematically strong, rare but important regimes (e.g., sparse linear identifiable graphs) may appear with negligibly small probability, requiring large draws or hand-crafted SoIs to study them meaningfully.

2. The coverage guarantee does not extend to continuous-variable SCMs; the authors note that finite-width neural networks cannot represent every continuous function, only approximate them.

3. Because graphs, functional forms, and noise distributions are fully synthetic, generated datasets may deviate from domain-specific constraints (e.g., epidemiological sparsity patterns) unless users carefully encode them in the SoI.

4. The open-source release can currently generate only observational data and support ATE, CATE and CTF-TE queries; no interventional training sets or discovery-style tasks are yet automated.

5. Queries that produce duplicates or NaNs are rejected and resampled , potentially filtering out precisely the hard instances that expose method weaknesses.

**Strengths Contributions:**

CausalProfiler marks an important step forward in evaluating causal machine learning methods. It is the first open-source benchmark generator in the field that provides clear guarantees on coverage and fully transparent assumptions.

1. This paper offers a rigorous, theory-based framework to create customizable benchmark datasets by sampling user-defined Spaces of Interest, helping to address reproducibility issues in causal ML research;

2. This paper introduces a thorough evaluation approach that highlights the limitations of relying on a single dataset by showing how testing across multiple synthetic scenarios leads to more generalizable insights into model performance. The framework’s value is demonstrated through detailed experiments revealing significant differences in method effectiveness across various identification settings. By combining solid theory, practical flexibility, and careful experimentation, CausalProfiler sets a new standard for reliable, assumption-aware validation in causal machine learning.

---

> ### Author Rebuttal · Authors · 2025-07-31
>
> We thank the reviewer for the detailed feedback. Below we respond to some comments.
>
> **Weakness 1: Rare SCM regimes may be unlikely without targeted SoIs**:
> We fully agree. For additional discussion, we gently refer the reviewer to our response to Weakness 4 from Reviewer 8e2f, which includes further clarifications of Appendix G.
>
> **Weakness 2: No coverage guarantee for continuous-variable SCMs**
>
> This is correct. The formal coverage guarantee (Theorem 5.1) holds only for Regional Discrete SCMs. In Appendix I, we offer a remark on why extending this guarantee to continuous-variable SCMs is nontrivial. While neural networks are dense in the space of continuous functions, this density does not imply strict coverage with finite capacity models. In particular, any fixed architecture with finite depth and width cannot represent all continuous mechanisms exactly. Extending the guarantee to notions like density rather than coverage remains an open question.
>
> Moreover, CausalProfiler is an open-source, modular framework designed for ongoing extension. We warmly welcome future contributions that explore other theoretical guarantees (e.g., sampling over specific function classes). We see this as a long-term, community-driven effort to advance rigorous empirical practices in Causal ML.
>
> **Weakness 3: Domain-specific constraints must be manually encoded**
>
> We agree, by design. Our goal is to make all assumptions explicit and user-defined, avoiding the "hidden alignment" that can arise from pre-baked implicit choices (as seen in some semi-synthetic datasets). Rather than automating constraints that may introduce untracked biases, we allow users to explicitly encode them via the SoI specification. We reply in Feedback 5 on mapping real domains into SoIs.
>
> **Weakness 4: Current version supports only observational data and limited queries**
>
> This is a fair point. The current version supports observational training data and a minimal but representative set of queries (ATE, CATE, CTF-TE). Our goal was to build a general-purpose, extensible framework that operates at all three levels of the Pearl Causal Hierarchy.
>
> We put significant effort into designing an accessible, well-tested codebase with end-to-end and unit tests to ensure stability as new features are added. Upon release, we will provide a detailed contribution guide, a public roadmap, and a voting system to help prioritize community-requested features. We emphasize that an open-source framework like CausalProfiler is inherently ongoing work—never truly ``complete"—and we hope it evolves with the needs and insights of the community.
>
> **Weakness 5: Rejection of NaN/duplicate queries may filter out hard instances**
>
> Thank you for raising this. We clarify that query ground truths are computed using a large number of samples (typically 10k), whereas users receive only a subset (e.g., 1k). Thus, queries can still be challenging to estimate with limited data. Queries are rejected only if they are undefined (e.g., conditioning on a zero-probability event), concerning the larger dataset CausalProfiler samples.
>
> We acknowledge that, in some settings, retaining both duplicates and all the NaN queries can be useful. We can make this behavior configurable as a parameter of CausalProfiler (e.g., via allow_nan and allow_duplicate flags and an if-statement at sampler.py::_sample_and_evaluate_queries.py::555) and add documentation around this.
> This is easily implementable and not a fundamental limitation. We chose to default to diverse, well-posed queries to ensure fair evaluation, but are happy to support more lenient modes as requested by the community.
>
> **Feedback 1: Can SoIs be stratified to ensure uniform coverage over structural attributes?**
>
> Stratification over structural attributes (e.g., number of variables, edge density) or any other SoI input parameters is not directly supported in the current version. However, this is conceptually feasible. Users can already define multiple SoIs targeting different regimes and aggregate results post hoc to approximate uniform coverage. Additionally, if stratification is desired after sampling, users may apply post-stratification or reweighting techniques to adjust for over- or under-representation in the sampled datasets.
>
> **Feedback 2: How does runtime scale beyond 30-variable graphs?**
>
> We use batch processing and vectorized operations to handle multiple samples in parallel efficiently. As a result, the framework scales well with the number of variables.
> Below, we report the average generation time (over 5 runs) for producing 10k samples and 50 queries (each estimated using 10k separate datapoints), using the same CPU hardware reported in the paper:
> | Num Variables | Mean Time (s) | Std Dev (s) |
> |---------------|---------------|-------------|
> | 10            | 0.19          | 0.01        |
> | 50            | 0.89          | 0.03        |
> | 100           | 1.81          | 0.03        |
> | 500           | 9.61          | 0.11        |
> | 1000          | 19.24         | 0.21        |
>
> In case it's of interest, we also report the runtime (in seconds) of each method in Experiment 1 as the number of nodes increases on the NN-Large setting, with the expected number of edges fixed to N. Some methods scale better than others. Each cell shows the mean and standard deviation across runs in the format (mean, std).
> | Node Range    | CausalNF     | DCM          | NCM          | VACA         |
> |---------------|--------------|--------------|--------------|--------------|
> | 30–40         | (1, 0.6)     | (24, 8.4)    | (12, 1.2)    | (11, 4.6)    |
> | 50–70         | (2, 0.4)     | (43, 12.8)   | (22, 2.6)    | (12, 4.8)    |
> | 70–90         | (3, 0.3)     | (53, 18)     | (29, 2.4)    | (12, 4.7)    |
> | 90–110        | (4, 0.4)     | (60, 23)     | (36, 2.5)    | (9, 2.5)     |
>
> Note: VACA's failure rate increases with node range: 12% → 40% → 58% → 74%, while all other methods have 0\% failure rate here.
>
> We will add these tables to the appendix.
>
> **Feedback 3: Will interventional data generation be added?** Yes. The simulator already supports interventional sampling, and we plan to extend the data generation pipeline to allow training on arbitrary interventional distributions. We refer the reviewer to our response to Weakness 4 for more context on extensibility and roadmap plans.
>
> **Feedback 4: Comparability to real datasets** We kindly refer the reviewer to our response to Reviewer Mc7P (Weakness 2).
>
> **Feedback 5: Mapping real domains into SoIs**
>
> This is an interesting direction and, in our view, the key missing piece for enabling fair comparisons between CausalProfiler and semi-synthetic methods. While the current framework relies on manually specified SoIs, we envision several ways to bridge the gap to real domains. In the near term, users can construct SoIs guided by domain expertise or empirical features—for example, using known graph motifs or structural assumptions.
>
> Longer term, we see potential in Bayesian approaches that infer distributions over SoIs directly from observational data—e.g., mixtures over graph structures, mechanism classes, or noise types. This could support principled semi-synthetic evaluation pipelines where SoIs are shaped by empirical evidence, rather than fixed assumptions. However, mapping from observational data to SCMs is a fundamentally underconstrained problem, and any such inference would need to be handled with care. Questions about realism, identifiability, and inductive bias will naturally arise. While these ideas are outside the scope of our current work, we believe they are highly promising and complementary to our proposed framework.
>
> **Feedback 6: Rankings of Experiment 1 vs Rankings using IHDP/Twins**
>
> We assume the reviewer is referring to the rankings of Experiment 1 as IHDP, and Twins are usually used to evaluate ATE estimations. Unfortunately, we haven't performed such a comparative evaluation.
>
> First, none of the methods we evaluate in Experiment 1 have been run on a common real-world or semi-synthetic dataset, making meaningful comparisons difficult without substantial additional effort.
>
> More importantly, CausalProfiler is not currently intended to replicate the structure of any specific real or semi-synthetic dataset. Its goal is to enable rigorous evaluation across explicitly defined SoIs. That said, one could design a domain-specific SoI informed by expert knowledge or empirical analysis of a real dataset like IHDP or Twins.
>
> We agree that exploring how CausalProfiler rankings align (or diverge) from those observed on real datasets is an interesting direction for future work, especially when an SoI is explicitly constructed to approximate a given domain (maybe with some of the ideas mentioned in the previous feedback).
>
> We hope our explanations are clear, and they could make you reassess the evaluation of our work. Please let us know if you have any additional questions or concerns.

---

> > ### Comment · Reviewer_c5Yz · 2025-08-01
> >
> > Thank you for the author's reply, which has basically addressed and explained the weaknesses. Regarding the questions:
> >
> > 1. Could you elaborate further on Question 1 in more detail?
> >
> > 2. For Question 2, could you analyze the complexity to illustrate the impact of the number of variables and samples?
> >
> > 3. For Question 4, has the author conducted any preliminary experiments or analyses?

---

> > > ### Author Response · Authors · 2025-08-02
> > >
> > > We thank the reviewer for the interest and engagement.
> > >
> > > **Q1. Stratification**
> > >
> > > CausalProfiler does not currently support internal stratified sampling over structural attributes (e.g., enforcing uniform edge density). Uniform sampling over the number of variables is supported.
> > >
> > > However, stratification can be approximated externally by defining multiple SoIs targeting different regimes. For example, suppose a user wants to generate 300 SCMs with a target distribution over edge densities: 25% with X edges, 25% with Y, and 50% with Z. The user can define 3 SoIs with different values of this parameter and sample from each in the desired proportions (e.g., 75, 75, and 150 SCMs). This enables **external stratification** through multiple CausalProfiler instances and post hoc aggregation.
> > >
> > > This differs from **internal stratified sampling**, where quotas would be enforced within a single sampling pass. Such a feature would require extending the DAG sampler to condition on global properties—something that may be feasible through rejection sampling or constrained graph generation, but is not currently implemented.
> > >
> > > Note: external stratification only works if the stratified attribute is directly controllable via SoI parameters. Emergent properties (e.g., graph depth) cannot be enforced without redesigning the sampling algorithm.
> > >
> > > **Q2. Complexity analysis**
> > >
> > > While the number of variables and samples are main components of complexity, many other factors (i.e., SoI inputs) matter. Here, we provide an initial analysis based on a setting consistent with the experimental setup of our initial answer (and Exp. 1). Assume a continuous SCM where each mechanism is modeled as a 2-layer NN (with hidden size 8), variable dimensionality is 1, and the queries are ATE. Note that in discrete SCMs, complexity becomes harder to characterize due to dependencies on the choice of sampling algorithm and the number of noise regions.
> > >
> > > Parameters: $V$: Number of variables, $E$: Expected number of edges per variable, $N$: Number of samples, $Q$: Number of queries
> > >
> > > **Time Complexity:**
> > >
> > > * Graph generation: $O(V^2)$
> > > * NN initialization (per variable): $O(E)$
> > > * NN inference (per variable): $O(E)$
> > > * Sample generation:
> > >   * For each of the $N$ samples, we:
> > >     * Sample noise
> > >     * Run a topological sort (once)
> > >     * Evaluate each of the $V$ variables via a forward pass through a NN with on average $E$ inputs → cost per sample: $O(V \cdot E)$
> > >   * → Total: $O(N \cdot V \cdot E)$
> > > * Query generation and evaluation:
> > >   * $O(Q \cdot N \cdot V \cdot E)$,
> > > * Overall dominant term (worst case):
> > >   $O(Q \cdot N \cdot V \cdot E)$
> > >
> > > This scaling is intuitive: each query requires $N$ samples, where each sample involves computing all $V$ variables, and each variable depends on approximately $E$ parents through a NN.
> > > We get constant-time speedups using vectorized operations and batch processing, e.g., processing samples in batches instead of looping over them.
> > >
> > > **Space Complexity:**
> > >
> > > * Graph structure: $O(V + E)$
> > > * NN parameters: $O(V \cdot E)$
> > > * Sample storage: $O(N \cdot V)$
> > > * Query outputs: $O(Q)$, working memory to compute a single query: $Q(V \cdot N)$
> > > * Total: $O(V \cdot E + N \cdot V + Q)$
> > >
> > > **Q3. Comparability to real datasets**
> > >
> > > We have not performed direct comparisons between our SCMs and real datasets, due to the challenges of mapping real-world data to SoIs (as noted in our response to your initial Question 5).
> > > However, we have conducted preliminary experiments comparing to established datasets used in research (not real observational data).
> > >
> > > In the interest of space, we redirect to our response to Weakness 2 of Reviewer Mc7P ("Lack of comparison with established datasets") for full details. Specifically, we used the SCMs from Appendix G and compared the structural and distributional ($\mathcal{L}_1$) properties with some established datasets. While we did not compare causal effects, this is feasible in principle by manually constructing an SoI to approximate a target dataset, akin to specifying an expert-designed SCM.
> > > Briefly:
> > >
> > > 1. Interventional tasks: We compared to the CANCER and EARTHQUAKE datasets, which include ground-truth causal graphs. Using t-SNE projections over shared metrics, we found that both datasets overlap with a subset of our SCMs—showing that CausalProfiler can replicate their structural and distributional properties while covering a broader space.
> > >
> > > 2. Counterfactual tasks: We compared to CausalNF's synthetic SCMs used for evaluation (also used as benchmark by other works) again using t-SNE. Our SCMs cover similar cases while offering significantly greater diversity in mechanisms and distributions.
> > >
> > > More broadly, this work aims to move the field away from narrow evaluations on a small number of SCMs and toward structured and transparent, distribution-level assessment across well-characterized SoIs. While mimicking real datasets is an excellent next step, this paper focuses on building the foundations for rigorous synthetic evaluation.

---

> > > > ### Comment · Reviewer_c5Yz · 2025-08-05
> > > >
> > > > Thank you for your reply.

---

### Official Review · Reviewer_8e2f · 2025-07-02

**Rating:** 4
**Confidence:** 2

**Summary:**

This paper proposes CausalProfiler, a synthetic generator designed to address the limitations in current causal ML evaluation, which usually rely on a small number of hand-crafted datasets, leading to non-generalizable conclusions. The authors introduce the "Space of Interest" (SOI) concept that allows users to define a class of Structural Causal Models (SCMs), causal queries, and data rather than using a fixed dataset. This shifts the evaluation paradigm toward a more principled and repeatable assessment of method performance under diverse and controllable conditions. Through demonstrations on state-of-the-art methods like CausalNF, DCM, NCM, and VACA, the authors show that performance and rankings vary across different SoIs, demonstrating the importance of this more rigorous evaluation paradigm.

**Additional Feedback:**

1. The experiments consistently use 1000 samples across most SoIs. It would be better if the authors provide justification for this choice.

2. Table 1 shows dramatically different error magnitudes between linear SCMs and neural network SCMs: much smaller errors for NN-based SCMs. I think the authors should explain the results as NN-based SCMs are usually more complex.

**Dataset Code Accessibility:**

Yes

**Dataset Code Comments:**

The code is provided in the supplementary material. Although I did not run the code, the repo is well-documented with detailed instructions.

**Ethical Considerations:**

No, there are no or only very minor ethics concerns

**Final Justification:**

The authors have addressed most of my concerns during the discussion period, but I still find no concrete evidence where systematic approach provides unique value over targeted evaluation. Therefore, I increased my score to 4, but less certain to recommend acceptance.

**Limitations Weaknesses:**

1. The authors argue against relying on a "handful of hand-crafted or semi-synthetic datasets", but they did not demonstrate why researchers cannot simply create targeted SCMs for their specific needs without the generator proposed. The authors should strengthen their motivation with concrete examples where ad-hoc synthetic evaluation fails but the SoI-based approach succeeds.

2. The bottleneck of the causal ML evaluation is not a lack of synthetic data generation capabilities, but the scarcity of real-world datasets with ground truth. Creating more sophisticated synthetic data does not solve this fundamental simulations-to-real gap if we do not consider realistic simulators.

3. Theorem 5.1 seems to provide minimal theoretical insight. The "coverage guarantee" essentially states that their sampling procedure can generate anything in the space they defined, which should be true by construction for any reasonable framework. Another issue is the generator's unaddressed distributional bias. The authors admit that "certain classes of SCMs remain very unlikely to be sampled" and the analysis in Appendix G shows that the default sampling is highly skewed, producing SCMs that mostly violate positivity and are rarely highly confounded.

**Strengths Contributions:**

1. The SoI abstraction is the key contribution of this paper, providing a principled way to specify evaluation domains rather than ad hoc dataset selection.

2. The author conduct rigorous verification experiments across Pearl's hierarchy to ensure the generator's correctness (i.e., testing its output against the Markov property (L1), the rules of do-calculus (L2), and the axioms of structural counterfactuals (L3)).

---

> ### Author Rebuttal · Authors · 2025-07-31
>
> We thank the reviewer for their constructive feedback and highly appreciate the careful reading of our submission.
>
> **Weakness 1: Why not hand-crafted SCMs? Why is SoI better?**
>
> While hand-crafted SCMs are valuable for illustrating theoretical results and for controlled testing, our concern lies in their typical use for empirical evaluation, which can lead to overinterpretation—for example, assuming that a method's performance generalizes to other SCMs.
>
> We point to an example from [1] (Section 3.2, second experiment), where two SCMs share the same observational distribution but have different counterfactual distributions. Despite being observationally indistinguishable, the CausalNF method consistently converges to one of the two SCMs. If one hand-picks the "easier" of these SCMs, they may report an overly optimistic estimate of performance.
>
> This illustrates the broader issue: evaluations based on a small number of manually selected SCMs can lead to misleading conclusions, as performance may vary drastically depending on these idiosyncratic choices. Such evaluations are inherently fragile and insufficient for drawing generalizable insights. In contrast, the SoI approach enables evaluators to sample over a diverse, explicitly defined family of SCMs and queries, helping to reduce the influence of idiosyncratic choices.
>
> Moreover, hand-crafted SCMs can also implicitly tailor to certain method assumptions or strengths. Without a formal specification of the underlying assumptions (e.g., functional form, graph sparsity), such evaluations can introduce subtle but impactful biases in favor of one method over another. SoIs, by design, make these assumptions explicit and allow them to be varied systematically.
> We are happy to revise the introduction to expand on this motivation.
>
> **Weakness 2: The lack of real data is the bottleneck, not the lack of synthetic data generation capabilities.**
>
> We do agree that the lack of real datasets constitutes a major limitation in the evaluation of Causal ML methods. However, as argued in [1], gathering more real data is not sufficient, and synthetic data plays a central role in the evaluation of Causal ML methods. Notably, it is impossible, because of the fundamental problem of causal inference [2], to access real counterfactual data. In addition, even if one is interested in intervention data that can be collected in specific settings, several common causal assumptions (such as faithfulness or sufficiency) are unfalsifiable. Hence, the only way to guarantee that such assumptions hold (or not) is to use synthetic data to distinguish identifiable from unidentifiable cases. As a result, we do believe that using more sophisticated synthetic data enables one to run more rigorous and extensive synthetic experiments instead of focusing on illustrative examples. Moreover, we would like to clarify that we do not pretend that CausalProfiler is sufficient to characterize the performance of Causal ML methods completely. It is not intended to replace any real data experiment but rather to provide new ways of performing systematic, repeatable, and transparent evaluations. We will clarify these points in the introduction and limitations sections.
>
> Regarding the simulation-to-real gap, we refer to the experimental results included in our response to Reviewer Mc7P (Weakness 2), which compare our generated datasets to existing semi-synthetic datasets.
>
> **Weakness 3: Theorem 5.1 feels trivial**
>
> Our intention with Theorem 5.1 was not to provide a deep theoretical contribution, but rather to formalize a minimal guarantee: under sufficiently expressive discrete mechanisms, any causal dataset within a given SoI has strictly positive probability of being generated. We agree this is modest, and to reflect its role, we are happy to rename it to a **Proposition** in the final version.
>
> That said, we respectfully clarify that this guarantee is not trivial (because of mechanism sampling). Even within "reasonable" frameworks, coverage is not always ensured. For instance, one might implement a discrete mechanism sampling strategy that seems reasonable—such as the Unbiased Random Assignment strategy described in Appendix D—but will fail to achieve coverage. Moreover, the coverage guarantee does not trivially extend to continuous SCMs, even if we use universal function approximators (e.g., randomly initialized NNs) for mechanisms. Universal approximation only ensures the ability to approximate any function arbitrarily well in the limit, but this is not sufficient for coverage in a sampling sense (the probability of sampling a specific function may be zero). As such, we believe it's worth making any coverage result explicit, while agreeing that it may be more appropriate to present it as a weaker result.
>
> **Weakness 4: Unaddressed distributional bias**
>
> CausalProfiler was created to facilitate the evaluation of Causal ML methods while advocating for transparency. Hence, we believed it was essential to warn users about the limitations and challenges involved in data generation, to help prevent over-interpretation of results or misleading conclusions. In that sense, we agree that distributional bias is not corrected in the current version of CausalProfiler—but we explicitly bring attention to it and provide an initial empirical analysis, rather than leaving this concern unexplored.
>
> We acknowledge that the description of results in Appendix G needs improvement, and we will update the paper to clarify the main takeaways. Among the analyzed SCMs, weak positivity is always satisfied, while strong positivity holds in ~6% of cases.
> According to our analysis, there does not appear to be a correlation between the cardinality of endogenous variables and whether the strong positivity assumption is met. Instead, the number of variables plays a larger role, which is expected, as the number of possible observations increases exponentially with the number of variables.
>
> Regarding confounded components, our results show that components of size greater than 2 only arise when the proportion of hidden variables, the edge probability, and the number of variables are sufficiently large. In practice, this means users must anticipate that a confounded component is created whenever a variable with more than two children is defined as unobserved. For instance, among graphs with a proportion of hidden variables of 0.3, 6% of the SCMs have confounded components of size strictly greater than 2. In conclusion, a small but non-negligible number of the sampled SCMs in Appendix G satisfy the strong positivity assumption and are highly confounded. As a result, a weighting strategy could be used to balance this distribution by giving more weight to the underrepresented SCMs. As noted in Reviewer c5Yz (Feedback 1), users could also define multiple SoIs, each designed to satisfy different properties, and then aggregate these SoIs to create a more balanced distribution.
>
> Reducing distributional bias is an important future research direction. However, we would like to emphasize that achieving a perfectly balanced (e.g., uniform) distribution over all metrics is inherently impossible. For instance, one might want to have perfect uniform sampling over all discrete functions for the causal mechanisms. However, this would inherently bias the sampling toward non-bijective functions, as bijective functions are not dense in the function space. As a result, different sampling algorithms may be needed depending on the type of balance or guarantees one wishes to enforce. Hence, we also encourage the community to contribute to the CausalProfiler repository by implementing and sharing alternative sampling algorithms aligned with their own goals for bias reduction.
>
> **Feedback 1: Justification for 1000 samples**
>
> While we did not perform an exhaustive grid search over all SoI parameters, we did experiment with several dataset sizes—50, 100, 200, 1000, and 2000.
>
> For the experiments included in the paper, our intent was not to conduct a deep empirical study of performance versus sample size, but rather to illustrate the types of empirical questions the framework can help answer. Accordingly, we selected 1000 samples as a default value that was sufficient for the SoIs we tested. We varied it only to explore the impact of data availability explicitly (Exp. 1).
>
> **Feedback 2: Why are errors lower for NN SCMs than linear ones?**
>
> This is indeed surprising. While we did not perform a dedicated ablation to isolate this effect, we offer a few hypotheses that may help explain it.
>
> First, the evaluated methods (Exp. 1) are themselves neural architectures. When the underlying SCM mechanisms are also neural networks, these models may benefit from an inductive bias match, leading to better fitting.
> Second, although neural networks are highly expressive, when their weights are randomly initialized (e.g., He initialization, as in our experiment), they tend to produce smooth functions with low-frequency variation [3]. Such functions may be easier to learn from limited data, as they exhibit gradual changes rather than sharp transitions. In contrast, some linear mechanisms—especially those with large or poorly conditioned coefficients—can produce high-variance outputs or abrupt changes in response to small input differences, making them more difficult to estimate reliably from finite samples.
>
> We hope our explanations are clear, and they could make you reassess the evaluation of our work. Please let us know if you have any additional questions or concerns.
>
> [1] Audrey Poinsot, et al. Position: Causal machine learning requires rigorous synthetic experiments for broader adoption. International Conference on Machine Learning Position Paper Track, 2025
>
> [2] Paul W. Holland. Statistics and causal inference. Journal of the American Statistical Association, 1986
>
> [3] Nasim Rahaman, et al. On the Spectral Bias of Neural Networks. International Conference on Machine Learning, 2019

---

> > ### Comment · Reviewer_8e2f · 2025-08-05
> >
> > Thank you for the detailed response and clarification. Below are my remaining questions regarding your response:
> >
> > *  The observationally equivalent SCM example demonstrates how naive cherry-picking can mislead, but my questions are
> >    - __How often do such cases occur?__ If systematic evaluation only occasionally finds problems that careful targeted testing misses, does the benefit justify building and using this complex framework?
> >    - __Can targeted approaches be easily improved?__ If researchers know to look for these failure modes (once identified), couldn't they simply include them in future targeted evaluations without requiring the full framework?
> > * You acknowledge that only 6% of generated SCMs satisfy basic assumptions (strong positivity), requiring users to extensively tune parameters to get meaningful results. If users must spend significant effort on tuning SoI parameters to generate useful SCMs, __how is this easier or better than researchers simply creating a few well-designed synthetic experiments tailored to their evaluation needs?__
> > * Can you elaborate more on the sensitivity analysis on the different dataset sizes?

---

> > > ### Author Response · Authors · 2025-08-06
> > >
> > > We thank the reviewer for continuing the conversation and for the follow-up questions.
> > >
> > > **Q1a: How often do such cases occur?**
> > >
> > > If the question refers to how often observationally equivalent (L1-eq) SCMs with different interventional or counterfactual values occur, the answer is: whenever the query is not identifiable. Within the identification domain, all L1-eq SCMs yield the same query value. Outside it, there may be infinitely many L1-eq SCMs with differing values. In practice, violations of assumptions like strong positivity or sufficiency are common, so these cases are not rare from a practitioner's perspective.
> > >
> > > If the question concerns how often misleading cherry-picking occurs in evaluation, to the best of our knowledge, there is no systematic quantification in the literature, though multiple works [1,10,11,12] point out the risks of relying on a few hand-picked datasets.
> > >
> > > Cherry-picking can be sufficient for illustration in a controlled example, but it is inadequate for validating that a property (e.g., identifiability) holds across a class of SCMs, or for exploring classes without guarantees. Indeed, manual selection covers only narrow regions. In contrast, SoIs let us explore beyond what "careful" testing might anticipate—especially for failure modes that arise not from known risks, but from unexpected ones.
> > >
> > > The value of CausalProfiler lies in enabling controlled exploration of diverse scenarios, promoting transparency and reproducibility, and allowing stress testing to assess robustness.
> > >
> > >
> > > **Q1b: Can targeted approach be easily improved once identified?**
> > >
> > > Yes, one potential use of CausalProfiler is to discover such failure modes so they can be added to future targeted evaluations.
> > > However, the benefit of CausalProfiler is not just finding isolated "failure" examples—it is to understand trends across variation, which cannot be inferred from a handful of curated examples. CausalProfiler can generate such settings automatically, with guaranteed diversity and control. We see it not as replacing targeted testing, but as complementing it. Notably, CausalProfiler can help researchers to:
> > > - Find unexpected strengths: identify SCMs where a method performs surprisingly well outside the identification domain, suggesting opportunities to relax assumptions
> > > - Expose weaknesses: locate SCMs where a method consistently fails, helping practitioners understand limitations and guiding the development of improved methods
> > > - Characterize heterogeneity: detect SCMs where performance varies widely (e.g., the example from [1] Section 3.2)
> > >
> > > **Q2: Given the 6%, is SoI tuning easier than well-designed custom experiments?**
> > >
> > > The reported 6% of SCMs satisfying strong positivity should be interpreted as a conservative lower bound. Our check uses finite samples, while strong positivity is defined in the infinite-sample regime. We flagged a violation whenever any configuration had an empirical frequency of 0 in 10k samples. Additionally, the 6% figure applies only to specific SoIs we tested—regional discrete SCMs with fewer than 50 noise regions—where strong positivity is harder to satisfy. In contrast, SoIs with linear mechanisms and strictly positive exogenous distributions (e.g., Gaussian) satisfy strong positivity by construction. Strong positivity could also be enforced by applying rejection sampling—discarding SCMs that fail the criterion—at the cost of additional sampling time.
> > >
> > > More generally, what constitutes useful SCMs depends on the user's goals. Some may wish to violate strong positivity to study the non-identifiable domain, while others may require it to hold in all the SCMs to remain in the identification regime. CausalProfiler is not intended for cases where only a few well-designed hand-crafted experiments are needed. Its value emerges when exploring a broad space of SCMs: users can declaratively specify or filter for assumptions of interest, and all sampling biases are explicit. This makes it a scalable and reproducible alternative to manual SCM design, especially when evaluations need to be repeated, varied, or extended across studies.
> > >
> > > **Q3: More detail on sensitivity to dataset size?**
> > >
> > > In a preliminary analysis, we performed a partial grid sweep over dataset sizes: 50, 100, 200, 1000, and 2000. Broadly:
> > > - Until 200: noisy estimates and sometimes unstable
> > > - 1000: good trade-off between stability and runtime
> > > - 2000: marginal improvement at extra computational cost
> > >
> > > Based on these exploratory runs (not highlighted in the paper as they did not constitute a sufficiently comprehensive study), we chose 1000 samples as a default that was stable for the tested SoIs.
> > >
> > > [10] Rachel Longjohn et al. Benchmark Data Repositories for Better Benchmarking. NeurIPS. 2024
> > >
> > > [11] Alicia Curth et al. Really doing great at estimating CATE? A critical look at ML benchmarking practices in treatment effect estimation. NeurIPS. 2021
> > >
> > > [12] Lu Cheng et al. Evaluation Methods and Measures for Causal Learning Algorithms. IEEE. 2022

---

> > > > ### Comment · Reviewer_8e2f · 2025-08-08
> > > >
> > > > Thank you for your additional clarification. I will raise my score by 1.

---

### Decision · Program_Chairs · 2025-09-18

**Decision:**

Reject

**Comment:**

This work proposes a scalable approach to the generation of synthetic datasets for benchmarking approaches to causal estimation. The core of the argument for the work is that many existing evaluations of causal estimation approaches are based on a small number of datasets (either synthetic, semi-synthetic, or real-world derived) that may be considered arbitrary or irrelevant with respect to the space of data generating processes of interest. This work proposes to instead generate datasets consistent with a user-specified “Space of Interest” (SoI), which captures the specification of the class of relevant SCMs, class of the causal queries, class of data, as well as other parameters. This then allows for benchmarking of approaches over a broader class of relevant data generating processes.

The reviews for the work were positive overall. The reviewers generally agreed that the work addresses a key gap in the evaluation of causal estimation approaches. They further agreed that the system design is rigorous and that the framework can help facilitate more transparent and robust evaluations. Furthermore, the experiments provide some insight into the performance of state of the art causal ML methods.

There are a few recurring issues that arose during the discussion period that warrant mention:
* Reviewers 8e2f and c5Yz note that the generator has some unaddressed distributional biases that are not fully characterized or mitigated. This is an important concern because it suggests that the distribution of sampled datasets may exhibit some arbitrariness over the distribution of nuisance factors of relevance to the problem. Arguably, it may be the case that in using this framework, the user trades one arbitrary set of status quo evaluations of limited relevance to their problem for another. These concerns are noted by the authors in the response as an important area for future research and one that they intend to discuss in some further detail in the discussion section of the paper.
* There is limited guidance for practitioners that are interested in translating their domain knowledge regarding a given problem into a specification of an SoI.
* Several reviewers highlighted that the paper would be stronger if the empirical findings regarding the relative effectiveness of estimators could be compared to those that results from experiments with commonly used real-world datasets (e.g., IHDP and Twins). In the response, the authors agreed that such a comparison would enhance the work, but did not address the concern directly through new experiments. One challenge raised by the authors is that it is difficult to translate the implicit assumptions underlying a given dataset into an SoI specification so as to enable meaningful comparison of algorithms in a comparable context. As this is an important limitation in its own right that is worth mentioning, I recommend that the authors emphasize this in future revision of the work.

Overall, I recommend that this work be accepted. Furthermore, I recommend that the authors carefully revise the paper to clarify the scope of the work and its limitations in line with the points raised during the review process.

===== FINAL UPDATE FROM DB Track PCs ====

The final decision for this paper has been taken by the program chairs after consultation with the SACs. All Senior Area Chairs have ranked papers according to the feedback from the AC during the review process. We decided to leave the original meta-review to reflect the opinion of the AC in light of the initial discussions with reviewers and SAC.